# DIFFUSION-PINN SAMPLER

## ABSTRACT

Recent success of diffusion models has inspired a surge of interest in developing sampling techniques using reverse diffusion processes. However, accurately estimating the drift term in the reverse stochastic differential equation (SDE) solely from the unnormalized target density poses significant challenges, hindering existing methods from achieving state-of-the-art performance. In this paper, we introduce the *Diffusion-PINN Sampler* (DPS), a novel diffusion-based sampling algorithm that estimates the drift term by solving the governing partial differential equation of the log-density of the underlying SDE marginals via physics-informed neural networks (PINN). We prove that the error of log-density approximation can be controlled by the PINN residual loss, enabling us to establish convergence guarantees of DPS. Experiments on a variety of sampling tasks demonstrate the effectiveness of our approach, particularly in accurately identifying mixing proportions when the target contains isolated components.

## 1 INTRODUCTION

Sampling from unnormalized distributions is a fundamental yet challenging task encountered across various scientific disciplines such as Bayesian statistics, computational physics, chemistry, and biology (Liu & Liu, 2001; Stoltz et al., 2010). Markov chain Monte Carlo (MCMC) and variational inference (VI) have historically been the go-to methods for this problem. However, these approaches exhibit limitations when dealing with complex target distributions (e.g., distributions with multimodality or heavy tails). Recently, the success of diffusion models for generative modeling (Song et al., 2020b; Ho et al., 2020; Nichol & Dhariwal, 2021; Kingma et al., 2021) have sparked considerable interest in tackling the sampling problem using the reverse diffusion processes that transport a given prior density to the target, governed by stochastic differential equations (SDE).

In diffusion-based generative models, the score function in the drift term of the reverse SDE is learned based on score matching techniques (Hyvärinen & Dayan, 2005; Vincent, 2011) that require samples from the target data distribution. However, for sampling tasks, we only have access to an unnormalized density function $\pi$, making it challenging to estimate the score function for the reverse SDE. From a stochastic optimal control perspective (Tzen & Raginsky, 2019; De Bortoli et al., 2021), several VI methods that parameterize the drift term with neural network approximation have been proposed (Zhang & Chen, 2021; Berner et al., 2022; Vargas et al., 2023b;a). Nevertheless, these approaches face challenges such as instability during training, the computational complexity associated with differentiating through SDE solvers, and mode collapse issues arising from training objectives based on reverse Kullback-Leibler (KL) divergences (Zhang & Chen, 2021; Vargas et al., 2023a). On the other hand, Huang et al. (2023) proposed a scheme based on the connection between score matching and non-parametric posterior mean estimation. More specifically, they use MCMC estimation of the scores to potentially alleviate the numerical bias intrinsic in parametric estimation methods such as neural networks. However, this method also introduces noise in the estimates and requires repetitive posterior sampling in each time step of the reverse SDE. Overall, despite their potential, diffusion-based sampling methods have not yet achieved state-of-the-art performance.

In addition to its connection with posterior mean estimation, the score function has also been shown to evolve according to a partial differential equation known as the *score Fokker-Planck equation* (score FPE) (Lai et al., 2023). This discovery has led to a novel regularization technique for enhancing score function estimation in diffusion models (Lai et al., 2023; Deveney et al., 2023). In this paper, we adopt this strategy for diffusion-based sampling methods. While the score function can be recovered by solving the score FPE using the score of target distribution $\pi$ as the initial condition, we demonstrate

that it may fail to identify correct mixing proportions when $\pi$ has isolated components, a common limitation known as the blindness of score-based methods (Wenliang, 2020; Zhang et al., 2022). To remedy this issue, we propose to solve the log-density FPE, a similar partial differential equation for the log-density function, using physics-informed neural networks (PINN) (Raissi et al., 2019; Wang et al., 2022). The estimated log-density function is then integrated into the reverse SDE, leading to a novel sampling algorithm termed *Diffusion-PINN Sampler* (DPS). We prove that the error of log-density estimation can be controlled by the PINN residual loss, which allows us to obtain convergence guarantee of DPS based on established results for score-based generative models (Chen et al., 2023b;a; Benton et al., 2023). Experiments on a variety of sampling tasks provide compelling numerical evidence for the superiority of our method compared to other baseline methods.

## 2 RELATED WORKS

Recently, several works have explored the combination of Physics-Informed Neural Networks (PINN) and sampling techniques. For instance, Máté & Fleuret (2023); Fan et al. (2024); Tian et al. (2024) address the continuity equation using PINN based on ODEs and achieve flow-based sampling through a linear interpolation (i.e., annealing) path between the target distribution and a simple prior, such as a Gaussian distribution. Besides, Berner et al. (2022) (in the appendix of their paper) and Sun et al. (2024) propose solving the log-density Hamilton–Jacobi–Bellman (HJB) equation via PINN to develop a SDE-based sampling algorithm. However, both approaches lack comprehensive numerical investigation and thorough theoretical analysis. In contrast, our work investigates a limitation of score-based Fokker-Planck equations (FPE) in identifying the mixing proportions of multi-modal distributions, introduces novel computational techniques for solving PDEs via PINN in the context of diffusion-based sampling, and provides the first complete theoretical analysis of the algorithm. See more discussion about related works in Appendix D.

## 3 BACKGROUND

**Notations.** Throughout the paper, $\Omega \subset \mathbb{R}^d$ denotes a bounded and closed domain. For simplicity, we do not distinguish a probabilistic measure from its density function. We use $\boldsymbol{x} = (x_1, \cdots, x_d)'$ to denote a vector in $\mathbb{R}^d$ and $\|\boldsymbol{x}\| = \sqrt{x_1^2 + \cdots + x_d^2}$ stands for the $L^2$-norm. Let $\nu$ denote a probability measure on $\mathbb{R}^d$, for any $\boldsymbol{f} : \mathbb{R}^d \to \mathbb{R}^m$, we denote $\|\boldsymbol{f}(\cdot)\|_{L^2(\Omega;\nu)}^2 := \int_\Omega \|\boldsymbol{f}(\boldsymbol{x})\|^2 \, \mathrm{d}\nu(\boldsymbol{x})$. For any $\boldsymbol{f} : \mathbb{R}^d \times [0, T] \to \mathbb{R}^m$, we define $\|\boldsymbol{f}_t(\cdot)\|_{L^2(\Omega;\nu)}^2 := \int_\Omega \|\boldsymbol{f}_t(\boldsymbol{x})\|^2 \, \mathrm{d}\nu(\boldsymbol{x})$ as a function of $t \in [0, T]$. For any $\boldsymbol{F} = (F_1, \cdots, F_d)' : \mathbb{R}^d \to \mathbb{R}^d$, we denote the divergence of $\boldsymbol{F}$ by $\nabla \cdot \boldsymbol{F} := \sum_{i=1}^d \partial_{x_i} F_i$. For any $F : \mathbb{R}^d \to \mathbb{R}$, we denote the Laplacian of $F$ by $\Delta F := \sum_{i=1}^d \partial_{x_i}^2 F$.

**Diffusion models.** In diffusion models, noise is progressively added to the training samples via a forward stochastic process described by the following stochastic differential equation (SDE)

$$\mathrm{d}\boldsymbol{x}_t = \boldsymbol{f}(\boldsymbol{x}_t, t) \, \mathrm{d}t + g(t) \, \mathrm{d}\boldsymbol{B}_t, \quad \boldsymbol{x}_0 \sim p_0(\cdot), \quad 0 \leqslant t \leqslant T, \tag{1}$$

where $p_0(\cdot)$ is the data distribution, $\boldsymbol{B}_t$ is a standard Brownian motion, and $\boldsymbol{f}(\boldsymbol{x}_t, t)$ and $g(t)$ are the drift and diffusion coefficients respectively. The derivatives of the log-density of the forward marginals, i.e., *scores*, are learned via score matching techniques (Vincent, 2011; Song et al., 2020b) and new samples from the data distribution can be obtained by simulating the following reverse process

$$\mathrm{d}\boldsymbol{x}_t = \left[\boldsymbol{f}(\boldsymbol{x}_t, t) - g^2(t)\nabla_{\boldsymbol{x}_t} \log p_t(\boldsymbol{x}_t)\right] \, \mathrm{d}t + g(t) \, \mathrm{d}\bar{\boldsymbol{B}}_t, \quad \boldsymbol{x}_T \sim p_T(\cdot), \tag{2}$$

where $p_t(\cdot)$ is the probability density of $\boldsymbol{x}_t$ and $\bar{\boldsymbol{B}}_t$ is a standard Brownian motion from $T$ to 0.

**Physics-informed neural networks (PINN).** PINN is a deep learning method for solving partial differential equations (PDEs) (Raissi et al., 2019). Consider the following general form of PDE

$$\mathcal{L}u(\boldsymbol{x}) = \varphi(\boldsymbol{x}), \quad \boldsymbol{x} \in \Omega \subseteq \mathbb{R}^d, \tag{3a}$$

$$\mathcal{B}u(\boldsymbol{x}) = \psi(\boldsymbol{x}), \quad \boldsymbol{x} \in \partial\Omega, \tag{3b}$$

where $\mathcal{L}$ and $\mathcal{B}$ are the differential operators on domain $\Omega$ and boundary $\partial\Omega$, respectively. PINN seeks an approximate solution using deep model $u_\theta(\boldsymbol{x})$ by minimizing the $L^2$ PINN residual losses

$$\ell_\Omega(u_\theta) := \|\mathcal{L}u_\theta(\boldsymbol{x}) - \varphi(\boldsymbol{x})\|^2_{L^2(\Omega;\nu)}, \tag{4a}$$

$$\ell_{\partial\Omega}(u_\theta) := \|\mathcal{B}u_\theta(\boldsymbol{x}) - \psi(\boldsymbol{x})\|^2_{L^2(\Omega;\nu)}, \tag{4b}$$

where $\nu$ is a probability measure for collocation point generation, often taken to be the uniform distribution on $\Omega$. The two terms $\ell_\Omega(u)$ and $\ell_{\partial\Omega}(u)$ in Eq. (4) reflect the approximation error on $\Omega$ and $\partial\Omega$ respectively. In practice, the losses in Eq. (4) can be optimized by gradient-based methods with Monte Carlo gradient estimation.

**Fokker-Planck equation.**  The evolution of the density $p_t(\boldsymbol{x})$ associated with the forward SDE (1) is governed by the Fokker-Planck equation (FPE) (Øksendal, 2003)

$$\partial_t p_t(\boldsymbol{x}) = \underbrace{\frac{1}{2}g^2(t)\Delta p_t(\boldsymbol{x}) - \nabla \cdot [\boldsymbol{f}(\boldsymbol{x},t)p_t(\boldsymbol{x})]}_{:=\mathcal{L}_{\text{FPE}}p_t(\boldsymbol{x})}. \tag{5}$$

Recently, Lai et al. (2023) derive an equivalent system of PDEs for the log density $\log p_t(\boldsymbol{x})$ and score $\nabla_{\boldsymbol{x}} \log p_t(\boldsymbol{x})$, termed as the log-density Fokker-Planck equation (log-density FPE) and the score Fokker-Planck equation (score FPE) respectively, as summarized in Theorem 1 (the proof can be found in Appendix A.1).

**Theorem 1** (Log-density FPE and score FPE; Proposition 3.1 in Lai et al. (2023)). *Assume the density $p_t(\boldsymbol{x})$ is sufficiently smooth on $\mathbb{R}^d \times [0,T]$. Then for all $(\boldsymbol{x},t) \in \mathbb{R}^d \times [0,T]$, the log-density $u_t(\boldsymbol{x}) := \log p_t(\boldsymbol{x})$ satisfies the PDE*

$$\partial_t u_t(\boldsymbol{x}) = \underbrace{\frac{1}{2}g^2(t)\Delta u_t(\boldsymbol{x}) + \frac{1}{2}g^2(t)\|\nabla_{\boldsymbol{x}} u_t(\boldsymbol{x})\|^2 - \boldsymbol{f}(\boldsymbol{x},t) \cdot \nabla_{\boldsymbol{x}} u_t(\boldsymbol{x}) - \nabla \cdot \boldsymbol{f}(\boldsymbol{x},t)}_{:=\mathcal{L}_{\text{L-FPE}}u_t(\boldsymbol{x})}, \tag{6}$$

*and the score $\boldsymbol{s}_t(\boldsymbol{x}) := \nabla_{\boldsymbol{x}} \log p_t(\boldsymbol{x})$ satisfies the PDE*

$$\partial_t \boldsymbol{s}_t(\boldsymbol{x}) = \underbrace{\nabla_{\boldsymbol{x}}\left[\frac{1}{2}g^2(t)\nabla \cdot \boldsymbol{s}_t(\boldsymbol{x}) + \frac{1}{2}g^2(t)\|\boldsymbol{s}_t(\boldsymbol{x})\|^2 - \boldsymbol{f}(\boldsymbol{x},t) \cdot \boldsymbol{s}_t(\boldsymbol{x}) - \nabla \cdot \boldsymbol{f}(\boldsymbol{x},t)\right]}_{:=\mathcal{L}_{\text{S-FPE}}\boldsymbol{s}_t(\boldsymbol{x})}. \tag{7}$$

## 4 DIFFUSION-PINN SAMPLER

We consider sampling from a probability density $\pi(\boldsymbol{x}) = \mu(\boldsymbol{x})/Z$ with $\boldsymbol{x} \in \mathbb{R}^d$, where $\mu(\boldsymbol{x})$ has an analytical form and $Z = \int_{\mathbb{R}^d} \mu(\boldsymbol{x})\mathrm{d}\boldsymbol{x}$ is the intractable normalizing constant. Throughout, we only consider the forward process (1) with an explicit conditional density of $\boldsymbol{x}_t|\boldsymbol{x}_0 \sim \pi_{t|0}(\cdot|\boldsymbol{x}_0)$. We denote by $\pi_t$ the marginal density of $\boldsymbol{x}_t$ associated with (1) from $\boldsymbol{x}_0 \sim \pi_0 = \pi$.

Inspired by diffusion models, sampling can be performed by simulating a reverse process (8) targeting at $\pi(\boldsymbol{x})$, given an accurate estimate of the perturbed scores $\boldsymbol{s}_\theta(\boldsymbol{x},t) \approx \nabla_{\boldsymbol{x}} \log \pi_t(\boldsymbol{x})$,

$$\mathrm{d}\boldsymbol{x}_t = \left[\boldsymbol{f}(\boldsymbol{x}_t,t) - g^2(t)\boldsymbol{s}_\theta(\boldsymbol{x}_t,t)\right]\mathrm{d}t + g(t)\,\mathrm{d}\bar{\boldsymbol{B}}_t, \quad \boldsymbol{x}_T \sim \pi_{\text{prior}}, \tag{8}$$

where $\pi_{\text{prior}}$ denotes the stationary distribution of the forward process (1) and $T$ is large enough such that $\pi_T \approx \pi_{\text{prior}}$. However, unlike generative models, sampling tasks lack training data from $\pi$, which hinders the application of denoising score matching for perturbed score estimation. In this section, we propose to solve the log-density FPE (6) with PINN to estimate the perturbed scores. While the score FPE can also be used for this purpose, we find that it may fail to learn the mixing proportions properly when the target contains isolated modes.

### 4.1 FAILURE OF SCORE FPE

Consider the case where the target is a mixture of Gaussians (MoG) with two distant modes. The following example shows that, for two MoGs with the same modes but different weights, the Fisher divergence between them can be arbitrarily small but the KL divergence between them remains large when the two modes are sufficiently separated. See Figure 1 (left) for an illustration of this phenomenon. More general theoretical results can be found in Appendix A.2.

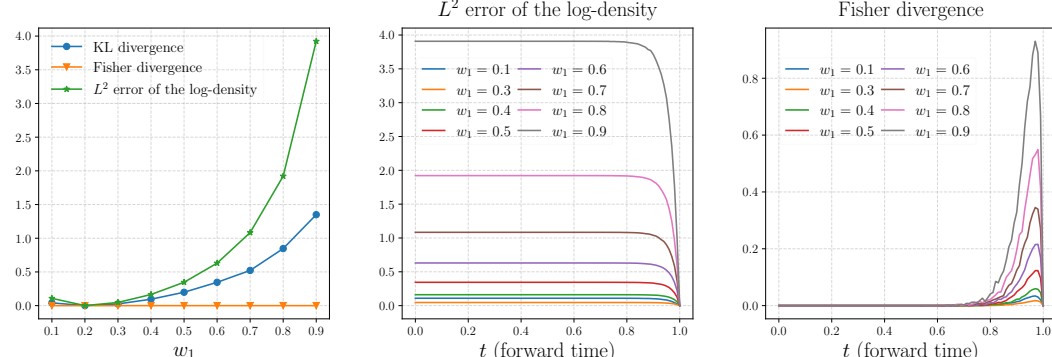

Figure 1: **Left**: KL divergence, Fisher divergence, and log-density error between $\pi^M$ and $\hat{\pi}^M$ as functions of $w_1$, where $\hat{w}_1 = 0.2$ and $\boldsymbol{a} = (-5, -5)'$. **Middle/Right**: The evolution of log-density error/Fisher divergence along the forward process respectively. The forward process achieves standard Gaussian at $t = 1$.

**Example 1.** *For any $\tau > 0$, there exists $M_\tau(d) > 0$ such that the following holds. For every $\boldsymbol{a} \in \mathbb{R}^d$ satisfied $\|\boldsymbol{a}\| \geqslant M_\tau(d)$, $w_1, w_2, \hat{w}_1, \hat{w}_2 \geqslant 0.1$, $w_1 + w_2 = 1$, and $\hat{w}_1 + \hat{w}_2 = 1$, MoG $\pi^M = w_1 \mathcal{N}(\boldsymbol{a}, I_d) + w_2 \mathcal{N}(-\boldsymbol{a}, I_d)$ and $\hat{\pi}^M = \hat{w}_1 \mathcal{N}(\boldsymbol{a}, I_d) + \hat{w}_2 \mathcal{N}(-\boldsymbol{a}, I_d)$ satisfy*

$$\mathrm{KL}(\pi^M \| \hat{\pi}^M) \geqslant w_1 \log \frac{w_1}{\hat{w}_1} + w_2 \log \frac{w_2}{\hat{w}_2} - \tau, \text{ but } F(\pi^M, \hat{\pi}^M) < \tau, \tag{9}$$

*where $F(\pi^M, \hat{\pi}^M)$ denotes the Fisher divergence between $\pi^M$ and $\hat{\pi}^M$ defined as*

$$F(\pi^M, \hat{\pi}^M) := \mathbb{E}_{\boldsymbol{x} \sim \pi^M} \left[ \left\| \nabla_{\boldsymbol{x}} \log \pi^M(\boldsymbol{x}) - \nabla_{\boldsymbol{x}} \log \hat{\pi}^M(\boldsymbol{x}) \right\|^2 \right].$$

### 4.1.1 SOLVING SCORE FPE STRUGGLES TO LEARN THE WEIGHTS

Let $\pi^M, \hat{\pi}^M$ be the MoGs in Example 1. For any $t \in [0, T]$, $\pi_t^M$ denotes the marginal distribution of $\boldsymbol{x}_t$ associated with the forward process (1) from $\boldsymbol{x}_0 \sim \pi^M$. We denote $\boldsymbol{s}_t^M(\boldsymbol{x}) := \nabla_{\boldsymbol{x}} \log \pi_t^M(\boldsymbol{x})$ which is the solution to (7) with $\boldsymbol{s}_0^M(\boldsymbol{x}) = \nabla_{\boldsymbol{x}} \log \pi^M(\boldsymbol{x})$. $\hat{\pi}_t^M$ and $\hat{\boldsymbol{s}}_t^M(\boldsymbol{x})$ are defined similarly.

Consider solving score-FPE (7) using the following PINN residual loss

$$\ell_{\text{S-res}}(\boldsymbol{s}; \boldsymbol{x}, t) := \left\| \partial_t \boldsymbol{s}_t(\boldsymbol{x}) - \mathcal{L}_{\text{S-FPE}} \boldsymbol{s}_t(\boldsymbol{x}) \right\|^2, \tag{10}$$

Though $\pi^M$ and $\hat{\pi}^M$ are equipped with different weights, their scores both satisfy the PDE (7) such that $\ell_{\text{S-res}}(\boldsymbol{s}^M; \boldsymbol{x}, t) = \ell_{\text{S-res}}(\hat{\boldsymbol{s}}^M; \boldsymbol{x}, t) = 0$ for any $(\boldsymbol{x}, t) \in \mathbb{R}^d \times [0, T]$. The PINN approach, therefore, can only distinguish $\pi^M$ and $\hat{\pi}^M$ through the initial condition. However, Example 1 shows that the difference between $\boldsymbol{s}_0^M(\boldsymbol{x})$ and $\hat{\boldsymbol{s}}_0^M(\boldsymbol{x})$ can be arbitrarily small, indicating the difficulty of correctly identifying the weights by solving the score FPE. Figure 1 (right) shows that the perturbed score can not tell the difference of weights until the every end of the forward process. On the other hand, it is noticeable that the perturbed log-density distinguishes the weights well throughout the forward process (Figure 1, middle). This suggests us to solve log-density FPE and compute the scores by taking the gradient of the approximated log-density.

### 4.2 SOLVING LOG-DENSITY FPE

To estimate the perturbed scores, we consider solving log-density FPE with initial condition:

$$\partial_t u_t(\boldsymbol{x}) = \mathcal{L}_{\text{L-FPE}} u_t(\boldsymbol{x}), \tag{11a}$$

$$u_0(\boldsymbol{x}) = \log \mu(\boldsymbol{x}), \tag{11b}$$

where the exact solution is $u_t^*(\boldsymbol{x}) = \log \mu_t(\boldsymbol{x}) := \log \pi_t(\boldsymbol{x}) + \log Z$ (which induces the same score as $\nabla_{\boldsymbol{x}} u_t^*(\boldsymbol{x}) = \nabla_{\boldsymbol{x}} \log \pi_t(\boldsymbol{x})$). In what follows, we describe how to find an approximation $u_\theta(\boldsymbol{x}, t)$ to $u_t^*(\boldsymbol{x})$ within the PINN framework.

---

**Algorithm 1** : Solving log-density FPE via PINN

---

**Require:** Unnormalized density $\mu(\boldsymbol{x})$, the number of training iterations $N$, the number of samples used to estimate the training objective (13) $M$, the running time of the forward process (1) $T$.

1: Initialize the parameterized solution $u_\theta(\boldsymbol{x}, t)$ using target-informed parameterization (12).
2: **for** $n = 1, \cdots, N$ **do**
3:      Sample i.i.d. $t_i \sim \mathcal{U}[0, T], 1 \leqslant i \leqslant M$.
4:      Sample i.i.d. $\boldsymbol{x}_i^0 \sim \nu_0$ and $\boldsymbol{z}_i \sim \pi_{\text{prior}}, 1 \leqslant i \leqslant M$.
5:      Sample collocation points by the forward process (1): $\boldsymbol{x}_i^{t_i} \sim \pi_{t_i|0}(\cdot|\boldsymbol{x}_i^0), 1 \leqslant i \leqslant M$.
6:      Compute the training objective (13) by Monte Carlo estimation

$$L_{\text{MCMC}}(u_\theta) := \frac{1}{M} \sum_{i=1}^{M} \beta^2(t_i) \cdot \left\| \partial_t u_\theta(x_i^{t_i}, t_i) - \mathcal{L}_{\text{L-FPE}} u_\theta(x_i^{t_i}, t_i) \right\|^2 + \frac{\lambda}{M} \sum_{i=1}^{M} \ell_{\text{reg}}(u_\theta; T, \boldsymbol{z}_i). \tag{14}$$

7:      Gradient-based optimization: $\theta \leftarrow \texttt{Optimizer}(\theta, \nabla_\theta L_{\text{MCMC}}(u_\theta))$.
8: **end for**
9: **return** Parameterized solution $u_\theta(\boldsymbol{x}, t)$.

---

**Target-informed parameterization.** To incorporate the initial condition (11b), we use the following parameterization for the log-density function

$$u_\theta(\boldsymbol{x}, t) = \frac{T - t}{T} \log \mu(\boldsymbol{x}) + \frac{t}{T} \times \text{NN}_\theta(\boldsymbol{x}, t), \quad \forall (\boldsymbol{x}, t) \in \mathbb{R}^d \times [0, T], \tag{12}$$

where $\text{NN}_\theta(\boldsymbol{x}, t) : \mathbb{R}^d \times [0, T] \to \mathbb{R}$ is a deep neural network. This parameterization satisfies the initial condition (11b), thus we only need to consider the PINN residual loss induced by (11a). Similar strategy is also used in consistency models (Song et al., 2023). However, this parameterization might cause huge computation cost when querying the log-density of the target is expensive. To address this, see discussions in Section E.3.

**Underlying distribution for collocation points.** When training PINN, it is very important to collect proper collocation points $(\boldsymbol{x}_t, t) \in \mathbb{R}^d \times [0, T]$ where $\boldsymbol{x}_t \sim \nu_t$. We expect samples from $\nu_t$ to cover the high-density domain of $\pi_t$ where PINN can provide a good approximation. To achieve this, we first generate samples $\boldsymbol{x}_0 \sim \nu_0$ by running a short chain of Langevin Monte Carlo[1] (LMC) for $\pi$ so that $\nu_0$ covers the high density domain of $\pi$. Given $\boldsymbol{x}_0 \sim \nu_0$, we obtain $\boldsymbol{x}_t \sim \nu_t$ by sampling from the conditional distribution of the forward process given $\boldsymbol{x}_0$, namely, $\boldsymbol{x}_t|\boldsymbol{x}_0 \sim \pi_{t|0}(\cdot|\boldsymbol{x}_0)$.

**Training objective.** One useful property of the forward process (1) is that $\boldsymbol{x}_T \sim \pi_T \approx \pi_{\text{prior}}$ when $T$ is large. In practice, we may use this property to further regularize the PINN residual loss, leading to the following training objective:

$$L_{\text{train}}(u_\theta) := \mathbb{E}_{t \sim \mathcal{U}[0,T]} \mathbb{E}_{\boldsymbol{x}_t \sim \nu_t} \left[ \beta^2(t) \cdot \| \partial_t u_\theta(\boldsymbol{x}_t, t) - \mathcal{L}_{\text{L-FPE}} u_\theta(\boldsymbol{x}_t, t) \|^2 \right]$$
$$+ \lambda \cdot \mathbb{E}_{\boldsymbol{z} \sim \pi_{\text{prior}}} \left[ \ell_{\text{reg}}(u_\theta; T, \boldsymbol{z}) \right], \tag{13}$$

where $\ell_{\text{reg}}(u_\theta; T, \boldsymbol{z}) := \| \nabla_{\boldsymbol{z}} u_\theta(\boldsymbol{z}, T) - \nabla_{\boldsymbol{z}} \log \pi_{\text{prior}}(\boldsymbol{z}) \|^2$ denotes the regularization term, $\beta(t)$ is a weight function and $\lambda$ is a regularization coefficient. We seek a good approximation $u_\theta(\boldsymbol{x}, t)$ by minimizing (13) via stochastic optimization methods where the stochastic gradient is computed by Monte Carlo estimation. Our algorithm is summarized in Algorithm 1.

Once $u_\theta(\boldsymbol{x}, t)$ is learned, the induced score approximation is then substituted into the reverse process (8), resulting in a new variant of diffusion-based sampling method that we call *Diffusion-PINN Sampler* (DPS).

---

[1]In this paper, we utilize a parallel version of LMC. Namely, we obtain samples through running multiple separate LMC chains for each initial sample. This helps us use the divergence of initialization to enhance exploration.

**Hutchinson's trick for the gradient of the PINN residual.** Hutchinson's trace estimator provides a stochastic method for estimating the trace of any square matrix and is commonly used in Laplacian estimation. However, directly using Hutchinson's trick here can result in biased gradient estimation. To address this issue, we propose a novel variant of Hutchinson's trick that allows unbiased gradient estimation. Recall that the PINN residual can be decomposed as

$$\partial_t u_\theta - \mathcal{L}_{\text{L-FPE}} u_\theta := \underbrace{\partial_t u_\theta + \boldsymbol{f} \cdot \nabla_{\boldsymbol{x}} u_\theta + \nabla \cdot \boldsymbol{f} - \frac{g^2(t)}{2} \|\nabla_{\boldsymbol{x}} u_\theta\|^2}_{:= \mathcal{L}_I u_\theta} - \frac{g^2(t)}{2} \Delta u_\theta.$$

Using this decomposition, the PINN residual loss $\|\partial_t u_\theta - \mathcal{L}_{\text{L-FPE}} u_\theta\|^2$ has the following gradient,

$$\nabla_\theta \|\partial_t u_\theta - \mathcal{L}_{\text{L-FPE}} u_\theta\|^2 = 2 \left( \mathcal{L}_I u_\theta - \frac{g^2(t)}{2} \Delta u_\theta \right) \nabla_\theta \left( \mathcal{L}_I u_\theta - \frac{g^2(t)}{2} \Delta u_\theta \right)$$

$$= 2 \left( \mathcal{L}_I u_\theta - \frac{g^2(t)}{2} \cdot \mathbb{E}_{v_1} \left[ v_1^\top \nabla_{\boldsymbol{x}} \left( v_1^\top \nabla_{\boldsymbol{x}} u_\theta \right) \right] \right) \nabla_\theta \left( \mathcal{L}_I u_\theta - \frac{g^2(t)}{2} \cdot \mathbb{E}_{v_2} \left[ v_2^\top \nabla_{\boldsymbol{x}} \left( v_2^\top \nabla_{\boldsymbol{x}} u_\theta \right) \right] \right)$$

$$= \mathbb{E}_{v_1, v_2} \left[ 2 \left( \mathcal{L}_I u_\theta - \frac{g^2(t)}{2} \cdot v_1^\top \nabla_{\boldsymbol{x}} \left( v_1^\top \nabla_{\boldsymbol{x}} u_\theta \right) \right) \nabla_\theta \left( \mathcal{L}_I u_\theta - \frac{g^2(t)}{2} \cdot v_2^\top \nabla_{\boldsymbol{x}} \left( v_2^\top \nabla_{\boldsymbol{x}} u_\theta \right) \right) \right]$$

where $v_1$ and $v_2$ are independent and satisfy $\mathbb{E}_{v_1}[v_1 v_1^\top] = \mathbb{E}_{v_2}[v_2 v_2^\top] = \boldsymbol{I}_d$. Therefore, the following objective yields an unbiased gradient estimate of the PINN residual loss,

$$\mathbb{E}_{v_1, v_2} \left[ \text{Detach} \left( 2 \left( \mathcal{L}_I u_\theta - \frac{g^2(t)}{2} \cdot v_1^\top \nabla_{\boldsymbol{x}} \left( v_1^\top \nabla_{\boldsymbol{x}} u_\theta \right) \right) \right) \left( \mathcal{L}_I u_\theta - \frac{g^2(t)}{2} \cdot v_2^\top \nabla_{\boldsymbol{x}} \left( v_2^\top \nabla_{\boldsymbol{x}} u_\theta \right) \right) \right].$$

## 5 Theoretical Guarantees

**Notations.** Let us denote $e_t(\boldsymbol{x}) := u_\theta(\boldsymbol{x}, t) - u_t^*(\boldsymbol{x})$ and $r_t(\boldsymbol{x}) := \partial_t u_\theta(\boldsymbol{x}, t) - \mathcal{L}_{\text{L-FPE}} u_\theta(\boldsymbol{x}, t)$. For any $C \in \mathbb{R}$, $t \in [0, T]$, we define the weighted PINN objective on $\Omega$ as

$$L_{\text{PINN}}(t; C) := \int_0^t e^{C(t-s)} \|r_s(\cdot)\|_{L^2(\Omega; \nu_s)}^2 \, \mathrm{d}s, \tag{15}$$

where $\{\nu_t\}_{t=0}^T$ denotes the underlying distribution for collocation points introduced in Section 4.2 which satisfies the FPE $\partial_t \nu_t(\boldsymbol{x}) = \mathcal{L}_{\text{FPE}} \nu_t(\boldsymbol{x})$.

### 5.1 Approximation error of PINN for log-density FPE

In this section, we provide an upper bound on the approximation error of PINN for solving the log-density FPE (6) on a constrained domain $\Omega$. Namely, we control $\|e_t(\cdot)\|_{L^2(\Omega; \nu_t)}^2$ and $\|\nabla_{\boldsymbol{x}} e_t(\cdot)\|_{L^2(\Omega; \nu_t)}^2$ by the residual loss $\|r_t(\cdot)\|_{L^2(\Omega; \nu_t)}^2$ and the weighted PINN objective (15). We make the following assumptions.

**Assumption 1.** $u^*$ and $u_\theta$ are the same on the boundary, i.e., $u_t^*(\boldsymbol{x}) = u_\theta(\boldsymbol{x}, t)$ on $\partial \Omega \times [0, T]$.

**Assumption 2.** For any $t \in [0, T]$, $g^2(t)$ is bounded: $m_1 \leqslant g^2(t) \leqslant M_1$ for some $m_1, M_1 > 0$.

**Assumption 3.** $\log \nu_t(\boldsymbol{x}), u_t^*(\boldsymbol{x}), u_\theta(\boldsymbol{x}, t) \in \mathcal{C}^2(\Omega \times [0, T])$.

Assumption 1 is necessary for us to ensure the uniqueness of the solution to (6) on $\Omega$, which is also considered in Deveney et al. (2023); Wang et al. (2022). Assumption 2, 3 are also considered in Deveney et al. (2023). Based on Assumption 3, there exists $B_0^\nu, B_0^*, \widehat{B}_0, B_1^\nu, B_1^*, \widehat{B}_1 \in \mathbb{R}_+$ and $B_2^\nu, B_2^*, \widehat{B}_2 \in \mathbb{R}$ depended on $\Omega$ such that for any $(\boldsymbol{x}, t) \in \Omega \times [0, T]$, we have

$$|\partial_t \log \nu_t(\boldsymbol{x})| \leqslant B_0^\nu, \quad |\partial_t u_t^*(\boldsymbol{x})| \leqslant B_0^*, \quad |\partial_t u_\theta(\boldsymbol{x}, t)| \leqslant \widehat{B}_0,$$

$$\|\nabla_{\boldsymbol{x}} \log \nu_t(\boldsymbol{x})\|^2 \leqslant B_1^\nu, \quad \|\nabla_{\boldsymbol{x}} u_t^*(\boldsymbol{x})\|^2 \leqslant B_1^*, \quad \|\nabla_{\boldsymbol{x}} u_\theta(\boldsymbol{x}, t)\|^2 \leqslant \widehat{B}_1,$$

$$\Delta \log \nu_t(\boldsymbol{x}) \leqslant B_2^\nu, \quad \Delta u_t^*(\boldsymbol{x}) \geqslant B_2^*, \quad \Delta u_\theta(\boldsymbol{x}, t) \geqslant \widehat{B}_2,$$

In practice, using weights clipping strategy as in Arjovsky et al. (2017), we can control the regularity of neural network approximation $u_\theta(\boldsymbol{x}, t)$, thus bound the constants $\widehat{B}_0, \widehat{B}_1, \widehat{B}_2$.

We summarize our main results in the following theorem. The proof is deferred to Appendix A.3, which generalizes the framework in Deveney et al. (2023).

**Theorem 2.** *Suppose that Assumption 1, 2, and 3 hold. We further assume that $u_\theta(\boldsymbol{x}, 0) = u_0^*(\boldsymbol{x})^2$ for any $\boldsymbol{x} \in \Omega$. Then for any positive constant $\varepsilon > 0$, the following holds for any $0 \leqslant t \leqslant T$,*

$$\|e_t(\cdot)\|_{L^2(\Omega; \nu_t)}^2 \leqslant \varepsilon L_{\mathrm{PINN}}(t; C_1(\varepsilon)), \tag{16}$$

*Moreover, for any $0 \leqslant t \leqslant T$,*

$$m_1 \|\nabla_{\boldsymbol{x}} e_t(\cdot)\|_{L^2(\Omega; \nu_t)}^2 \leqslant \varepsilon \|r_t(\cdot)\|_{L^2(\Omega; \nu_t)}^2 + C_3(\varepsilon) L_{\mathrm{PINN}}(t; C_1(\varepsilon)) + C_2 \sqrt{\varepsilon L_{\mathrm{PINN}}(t; C_1(\varepsilon))}, \tag{17}$$

*where $C_2 := 2\sqrt{2}(\widehat{B}_0^2 + B_0^{*2})^{1/2}$, $C_3(\varepsilon) := \varepsilon(C_1(\varepsilon) + B_0^\nu)$, and $C_1(\varepsilon)$ is a constant depended on $B_1^\nu, B_1^*. \widehat{B}_1, B_2^\nu, B_2^*. \widehat{B}_2$ and $m_1, M_1$.*

**Remark 1.** *The results of Wang et al. (2022) show that the $L^2$-error cannot be universally bounded by the PINN residual with universal constants independent of the approximate solution. Therefore, some natural continuous assumption (Assumption 3) about the approximate solution are necessary to control the $L^2$-error by the PINN residual. It is noted that this continuous assumption can be satisfied by regularizing the neural network via weight clipping (Arjovsky et al., 2017), and would not sacrifice much approximation accuracy as the true solution is initialized as the log-density of the target and follows the diffusion process (e.g., the OU process) that would only become smoother as time evolves. Moreover, our upper bound of $L^2$-error depends on continuous constants rather than an universal bound. In this regard, our analysis aligns with the results of Wang et al. (2022), but with a more flexible bound based on some natural continuous assumption in the context of diffusion-based sampling.*

## 5.2 CONVERGENCE OF DIFFUSION-PINN SAMPLER

In this section, we present our convergence analysis of DPS based on Theorem 2 and the analysis of score-based generative modeling in Chen et al. (2023a). Following Chen et al. (2023b;a), we focus on the forward process with $\boldsymbol{f}(\boldsymbol{x}, t) = -\frac{1}{2}\boldsymbol{x}$ and $g(t) \equiv 1$, which is driven by

$$\mathrm{d}\boldsymbol{x}_t = -\frac{1}{2}\boldsymbol{x}_t \, \mathrm{d}t + \mathrm{d}\boldsymbol{B}_t, \quad \boldsymbol{x}_0 \sim \pi, \quad 0 \leqslant t \leqslant T, \tag{18}$$

In practice, we use a discrete-time approximation for the reverse process. Let $0 = t_0 < \cdots < t_N = T$ be the discretization points and $h_k := t_k - t_{k-1}$ be the step size for $1 \leqslant k \leqslant N$. Let $t_k' := T - t_{N-k}$ for $0 \leqslant k \leqslant N$ be the corresponding discretization points in the reverse SDE. In our analysis, we consider the exponential integrator scheme which leads to the following sampling dynamics for $0 \leqslant k \leqslant N - 1$,

$$\mathrm{d}\widehat{\boldsymbol{y}}_t = \left( \frac{1}{2} \widehat{\boldsymbol{y}}_t + \boldsymbol{s}_{T - t_k'}(\widehat{\boldsymbol{y}}_{t_k'}) \right) \mathrm{d}t + \mathrm{d}\boldsymbol{B}_t, \quad \widehat{\boldsymbol{y}}_0 \sim \mathcal{N}(\boldsymbol{0}, \boldsymbol{I}_d), \quad t \in [t_k', t_{k+1}'], \tag{19}$$

where $\boldsymbol{s}_t(\boldsymbol{x}) \approx \nabla_{\boldsymbol{x}} \log \pi_t(\boldsymbol{x})$ denotes the score approximation. Let $\hat{\pi}_T$ denote the distribution of $\hat{\boldsymbol{y}}_T$ from (19). We summarize all the assumptions we need as follows.

**Assumption 4.** *The target distribution admits a density $\pi \in \mathcal{C}^2(\mathbb{R}^d)$ where $\nabla_{\boldsymbol{x}} \log \pi(\boldsymbol{x})$ is $K$-Lipschitz and has the finite second moment, i.e., $M_2 := \mathbb{E}_\pi \left[ \|\boldsymbol{x}\|^2 \right] < \infty$.*

**Assumption 5.** *For any $\delta > 0$, there exists bounded $\Omega$ such that $\int_{\Omega^c} \pi_t(\boldsymbol{x}) \|\nabla_{\boldsymbol{x}} \log \pi_t(\boldsymbol{x})\|^2 \, \mathrm{d}\boldsymbol{x} \leqslant \delta$ for any $t \in [0, T]$.*

**Assumption 6.** *For any $(\boldsymbol{x}, t) \in \Omega \times [0, T]$, there exists $R_t \geqslant 0$ depended on $t$, so that $\frac{\pi_t(\boldsymbol{x})}{\nu_t(\boldsymbol{x})} \leqslant R_t$.*

Theorem 3 summarizes our main theoretical results of DPS. The proof can be found in Appendix A.4, which is based on the convergence results of score-based generative modeling in Chen et al. (2023a).

**Theorem 3.** *Suppose that $T \geqslant 1$, $K \geqslant 2$, and Assumptions 1-6 hold. For any $\delta > 0$, let $\Omega$ be chosen as in Assumption 5. For any positive constant $\varepsilon > 0$, we further assume that $u_\theta(\boldsymbol{x}, t)$ satisfies*

$$\varepsilon \sum_{k=1}^N h_k R_{t_k} \|r_{t_k}(\cdot)\|_{L^2(\Omega; \nu_{t_k})}^2 \leqslant \delta_1 \quad and \quad \varepsilon \sum_{k=1}^N h_k R_{t_k} L_{\mathrm{PINN}}(t_k; C_1(\varepsilon)) \leqslant \delta_2. \tag{20}$$

---

[2]This is a reasonable assumption due to the target-informed parameterization introduced in Section 4.2.

*Then there is a universal constant $\alpha \geqslant 2$ such that the following holds. Using step size $h_k = h \min\{\max\{t_k, 1/(4K)\}, 1\}$, $0 < h \leqslant 1/(\alpha d)$, and $\boldsymbol{s}_t(\boldsymbol{x}) = \nabla_{\boldsymbol{x}} u_\theta(\boldsymbol{x}, t) \cdot \mathbb{1}\{\boldsymbol{x} \in \Omega\}$ in (19), we have the following upper bound on the KL divergence between the target and the approximate distribution*

$$\mathrm{KL}(\pi \| \hat{\pi}_T) \lesssim (d + M_2) \cdot e^{-T} + d^2 h(\log K + T) + T\delta + \delta_1 + C_5(\varepsilon)\delta_2 + C_2 \sqrt{\sum_{k=1}^{N} h_k R_{t_k} \delta_2}. \quad (21)$$

*where $C_5(\varepsilon) := C_1(\varepsilon) + B_0^\nu$, $C_2$ and $C_1(\varepsilon)$ are defined in Theorem 2.*

### 5.3 THEORETICAL COMPARISON BETWEEN DIFFERENT SAMPLING METHODS FOR COLLOCATION POINTS

In practice, we typically lack prior knowledge of the high-probability regions of the diffusion path starting from the target distribution. As a result, specifying a sufficiently large support for uniform sampling of collocation points, becomes challenging and inefficient, especially in high-dimensional settings. In contrast, we employ a more sophisticated strategy for generating collocation points that integrates Langevin Monte Carlo (LMC) with the forward pass (see Section 4.2 for details). Similar to Theorem 2 and 3, theoretical guarantee of uniformly sampled collocation points can be established, albeit in a weaker form. Specifically, our results indicate that employing LMC and the forward pass for sampling collocation points is advantageous over uniform sampling. This is because, in the uniform case, the KL bound includes a factor proportional to the volume of the support, $\mathrm{Vol}(\Omega)$, which can be prohibitively large in high dimensions. In contrast, our bound depends on the density ratio $\pi_t/\nu_t$, which is more manageable due to LMC and converges to 1 as $t$ increases, thanks to the forward process. Detailed results and proofs for uniform collocation points are provided in Appendix B.

## 6 NUMERICAL EXPERIMENTS

In this section, we conduct experiments on various sampling tasks to demonstrate the effectiveness and efficiency of the Diffusion-PINN Sampler (DPS) compared to previous methods. Our sampling tasks includes 9-Gaussians ($d = 2$), Rings ($d = 2$), Funnel (Neal, 2003) ($d = 10$), and Double-well ($d = 30$), which are commonly used to evaluate diffusion-based sampling algorithms (Zhang & Chen, 2021; Berner et al., 2022; Grenioux et al., 2024). For multimodal distributions, the modes are designed to be well-separated, with challenging mixing proportions between different modes (see more details in Appendix E.2). For DPS, we employ a time-rescaled forward process and use a weight function $\beta(t) = 2(1 - t)$ for the PINN residual loss to improve numerical stability. To generate collocation points for each task, we run a short chain of LMC with a relatively large step size for better coverage

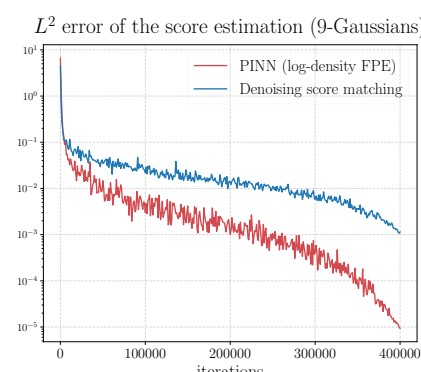

Figure 2: Comparison between solving log-density FPE by PINN and denoising score matching on score estimation.

of the high-density domain. For 9-Gaussians, Rings, and Double-well, the PINN residual loss alone suffices for good performance, so we set the regularization coefficient $\lambda = 0$. For Funnel, however, regularization proves helpful, and we set $\lambda = 1$ (details in Section 6.3). More details on experiment settings can be found in Appendix E.3.

**Baselines.** We benchmark DPS performance against a wide range of strong baseline methods. For MCMC methods, we consider the Langevin Monte Carlo (LMC) and Hamiltonian Monte Carlo (Neal, 2012) (HMC) . For particle-based VI methods, we include Stein Variational Gradient Descen (Liu & Wang, 2016) (SVGD). As for sampling methods using reverse diffusion, we include RDMC (Huang et al., 2023) and SLIPS (Grenioux et al., 2024). We also compare with the VI-based PIS (Zhang & Chen, 2021) and DIS (Berner et al., 2022), and their recent improved variants PIS-LV and DIS-LV proposed in Richter et al. (2023). See Appendix E.1 for more details.

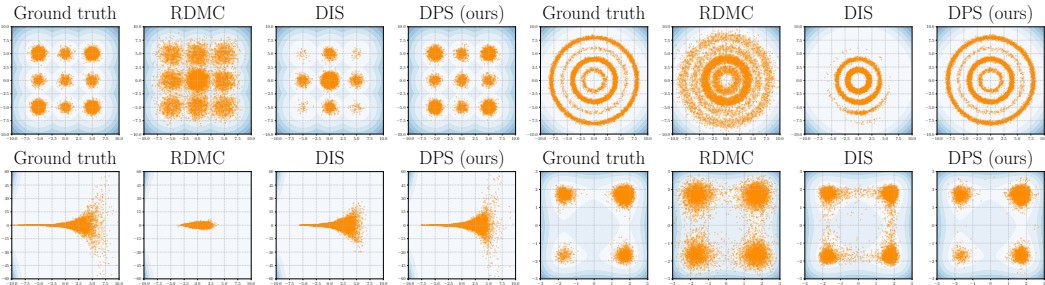

Figure 3: Sampling performance of different methods for 9-Gaussians ($d = 2$), Rings ($d = 2$), Funnel ($d = 10$), and Double-well ($d = 30$).

Table 1: KL divergence ($\downarrow$) to the ground truth obtained by different methods. Bold font indicates the best results. We use the KL divergence of the first two dimensions for Funnel ($d = 10$) and the KL divergence of the first five dimensions for Double-well ($d = 30$). All the KL divergence is computed by the ITE package (Szabó, 2014).

| Target | LMC | HMC | SVGD | RDMC | SLIPS | PIS | DIS | PIS-LV | DIS-LV | DPS (ours) |
|---|---|---|---|---|---|---|---|---|---|---|
| 9-Gaussians | $1.6568_{\pm 0.0189}$ | $1.8932_{\pm 0.0239}$ | $0.9712_{\pm 0.0153}$ | $1.0844_{\pm 0.0132}$ | $0.0901_{\pm 0.0071}$ | $2.0042_{\pm 0.0203}$ | $2.2758_{\pm 0.0240}$ | $2.1301_{\pm 0.0224}$ | $0.0682_{\pm 0.0081}$ | $\mathbf{0.0131_{\pm 0.0093}}$ |
| Rings | $2.4754_{\pm 0.0302}$ | $2.5894_{\pm 0.0170}$ | $0.1608_{\pm 0.0119}$ | $0.7487_{\pm 0.0073}$ | $0.4127_{\pm 0.0144}$ | $2.6985_{\pm 0.0290}$ | $2.3433_{\pm 0.0275}$ | $0.0124_{\pm 0.0204}$ | $0.0369_{\pm 0.0178}$ | $\mathbf{0.0176_{\pm 0.0059}}$ |
| Funnel | $0.1908_{\pm 0.0156}$ | $0.6137_{\pm 0.0141}$ | $0.1006_{\pm 0.0188}$ | $2.2050_{\pm 0.0364}$ | $0.1971_{\pm 0.0133}$ | $0.4377_{\pm 0.0199}$ | $0.2383_{\pm 0.0169}$ | $0.1521_{\pm 0.0230}$ | $\mathbf{0.0362_{\pm 0.0167}}$ | $0.0846_{\pm 0.0122}$ |
| Double-well | $0.1915_{\pm 0.0122}$ | $1.6729_{\pm 0.0303}$ | $1.3768_{\pm 0.0683}$ | $1.5735_{\pm 0.0162}$ | $0.4840_{\pm 0.0145}$ | $0.0969_{\pm 0.0114}$ | $0.6796_{\pm 0.0139}$ | $0.0478_{\pm 0.0280}$ | $0.0358_{\pm 0.0256}$ | $\mathbf{0.0273_{\pm 0.0113}}$ |

## 6.1 SCORE ESTIMATION

We first evaluate the accuracy of score function estimates obtained by solving the log-density FPE (Algorithm 1). To do that, we conduct an experiment on the 9-Gaussians target $\pi$ where we know the ground truth scores throughout the entire forward process. Figure 2 shows the $L^2(\pi)$ error of the score estimation for our method compared to denoising score matching (Vincent, 2011; Song et al., 2020a). We see clearly that our method provides more accurate score estimation than denoising score matching.

Table 2: $L^2$ error ($\downarrow$) of the mixing proportions estimation when sampling multimodal target distributions using different methods. Bold font indicates the best results. All the estimation is computed with 1,000 samples.

| Target | LMC | HMC | SVGD | RDMC | SLIPS | PIS | DIS | PIS-LV | DIS-LV | DPS (ours) |
|---|---|---|---|---|---|---|---|---|---|---|
| 9-Gaussians | $0.5199_{\pm 0.0159}$ | $0.8007_{\pm 0.2231}$ | $0.2098_{\pm 0.0097}$ | $0.1313_{\pm 0.0099}$ | $0.0018_{\pm 0.0005}$ | $0.4893_{\pm 0.0110}$ | $0.7268_{\pm 0.0146}$ | $0.4217_{\pm 0.0009}$ | $0.0013_{\pm 0.0004}$ | $\mathbf{0.0006_{\pm 0.0003}}$ |
| Rings | $0.6005_{\pm 0.0251}$ | $0.7954_{\pm 0.3622}$ | $0.1767_{\pm 0.0355}$ | $0.0537_{\pm 0.0035}$ | $0.2471_{\pm 0.0144}$ | $0.8016_{\pm 0.0194}$ | $0.5233_{\pm 0.0194}$ | $0.0010_{\pm 0.0013}$ | $\mathbf{0.0007_{\pm 0.0004}}$ | $\mathbf{0.0006_{\pm 0.0006}}$ |
| Double-well | $0.0673_{\pm 0.0082}$ | $0.9773_{\pm 0.4020}$ | $0.2400_{\pm 0.0174}$ | $0.2154_{\pm 0.0075}$ | $0.1645_{\pm 0.0113}$ | $0.0044_{\pm 0.0011}$ | $0.0684_{\pm 0.0035}$ | $\mathbf{0.0005_{\pm 0.0004}}$ | $0.0012_{\pm 0.0009}$ | $\mathbf{0.0004_{\pm 0.0002}}$ |

## 6.2 SAMPLE QUALITY

In this section, we compare DPS with the aforementioned baseline methods on various target distributions. We use KL divergence to evaluate the quality of samples provided by different methods in low dimensional problems (9-Gaussians, Rings), and use the projected KL divergence instead for Funnel and Double-well that are problems with relatively higher dimensions. The results are reported in Table 1. Figure 3 visualizes the samples from different methods. We clearly see that DPS provides the best approximation accuracy and sample quality among all methods. Although we use LMC to generate collocation points, DPS greatly outperforms LMC, indicating the power of diffusion-based sampling methods with learned score functions.

For multimodal distributions, we estimate the mixing proportions for different modes using samples generated by different methods, and evaluate the estimation accuracy in terms of $L^2$ error to the true weights. The results are shown in Table 2. It is clear that DPS provides accurate weights estimation while other baselines tend to struggle to learn the weights.

## 6.3 ABLATION STUDY

In this section, we compare the performance of score estimation between solving the score FPE and the log-density FPE, and investigate the effect of regularization in DPS.

We first solve the corresponding score FPE and log-density FPE for a MoG with two distant modes: $\pi^M = 0.2\mathcal{N}((-5, -5)', \boldsymbol{I}_2) + 0.8\mathcal{N}((5, 5)', \boldsymbol{I}_2)$. The left plot in Figure 4 show the PINN residual loss and the score estimation error as functions of the number of iterations. We see that for the score FPE, the score approximation error decreases rapidly at first but quickly levels off, while

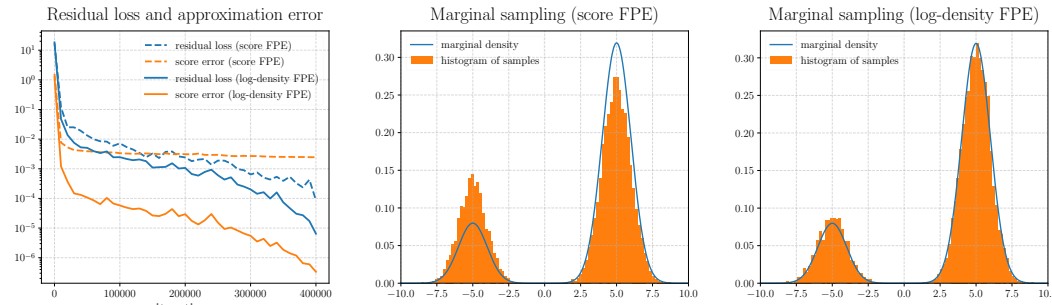

Figure 4: **Left**: PINN residual loss and score approximation error during solving score/log-density FPE; **Middle/Right**: Marginals of the first dimension from DPS by solving score/log-density FPE for MoG with two modes.

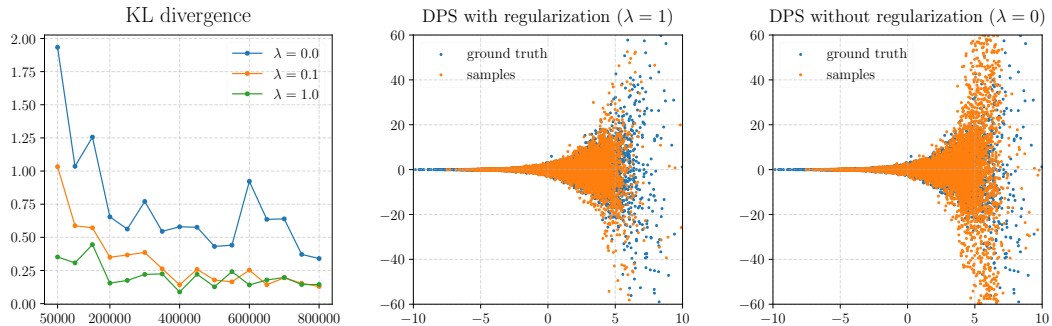

Figure 5: **Left**: KL divergence to the ground truth during solving log-density FPE with different regularization for Funnel. **Middle/Right**: Sampling performance of DPS with/without regularization for Funnel.

the PINN residual loss continues to decrease with more iterations. In contrast, when solving the log-density FPE, the PINN residual loss and the score approximation error decrease consistently, resulting in more accurate score approximation overall. The middle and right plots in Figure 4 display the histogram based on samples generated from the reverse SDE using the score estimates from both methods, together with the true marginal density. We observe that the score FPE-based method fails to identify the correct mixing proportions, whereas the log-density FPE-based method successfully recovers the correct weights.

Next, we solve the log-density PFE with different regularization coefficients $\lambda$ on the Funnel target. Figure 5 (left) shows the KL divergence for various $\lambda$ as a function of the number of iterations. We see that, compared to the non-regularized case ($\lambda = 0$), both the convergence speed and overall approximation accuracy have been greatly improved when regularization is applied. The middle and right plots in Figure 5 show the samples generated from DPS with $\lambda = 1$ and $\lambda = 0$ respectively. With regularization, DPS provides a better fit to the target distribution, more accurately capturing the thickness in the tails. This indicates that regularization could be beneficial for heavy-tail distributions.

## 7 CONCLUSION

In this work, we proposed *Diffusion-PINN Sampler* (DPS), a novel method that leverages Physics-Informed Neural Networks (PINN) and diffusion models for accurate sampling from complex target distributions. By solving the log-density FPE that governs the evolution of the log-density of the underlying SDE marginals via PINN, DPS demonstrates accurate sampling capabilities even for distributions with multiple modes or heavy tails, and it excels in identifying mixing proportions when the target features isolated modes. The control of log-density estimation error via PINN residual loss ensures convergence guarantees to the target distribution, building upon established results for score-based diffusion models. We demonstrated the effectiveness of our approach on multiple numerical examples. Limitations are discussed in Appendix C.

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

## A PROOFS

### A.1 PROOF OF THEOREM 1

*Proof of Theorem 1.* Recall that $p_t(\boldsymbol{x})$ denotes the marginal density of $\boldsymbol{x}_t$ following the forward process (1), and satisfies

$$\partial_t p_t(\boldsymbol{x}) = \frac{1}{2} g^2(t) \Delta p_t(\boldsymbol{x}) - \nabla \cdot [\boldsymbol{f}(\boldsymbol{x}, t) p_t(\boldsymbol{x})]. \tag{22}$$

Therefore, the log-density $u_t(\boldsymbol{x}) := \log p_t(\boldsymbol{x})$ satisfies

$$\partial_t u_t(\boldsymbol{x}) = \frac{\partial_t p_t(\boldsymbol{x})}{p_t(\boldsymbol{x})} = \frac{1}{2} g^2(t) \frac{\Delta p_t(\boldsymbol{x})}{p_t(\boldsymbol{x})} - \frac{\nabla \cdot [\boldsymbol{f}(\boldsymbol{x}, t) p_t(\boldsymbol{x})]}{p_t(\boldsymbol{x})}. \tag{23}$$

Note that we have the identities

$$\Delta p_t(\boldsymbol{x}) = \nabla \cdot [p_t(\boldsymbol{x}) \nabla_{\boldsymbol{x}} u_t(\boldsymbol{x})] = \nabla_{\boldsymbol{x}} p_t(\boldsymbol{x}) \cdot \nabla_{\boldsymbol{x}} u_t(\boldsymbol{x}) + p_t(\boldsymbol{x}) \Delta u_t(\boldsymbol{x}),$$
$$\nabla \cdot [\boldsymbol{f}(\boldsymbol{x}, t) p_t(\boldsymbol{x})] = \nabla_{\boldsymbol{x}} p_t(\boldsymbol{x}) \cdot \boldsymbol{f}(\boldsymbol{x}, t) + p_t(\boldsymbol{x}) [\nabla \cdot \boldsymbol{f}(\boldsymbol{x}, t)]. \tag{24}$$

Plugging (24) into (23), we have

$$\partial_t u_t(\boldsymbol{x}) = \frac{1}{2} g^2(t) \Delta u_t(\boldsymbol{x}) + \frac{1}{2} g^2(t) \|\nabla_{\boldsymbol{x}} u_t(\boldsymbol{x})\|^2 - \boldsymbol{f}(\boldsymbol{x}, t) \cdot \nabla_{\boldsymbol{x}} u_t(\boldsymbol{x}) - \nabla \cdot \boldsymbol{f}(\boldsymbol{x}, t).$$

Since $\log p_t(\boldsymbol{x})$ is sufficiently smooth, we can swap the order of differentiations and get

$$\partial_t \boldsymbol{s}_t(\boldsymbol{x}) = \partial_t \nabla_{\boldsymbol{x}} u_t(\boldsymbol{x}) = \nabla_{\boldsymbol{x}} \partial_t u_t(\boldsymbol{x}).$$

Hence, the theorem is proved. $\qquad \square$

### A.2 OMITTED PROOF IN EXAMPLE 1

**Notations.** For two probability measures $\nu_1$ and $\nu_2$ in $\mathbb{R}^d$, we define the $L^2(p)$ error of their scores as $\mathrm{SE}_p(\nu_1 \| \nu_2) := \mathbb{E}_{\boldsymbol{x} \sim p}[\|\nabla_{\boldsymbol{x}} \log \nu_1(\boldsymbol{x}) - \nabla_{\boldsymbol{x}} \log \nu_2(\boldsymbol{x})\|^2]$ where $p$ also denotes a probability measure. Note that if we choose $p = \nu_1$, we have $\mathrm{SE}_{\nu_1}(\nu_1 \| \nu_2) = F(\nu_1, \nu_2)$ where $F(\nu_1, \nu_2)$ denotes the Fisher divergence between $\nu_1$ and $\nu_2$. For any $\boldsymbol{a} \in \mathbb{R}^d$, we denote $\gamma_{\boldsymbol{a}}(\boldsymbol{x}) := \exp(-\|\boldsymbol{x} - \boldsymbol{a}\|^2 / 2)$. For simplify, we denote $\mathbb{E}_{\boldsymbol{x} \sim \mathcal{N}(\boldsymbol{a}, I_d)}[\cdot]$ by $\mathbb{E}_{\gamma_{\boldsymbol{a}}}[\cdot]$. Thus the probability density of $\mathcal{N}(\boldsymbol{a}, I_d)$ is $p(\boldsymbol{x}) = \gamma_{\boldsymbol{a}}(\boldsymbol{x}) / (\sqrt{2\pi})^d$. For the MoG $\pi^M = w_1 \mathcal{N}(\boldsymbol{a}_1, I_d) + w_2 \mathcal{N}(\boldsymbol{a}_2, I_d)$, the score is given by

$$\nabla_{\boldsymbol{x}} \log \pi^M(\boldsymbol{x}) = \frac{w_1 \boldsymbol{a}_1 \gamma_{\boldsymbol{a}_1}(\boldsymbol{x}) + w_2 \boldsymbol{a}_2 \gamma_{\boldsymbol{a}_2}(\boldsymbol{x})}{w_1 \gamma_{\boldsymbol{a}_1}(\boldsymbol{x}) + w_2 \gamma_{\boldsymbol{a}_2}(\boldsymbol{x})} - \boldsymbol{x}. \tag{25}$$

Then we show our general results in Theorem 4 where we state a lower bound of $\mathrm{KL}(\pi^M \| \hat{\pi}^M)$ and an upper bound of $\mathrm{SE}_p(\pi^M \| \hat{\pi}^M)$.

**Theorem 4.** *Consider two MoGs in $\mathbb{R}^d$: $\pi^M = w_1 \mathcal{N}(\boldsymbol{a}_1, I_d) + w_2 \mathcal{N}(\boldsymbol{a}_2, I_d)$, $\hat{\pi}^M = \hat{w}_1 \mathcal{N}(\boldsymbol{a}_1, I_d) + \hat{w}_2 \mathcal{N}(\boldsymbol{a}_2, I_d)$ where $\boldsymbol{a}_1, \boldsymbol{a}_2 \in \mathbb{R}^d$, $w_1, w_2, \hat{w}_1, \hat{w}_2 > 0$, $w_1 + w_2 = 1$ and $\hat{w}_1 + \hat{w}_2 = 1$. Then* $\mathrm{KL}(\pi^M \| \hat{\pi}^M)$ *is lower bounded by*

$$\mathrm{KL}(\pi^M \| \hat{\pi}^M) \geqslant w_1 \left( \log w_1 - \log \left( \hat{w}_1 + \exp \left( -\frac{\|\boldsymbol{a}_1 - \boldsymbol{a}_2\|^2}{4} \right) \right) \right)$$

$$+ w_2 \left( \log w_2 - \log \left( \hat{w}_2 + \exp \left( -\frac{\|\boldsymbol{a}_1 - \boldsymbol{a}_2\|^2}{4} \right) \right) \right) \tag{26}$$

$$- (\log 4 + d) \exp \left( \frac{d}{2} \log 2 - \frac{\|\boldsymbol{a}_1 - \boldsymbol{a}_2\|^2}{64} \right).$$

*Let $p(\boldsymbol{x})$ denote any distribution that is absolutely continuous w.r.t. $\mu$, then $\mathrm{SE}_p(\pi^M \| \hat{\pi}^M)$ is upper bounded by*

$$\mathrm{SE}_p(\pi^M \| \hat{\pi}^M) \leqslant 2 \exp \left( -\frac{\|\boldsymbol{a}_1 - \boldsymbol{a}_2\|^2}{2} \right) \left[ \frac{w_2^2}{w_1^2} + \frac{\hat{w}_2^2}{\hat{w}_1^2} + \frac{w_1^2}{w_2^2} + \frac{\hat{w}_1^2}{\hat{w}_2^2} \right] \|\boldsymbol{a}_1 - \boldsymbol{a}_2\|^2$$

$$+ 8 \left[ \|\boldsymbol{a}_1\|^2 + \|\boldsymbol{a}_2\|^2 \right] \int_{\Omega_3} p(\boldsymbol{x}) \, \mathrm{d}\boldsymbol{x}, \tag{27}$$

where $\Omega_1 = \left\{ \boldsymbol{x} \in \mathbb{R}^d : \|\boldsymbol{x} - \boldsymbol{a}_1\| \leqslant \frac{\|\boldsymbol{a}_1 - \boldsymbol{a}_2\|}{4} \right\}$, $\Omega_2 = \left\{ \boldsymbol{x} \in \mathbb{R}^d : \|\boldsymbol{x} - \boldsymbol{a}_2\| \leqslant \frac{\|\boldsymbol{a}_1 - \boldsymbol{a}_2\|}{4} \right\}$, and $\Omega_3 = \Omega_1^c \bigcap \Omega_2^c$.

**Remark 2.** *If we choose $p(\boldsymbol{x}) = \pi^M(\boldsymbol{x})$ in Theorem 4, the Fisher divergence $F(\pi^M, \hat{\pi}^M)$ is upper bounded by*

$$
\begin{aligned}
F(\pi^M, \hat{\pi}^M) \leqslant{} & 2 \exp\left( -\frac{\|\boldsymbol{a}_1 - \boldsymbol{a}_2\|^2}{2} \right) \left[ \frac{w_2^2}{w_1^2} + \frac{\hat{w}_2^2}{\hat{w}_1^2} + \frac{w_1^2}{w_2^2} + \frac{\hat{w}_1^2}{\hat{w}_2^2} \right] \|\boldsymbol{a}_1 - \boldsymbol{a}_2\|^2 \\
& + 8 \left[ \|\boldsymbol{a}_1\|^2 + \|\boldsymbol{a}_2\|^2 \right] \exp\left( \frac{d}{2} \log 2 - \frac{\|\boldsymbol{a}_1 - \boldsymbol{a}_2\|^2}{64} \right),
\end{aligned} \tag{28}
$$

*where we use the following inequality*

$$
\begin{aligned}
\int_{\Omega_3} \pi^M(\boldsymbol{x}) \, \mathrm{d}\boldsymbol{x} ={} & w_1 \int_{\Omega_3} \frac{1}{(\sqrt{2\pi})^d} \gamma_{\boldsymbol{a}_1}(\boldsymbol{x}) \, \mathrm{d}\boldsymbol{x} + w_2 \int_{\Omega_3} \frac{1}{(\sqrt{2\pi})^d} \gamma_{\boldsymbol{a}_2}(\boldsymbol{x}) \, \mathrm{d}\boldsymbol{x} \\
\leqslant{} & w_1 \exp\left( \frac{d}{2} \log 2 - \frac{\|\boldsymbol{a}_1 - \boldsymbol{a}_2\|^2}{64} \right) + w_2 \exp\left( \frac{d}{2} \log 2 - \frac{\|\boldsymbol{a}_1 - \boldsymbol{a}_2\|^2}{64} \right) \\
={} & \exp\left( \frac{d}{2} \log 2 - \frac{\|\boldsymbol{a}_1 - \boldsymbol{a}_2\|^2}{64} \right).
\end{aligned}
$$

*Thus Example 1 holds naturally.*

*Proof of Theorem 4.* We first prove (26). We can decompose $\mathrm{KL}\left( \pi^M \| \hat{\pi}^M \right)$ as

$$
\begin{aligned}
\mathrm{KL}\left( \pi^M \| \hat{\pi}^M \right) ={} & \mathbb{E}_{\pi^M} \left[ \log\left( \frac{\pi^M(\boldsymbol{x})}{\hat{\pi}^M(\boldsymbol{x})} \right) \right] \\
={} & w_1 \mathbb{E}_{\gamma_{\boldsymbol{a}_1}} \left[ \log\left( \frac{w_1 \gamma_{\boldsymbol{a}_1}(\boldsymbol{x}) + w_2 \gamma_{\boldsymbol{a}_2}(\boldsymbol{x})}{\hat{w}_1 \gamma_{\boldsymbol{a}_1}(\boldsymbol{x}) + \hat{w}_2 \gamma_{\boldsymbol{a}_2}(\boldsymbol{x})} \right) \right] + w_2 \mathbb{E}_{\gamma_{\boldsymbol{a}_2}} \left[ \log\left( \frac{w_1 \gamma_{\boldsymbol{a}_1}(\boldsymbol{x}) + w_2 \gamma_{\boldsymbol{a}_2}(\boldsymbol{x})}{\hat{w}_1 \gamma_{\boldsymbol{a}_1}(\boldsymbol{x}) + \hat{w}_2 \gamma_{\boldsymbol{a}_2}(\boldsymbol{x})} \right) \right].
\end{aligned} \tag{29}
$$

Note that

$$
\begin{aligned}
\mathbb{E}_{\gamma_{\boldsymbol{a}_1}} \left[ \log\left( \frac{w_1 \gamma_{\boldsymbol{a}_1}(\boldsymbol{x}) + w_2 \gamma_{\boldsymbol{a}_2}(\boldsymbol{x})}{\hat{w}_1 \gamma_{\boldsymbol{a}_1}(\boldsymbol{x}) + \hat{w}_2 \gamma_{\boldsymbol{a}_2}(\boldsymbol{x})} \right) \right] ={} & \mathbb{E}_{\gamma_{\boldsymbol{a}_1}} \left[ \log\left( \frac{w_1 + w_2 \gamma_{\boldsymbol{a}_2}(\boldsymbol{x}) / \gamma_{\boldsymbol{a}_1}(\boldsymbol{x})}{\hat{w}_1 + \hat{w}_2 \gamma_{\boldsymbol{a}_2}(\boldsymbol{x}) / \gamma_{\boldsymbol{a}_1}(\boldsymbol{x})} \right) \right] \\
\geqslant{} & \log w_1 - \mathbb{E}_{\gamma_{\boldsymbol{a}_1}} \left[ \log\left( \hat{w}_1 + \hat{w}_2 \frac{\gamma_{\boldsymbol{a}_2}(\boldsymbol{x})}{\gamma_{\boldsymbol{a}_1}(\boldsymbol{x})} \right) \right].
\end{aligned} \tag{30}
$$

Let $\widetilde{\Omega}_1 = \{ \boldsymbol{x} \in \mathbb{R}^d : \|\boldsymbol{x} - \boldsymbol{a}_1\| \leqslant \frac{\|\boldsymbol{a}_1 - \boldsymbol{a}_2\|}{4} \}$, $\widetilde{\Omega}_2 = \widetilde{\Omega}_1^c \bigcap \{ \boldsymbol{x} \in \mathbb{R}^d : \hat{w}_2 \gamma_{\boldsymbol{a}_2}(\boldsymbol{x}) / \gamma_{\boldsymbol{a}_1}(\boldsymbol{x}) \leqslant \hat{w}_1 \}$, and $\widetilde{\Omega}_3 = (\widetilde{\Omega}_1 \bigcup \widetilde{\Omega}_2)^c = \widetilde{\Omega}_1^c \bigcap \widetilde{\Omega}_2^c$. Then for any $\boldsymbol{x} \in \widetilde{\Omega}_1$, we have $\|\boldsymbol{x} - \boldsymbol{a}_2\| \geqslant \|\boldsymbol{a}_1 - \boldsymbol{a}_2\| - \|\boldsymbol{x} - \boldsymbol{a}_1\| \geqslant \frac{3}{4} \|\boldsymbol{a}_1 - \boldsymbol{a}_2\|$, thus $\gamma_{\boldsymbol{a}_2}(\boldsymbol{x}) / \gamma_{\boldsymbol{a}_1}(\boldsymbol{x}) = \exp\left( \frac{\|\boldsymbol{x} - \boldsymbol{a}_1\|^2 - \|\boldsymbol{x} - \boldsymbol{a}_2\|^2}{2} \right) \leqslant \exp\left( -\frac{\|\boldsymbol{a}_1 - \boldsymbol{a}_2\|^2}{4} \right)$. Then we have

$$
\begin{aligned}
& \int_{\widetilde{\Omega}_1} \frac{1}{(\sqrt{2\pi})^d} \gamma_{\boldsymbol{a}_1}(\boldsymbol{x}) \log\left( \hat{w}_1 + \hat{w}_2 \frac{\gamma_{\boldsymbol{a}_2}(\boldsymbol{x})}{\gamma_{\boldsymbol{a}_1}(\boldsymbol{x})} \right) \, \mathrm{d}\boldsymbol{x} \\
& \leqslant \log\left( \hat{w}_1 + \hat{w}_2 \exp\left( -\frac{\|\boldsymbol{a}_1 - \boldsymbol{a}_2\|^2}{4} \right) \right) \leqslant \log\left( \hat{w}_1 + \exp\left( -\frac{\|\boldsymbol{a}_1 - \boldsymbol{a}_2\|^2}{4} \right) \right).
\end{aligned} \tag{31}
$$

Note that

$$
\begin{aligned}
\int_{\widetilde{\Omega}_1^c} \frac{1}{(\sqrt{2\pi})^d} \gamma_{\boldsymbol{a}_1}(\boldsymbol{x}) \, \mathrm{d}\boldsymbol{x} \leqslant{} & \int_{\widetilde{\Omega}_1^c} \exp\left( -\frac{\|\boldsymbol{a}_1 - \boldsymbol{a}_2\|^2}{64} \right) \cdot \frac{1}{(\sqrt{2\pi})^d} \exp\left( -\frac{\|\boldsymbol{x} - \boldsymbol{a}_1\|^2}{4} \right) \, \mathrm{d}\boldsymbol{x} \\
\leqslant{} & \exp\left( \frac{d}{2} \log 2 - \frac{\|\boldsymbol{a}_1 - \boldsymbol{a}_2\|^2}{64} \right),
\end{aligned} \tag{32}
$$

and

$$\int_{\widetilde{\Omega}_1^c} \frac{1}{\left(\sqrt{2\pi}\right)^d} \gamma_{\boldsymbol{a}_1}(\boldsymbol{x}) \frac{\|\boldsymbol{x}-\boldsymbol{a}_1\|^2}{2} \, \mathrm{d}\boldsymbol{x}$$

$$\leqslant \int_{\widetilde{\Omega}_1^c} \exp\left(-\frac{\|\boldsymbol{a}_1-\boldsymbol{a}_2\|^2}{64}\right) \cdot \frac{1}{\left(\sqrt{2\pi}\right)^d} \exp\left(-\frac{\|\boldsymbol{x}-\boldsymbol{a}_1\|^2}{4}\right) \frac{\|\boldsymbol{x}-\boldsymbol{a}_1\|^2}{2} \, \mathrm{d}\boldsymbol{x}$$

$$\leqslant \exp\left(\frac{d}{2}\log 2 - \frac{\|\boldsymbol{a}_1-\boldsymbol{a}_2\|^2}{64}\right) \cdot \mathbb{E}_{\boldsymbol{x}\sim\mathcal{N}(\boldsymbol{a}_1,2I_d)}\left[\frac{\|\boldsymbol{x}-\boldsymbol{a}_1\|^2}{2}\right]$$

$$= \exp\left(\log d + \frac{d}{2}\log 2 - \frac{\|\boldsymbol{a}_1-\boldsymbol{a}_2\|^2}{64}\right). \tag{33}$$

For every $\boldsymbol{x} \in \widetilde{\Omega}_2$, we have $\log\left(\hat{w}_1 + \hat{w}_2 \gamma_{\boldsymbol{a}_2}(\boldsymbol{x})/\gamma_{\boldsymbol{a}_1}(\boldsymbol{x})\right) \leqslant \log 2$. Using (32) and (33),

$$\int_{\widetilde{\Omega}_2} \frac{1}{\left(\sqrt{2\pi}\right)^d} \gamma_{\boldsymbol{a}_1}(\boldsymbol{x}) \log\left(\hat{w}_1 + \hat{w}_2 \frac{\gamma_{\boldsymbol{a}_2}(\boldsymbol{x})}{\gamma_{\boldsymbol{a}_1}(\boldsymbol{x})}\right) \, \mathrm{d}\boldsymbol{x} \leqslant \log 2 \cdot \exp\left(\frac{d}{2}\log 2 - \frac{\|\boldsymbol{a}_1-\boldsymbol{a}_2\|^2}{64}\right). \tag{34}$$

Similarly, for any $\boldsymbol{x} \in \widetilde{\Omega}_3$, we have $\log\left(\hat{w}_1 + \hat{w}_2 \gamma_{\boldsymbol{a}_2}(\boldsymbol{x})/\gamma_{\boldsymbol{a}_1}(\boldsymbol{x})\right) \leqslant \log 2 + \frac{\|\boldsymbol{x}-\boldsymbol{a}_1\|^2}{2}$. Thus

$$\int_{\widetilde{\Omega}_3} \frac{1}{\left(\sqrt{2\pi}\right)^d} \gamma_{\boldsymbol{a}_1}(\boldsymbol{x}) \log\left(\hat{w}_1 + \hat{w}_2 \frac{\gamma_{\boldsymbol{a}_2}(\boldsymbol{x})}{\gamma_{\boldsymbol{a}_1}(\boldsymbol{x})}\right) \, \mathrm{d}\boldsymbol{x} \leqslant (\log 2 + d) \exp\left(\frac{d}{2}\log 2 - \frac{\|\boldsymbol{a}_1-\boldsymbol{a}_2\|^2}{64}\right). \tag{35}$$

Putting (31), (34), and (35) together, $\mathbb{E}_{\gamma_{\boldsymbol{a}_1}}\left[\log\left(\hat{w}_1 + \hat{w}_2 \gamma_{\boldsymbol{a}_2}(\boldsymbol{x})/\gamma_{\boldsymbol{a}_1}(\boldsymbol{x})\right)\right]$ is upper bounded by

$$\mathbb{E}_{\gamma_{\boldsymbol{a}_1}}\left[\log\left(\hat{w}_1 + \hat{w}_2 \frac{\gamma_{\boldsymbol{a}_2}(\boldsymbol{x})}{\gamma_{\boldsymbol{a}_1}(\boldsymbol{x})}\right)\right]$$

$$= \left(\int_{\widetilde{\Omega}_1} + \int_{\widetilde{\Omega}_2} + \int_{\widetilde{\Omega}_3}\right) \frac{1}{\left(\sqrt{2\pi}\right)^d} \gamma_{\boldsymbol{a}_1}(\boldsymbol{x}) \log\left(\hat{w}_1 + \hat{w}_2 \frac{\gamma_{\boldsymbol{a}_2}(\boldsymbol{x})}{\gamma_{\boldsymbol{a}_1}(\boldsymbol{x})}\right) \, \mathrm{d}\boldsymbol{x} \tag{36}$$

$$\leqslant \log\left(\hat{w}_1 + \exp\left(-\frac{\|\boldsymbol{a}_1-\boldsymbol{a}_2\|^2}{4}\right)\right) + (\log 4 + d) \exp\left(\frac{d}{2}\log 2 - \frac{\|\boldsymbol{a}_1-\boldsymbol{a}_2\|^2}{64}\right).$$

Plugging (36) into (30), we have

$$w_1 \mathbb{E}_{\gamma_{\boldsymbol{a}_1}}\left[\log\left(\frac{w_1\gamma_{\boldsymbol{a}_1}(\boldsymbol{x}) + w_2\gamma_{\boldsymbol{a}_2}(\boldsymbol{x})}{\hat{w}_1\gamma_{\boldsymbol{a}_1}(\boldsymbol{x}) + \hat{w}_2\gamma_{\boldsymbol{a}_2}(\boldsymbol{x})}\right)\right]$$

$$\geqslant w_1\left[\log w_1 - \log\left(\hat{w}_1 + \exp\left(-\frac{\|\boldsymbol{a}_1-\boldsymbol{a}_2\|^2}{4}\right)\right)\right] \tag{37}$$

$$- w_1(\log 4 + d)\exp\left(\frac{d}{2}\log 2 - \frac{\|\boldsymbol{a}_1-\boldsymbol{a}_2\|^2}{64}\right).$$

Similarly, we have

$$w_2 \mathbb{E}_{\gamma_{\boldsymbol{a}_2}}\left[\log\left(\frac{w_1\gamma_{\boldsymbol{a}_1}(\boldsymbol{x}) + w_2\gamma_{\boldsymbol{a}_2}(\boldsymbol{x})}{\hat{w}_1\gamma_{\boldsymbol{a}_1}(\boldsymbol{x}) + \hat{w}_2\gamma_{\boldsymbol{a}_2}(\boldsymbol{x})}\right)\right]$$

$$\geqslant w_2\left(\log w_2 - \log\left(\hat{w}_2 + \exp\left(-\frac{\|\boldsymbol{a}_1-\boldsymbol{a}_2\|^2}{4}\right)\right)\right) \tag{38}$$

$$- w_2(\log 4 + d)\exp\left(\frac{d}{2}\log 2 - \frac{\|\boldsymbol{a}_1-\boldsymbol{a}_2\|^2}{64}\right).$$

Plugging (37) and (38) into (29), we obtain the lower bound (26) in Theorem 4. Then we prove (27). Using (25), we obtain

$$
\begin{aligned}
&\nabla_{\boldsymbol{x}} \log \pi^M(\boldsymbol{x}) - \nabla_{\boldsymbol{x}} \log \hat{\pi}^M(\boldsymbol{x}) \\
&= \frac{w_1 \boldsymbol{a}_1 \gamma_{\boldsymbol{a}_1}(\boldsymbol{x}) + w_2 \boldsymbol{a}_2 \gamma_{\boldsymbol{a}_2}(\boldsymbol{x})}{w_1 \gamma_{\boldsymbol{a}_1}(\boldsymbol{x}) + w_2 \gamma_{\boldsymbol{a}_2}(\boldsymbol{x})} - \frac{\hat{w}_1 \boldsymbol{a}_1 \gamma_{\boldsymbol{a}_1}(\boldsymbol{x}) + \hat{w}_2 \boldsymbol{a}_2 \gamma_{\boldsymbol{a}_2}(\boldsymbol{x})}{\hat{w}_1 \gamma_{\boldsymbol{a}_1}(\boldsymbol{x}) + \hat{w}_2 \gamma_{\boldsymbol{a}_2}(\boldsymbol{x})} \\
&= \frac{w_1 \boldsymbol{a}_1 + w_2 \boldsymbol{a}_2 \gamma_{\boldsymbol{a}_2}(\boldsymbol{x})/\gamma_{\boldsymbol{a}_1}(\boldsymbol{x})}{w_1 + w_2 \gamma_{\boldsymbol{a}_2}(\boldsymbol{x})/\gamma_{\boldsymbol{a}_1}(\boldsymbol{x})} - \frac{\hat{w}_1 \boldsymbol{a}_1 + \hat{w}_2 \boldsymbol{a}_2 \gamma_{\boldsymbol{a}_2}(\boldsymbol{x})/\gamma_{\boldsymbol{a}_1}(\boldsymbol{x})}{\hat{w}_1 + \hat{w}_2 \gamma_{\boldsymbol{a}_2}(\boldsymbol{x})/\gamma_{\boldsymbol{a}_1}(\boldsymbol{x})} \\
&= \frac{w_1 \boldsymbol{a}_1 \gamma_{\boldsymbol{a}_1}(\boldsymbol{x})/\gamma_{\boldsymbol{a}_2}(\boldsymbol{x}) + w_2 \boldsymbol{a}_2}{w_1 \gamma_{\boldsymbol{a}_1}(\boldsymbol{x})/\gamma_{\boldsymbol{a}_2}(\boldsymbol{x}) + w_2} - \frac{\hat{w}_1 \boldsymbol{a}_1 \gamma_{\boldsymbol{a}_1}(\boldsymbol{x})/\gamma_{\boldsymbol{a}_2}(\boldsymbol{x}) + \hat{w}_2 \boldsymbol{a}_2}{\hat{w}_1 \gamma_{\boldsymbol{a}_1}(\boldsymbol{x})/\gamma_{\boldsymbol{a}_2}(\boldsymbol{x}) + \hat{w}_2}.
\end{aligned}
\tag{39}
$$

Recall that $\Omega_1 = \{\boldsymbol{x} \in \mathbb{R}^d : \|\boldsymbol{x} - \boldsymbol{a}_1\| \leqslant \frac{\|\boldsymbol{a}_1 - \boldsymbol{a}_2\|}{4}\}$, $\Omega_2 = \{\boldsymbol{x} \in \mathbb{R}^d : \|\boldsymbol{x} - \boldsymbol{a}_2\| \leqslant \frac{\|\boldsymbol{a}_1 - \boldsymbol{a}_2\|}{4}\}$ and $\Omega_3 = \Omega_1^c \bigcap \Omega_2^c$. For any $\boldsymbol{x} \in \Omega_1$, we can rewrite (39) as

$$
\begin{aligned}
&\nabla_{\boldsymbol{x}} \log \pi^M(\boldsymbol{x}) - \nabla_{\boldsymbol{x}} \log \hat{\pi}^M(\boldsymbol{x}) \\
&= \boldsymbol{a}_1 + \frac{w_2(\boldsymbol{a}_2 - \boldsymbol{a}_1)\gamma_{\boldsymbol{a}_2}(\boldsymbol{x})/\gamma_{\boldsymbol{a}_1}(\boldsymbol{x})}{w_1 + w_2 \gamma_{\boldsymbol{a}_2}(\boldsymbol{x})/\gamma_{\boldsymbol{a}_1}(\boldsymbol{x})} - \left\{ \boldsymbol{a}_1 + \frac{\hat{w}_2(\boldsymbol{a}_2 - \boldsymbol{a}_1)\gamma_{\boldsymbol{a}_2}(\boldsymbol{x})/\gamma_{\boldsymbol{a}_1}(\boldsymbol{x})}{\hat{w}_1 + \hat{w}_2 \gamma_{\boldsymbol{a}_2}(\boldsymbol{x})/\gamma_{\boldsymbol{a}_1}(\boldsymbol{x})} \right\} \\
&= \frac{w_2(\boldsymbol{a}_2 - \boldsymbol{a}_1)\gamma_{\boldsymbol{a}_2}(\boldsymbol{x})/\gamma_{\boldsymbol{a}_1}(\boldsymbol{x})}{w_1 + w_2 \gamma_{\boldsymbol{a}_2}(\boldsymbol{x})/\gamma_{\boldsymbol{a}_1}(\boldsymbol{x})} - \frac{\hat{w}_2(\boldsymbol{a}_2 - \boldsymbol{a}_1)\gamma_{\boldsymbol{a}_2}(\boldsymbol{x})/\gamma_{\boldsymbol{a}_1}(\boldsymbol{x})}{\hat{w}_1 + \hat{w}_2 \gamma_{\boldsymbol{a}_2}(\boldsymbol{x})/\gamma_{\boldsymbol{a}_1}(\boldsymbol{x})}.
\end{aligned}
\tag{40}
$$

Note that $|\gamma_{\boldsymbol{a}_2}(\boldsymbol{x})/\gamma_{\boldsymbol{a}_1}(\boldsymbol{x})|^2 = \exp(\|\boldsymbol{x} - \boldsymbol{a}_1\|^2 - \|\boldsymbol{x} - \boldsymbol{a}_2\|^2) \leqslant \exp(-\|\boldsymbol{a}_1 - \boldsymbol{a}_2\|^2/2)$ for every $\boldsymbol{x} \in \Omega_1$. Then use (40), we obtain

$$
\begin{aligned}
&\int_{\Omega_1} \left\| \nabla_{\boldsymbol{x}} \log \pi^M(\boldsymbol{x}) - \nabla_{\boldsymbol{x}} \log \hat{\pi}^M(\boldsymbol{x}) \right\|^2 p(\boldsymbol{x})\, \mathrm{d}\boldsymbol{x} \\
&= \int_{\Omega_1} \left\| \frac{w_2(\boldsymbol{a}_2 - \boldsymbol{a}_1)\gamma_{\boldsymbol{a}_2}(\boldsymbol{x})/\gamma_{\boldsymbol{a}_1}(\boldsymbol{x})}{w_1 + w_2 \gamma_{\boldsymbol{a}_2}(\boldsymbol{x})/\gamma_{\boldsymbol{a}_1}(\boldsymbol{x})} - \frac{\hat{w}_2(\boldsymbol{a}_2 - \boldsymbol{a}_1)\gamma_{\boldsymbol{a}_2}(\boldsymbol{x})/\gamma_{\boldsymbol{a}_1}(\boldsymbol{x})}{\hat{w}_1 + \hat{w}_2 \gamma_{\boldsymbol{a}_2}(\boldsymbol{x})/\gamma_{\boldsymbol{a}_1}(\boldsymbol{x})} \right\|^2 p(\boldsymbol{x})\, \mathrm{d}\boldsymbol{x} \\
&\leqslant 2 \int_{\Omega_1} \left\| \frac{w_2(\boldsymbol{a}_2 - \boldsymbol{a}_1)}{w_1} \right\|^2 \left| \frac{\gamma_{\boldsymbol{a}_2}(\boldsymbol{x})}{\gamma_{\boldsymbol{a}_1}(\boldsymbol{x})} \right|^2 p(\boldsymbol{x})\, \mathrm{d}\boldsymbol{x} + 2 \int_{\Omega_1} \left\| \frac{\hat{w}_2(\boldsymbol{a}_2 - \boldsymbol{a}_1)}{\hat{w}_1} \right\|^2 \left| \frac{\gamma_{\boldsymbol{a}_2}(\boldsymbol{x})}{\gamma_{\boldsymbol{a}_1}(\boldsymbol{x})} \right|^2 p(\boldsymbol{x})\, \mathrm{d}\boldsymbol{x} \\
&\leqslant 2 \exp\left( -\frac{\|\boldsymbol{a}_1 - \boldsymbol{a}_2\|^2}{2} \right) \left[ \frac{w_2^2}{w_1^2} + \frac{\hat{w}_2^2}{\hat{w}_1^2} \right] \|\boldsymbol{a}_2 - \boldsymbol{a}_1\|^2.
\end{aligned}
\tag{41}
$$

Similarly, we obtain

$$
\begin{aligned}
&\int_{\Omega_2} \left\| \nabla_{\boldsymbol{x}} \log \pi^M(\boldsymbol{x}) - \nabla_{\boldsymbol{x}} \log \hat{\pi}^M(\boldsymbol{x}) \right\|^2 p(\boldsymbol{x})\, \mathrm{d}\boldsymbol{x} \\
&\leqslant 2 \exp\left( -\frac{\|\boldsymbol{a}_1 - \boldsymbol{a}_2\|^2}{2} \right) \left[ \frac{w_1^2}{w_2^2} + \frac{\hat{w}_1^2}{\hat{w}_2^2} \right] \|\boldsymbol{a}_1 - \boldsymbol{a}_2\|^2.
\end{aligned}
\tag{42}
$$

Using (39), we obtain that

$$
\begin{aligned}
&\int_{\Omega_3} \left\| \nabla_{\boldsymbol{x}} \log \pi^M(\boldsymbol{x}) - \nabla_{\boldsymbol{x}} \log \hat{\pi}^M(\boldsymbol{x}) \right\|^2 p(\boldsymbol{x})\, \mathrm{d}\boldsymbol{x} \\
&= \int_{\Omega_3} \left\| \frac{w_1 \boldsymbol{a}_1 + w_2 \boldsymbol{a}_2 \gamma_{\boldsymbol{a}_2}(\boldsymbol{x})/\gamma_{\boldsymbol{a}_1}(\boldsymbol{x})}{w_1 + w_2 \gamma_{\boldsymbol{a}_2}(\boldsymbol{x})/\gamma_{\boldsymbol{a}_1}(\boldsymbol{x})} - \frac{\hat{w}_1 \boldsymbol{a}_1 + \hat{w}_2 \boldsymbol{a}_2 \gamma_{\boldsymbol{a}_2}(\boldsymbol{x})/\gamma_{\boldsymbol{a}_1}(\boldsymbol{x})}{\hat{w}_1 + \hat{w}_2 \gamma_{\boldsymbol{a}_2}(\boldsymbol{x})/\gamma_{\boldsymbol{a}_1}(\boldsymbol{x})} \right\|^2 p(\boldsymbol{x})\, \mathrm{d}\boldsymbol{x} \\
&\leqslant 4 \int_{\Omega_3} \left\| \frac{w_1 \boldsymbol{a}_1}{w_1 + w_2 \gamma_{\boldsymbol{a}_2}(\boldsymbol{x})/\gamma_{\boldsymbol{a}_1}(\boldsymbol{x})} \right\|^2 p(\boldsymbol{x})\, \mathrm{d}\boldsymbol{x} + 4 \int_{\Omega_3} \left\| \frac{\hat{w}_1 \boldsymbol{a}_1}{\hat{w}_1 + \hat{w}_2 \gamma_{\boldsymbol{a}_2}(\boldsymbol{x})/\gamma_{\boldsymbol{a}_1}(\boldsymbol{x})} \right\|^2 p(\boldsymbol{x})\, \mathrm{d}\boldsymbol{x} \\
&\quad + 4 \int_{\Omega_3} \left\| \frac{w_2 \boldsymbol{a}_2 \gamma_{\boldsymbol{a}_2}(\boldsymbol{x})/\gamma_{\boldsymbol{a}_1}(\boldsymbol{x})}{w_1 + w_2 \gamma_{\boldsymbol{a}_2}(\boldsymbol{x})/\gamma_{\boldsymbol{a}_1}(\boldsymbol{x})} \right\|^2 p(\boldsymbol{x})\, \mathrm{d}\boldsymbol{x} + 4 \int_{\Omega_3} \left\| \frac{\hat{w}_2 \boldsymbol{a}_2 \gamma_{\boldsymbol{a}_2}(\boldsymbol{x})/\gamma_{\boldsymbol{a}_1}(\boldsymbol{x})}{\hat{w}_1 + \hat{w}_2 \gamma_{\boldsymbol{a}_2}(\boldsymbol{x})/\gamma_{\boldsymbol{a}_1}(\boldsymbol{x})} \right\|^2 p(\boldsymbol{x})\, \mathrm{d}\boldsymbol{x} \\
&\leqslant 8 \left[ \|\boldsymbol{a}_1\|^2 + \|\boldsymbol{a}_2\|^2 \right] \int_{\Omega_3} p(\boldsymbol{x})\, \mathrm{d}\boldsymbol{x}.
\end{aligned}
\tag{43}
$$

Note that we have the following decomposition

$$
\begin{aligned}
\mathrm{SE}_p(\pi^M \| \hat{\pi}^M) &= \int_{\mathbb{R}^d} \left\| \nabla_{\boldsymbol{x}} \log \pi^M(\boldsymbol{x}) - \nabla_{\boldsymbol{x}} \log \hat{\pi}^M(\boldsymbol{x}) \right\|^2 p(\boldsymbol{x}) \, \mathrm{d}\boldsymbol{x} \\
&= \left( \int_{\Omega_1} + \int_{\Omega_2} + \int_{\Omega_3} \right) \left\| \nabla_{\boldsymbol{x}} \log \pi^M(\boldsymbol{x}) - \nabla_{\boldsymbol{x}} \log \hat{\pi}^M(\boldsymbol{x}) \right\|^2 p(\boldsymbol{x}) \, \mathrm{d}\boldsymbol{x}.
\end{aligned}
\tag{44}
$$

Plugging (41), (42), and (43) into (44), we obtain the upper bound (27) in Theorem 4. $\qquad \square$

Similarly, we have the following corollary, providing an example that shares the same property as Example 1, but with bounded variance.

**Corollary 1.** *Consider two MoGs in $\mathbb{R}^d$:*

$$
\pi^M = w_1 \mathcal{N}(\boldsymbol{a}_1, \sigma^2 I_d) + w_2 \mathcal{N}(\boldsymbol{a}_2, \sigma^2 I_d), \ \hat{\pi}^M = \hat{w}_1 \mathcal{N}(\boldsymbol{a}_1, \sigma^2 I_d) + \hat{w}_2 \mathcal{N}(\boldsymbol{a}_2, \sigma^2 I_d),
$$

*where $\boldsymbol{a}_1, \boldsymbol{a}_2 \in \mathbb{R}^d$, $w_1, w_2, \hat{w}_1, \hat{w}_2 > 0$, $\sigma^2 > 0$, $w_1 + w_2 = 1$ and $\hat{w}_1 + \hat{w}_2 = 1$. Then $\mathrm{KL}\left(\pi^M \| \hat{\pi}^M\right)$ is lower bounded by*

$$
\begin{aligned}
\mathrm{KL}\left(\pi^M | \hat{\pi}^M\right) \geqslant \ & w_1 \left( \log w_1 - \log \left( \hat{w}_1 + \exp \left( -\frac{\|\boldsymbol{a}_1 - \boldsymbol{a}_2\|^2}{4\sigma^2} \right) \right) \right) \\
& + w_2 \left( \log w_2 - \log \left( \hat{w}_2 + \exp \left( -\frac{\|\boldsymbol{a}_1 - \boldsymbol{a}_2\|^2}{4\sigma^2} \right) \right) \right) \\
& - (\log 4 + d) \exp \left( \frac{d}{2} \log 2 - \frac{\|\boldsymbol{a}_1 - \boldsymbol{a}_2\|^2}{64\sigma^2} \right),
\end{aligned}
$$

*and the Fisher divergence can be bounded by*

$$
\begin{aligned}
F(\pi^M, \hat{\pi}^M) \leqslant \ & 2 \exp \left( -\frac{\|\boldsymbol{a}_1 - \boldsymbol{a}_2\|^2}{2\sigma^2} \right) \left[ \frac{w_2^2}{w_1^2} + \frac{\hat{w}_2^2}{\hat{w}_1^2} + \frac{w_1^2}{w_2^2} + \frac{\hat{w}_1^2}{\hat{w}_2^2} \right] \frac{\|\boldsymbol{a}_1 - \boldsymbol{a}_2\|^2}{\sigma^4} \\
& + \frac{8 \left[ \|\boldsymbol{a}_1\|^2 + \|\boldsymbol{a}_2\|^2 \right]}{\sigma^4} \exp \left( \frac{d}{2} \log 2 - \frac{\|\boldsymbol{a}_1 - \boldsymbol{a}_2\|^2}{64\sigma^2} \right).
\end{aligned}
$$

### A.3 PROOF OF THEOREM 2

First, we present the divergence theorem and Green's first identity, which is very useful in our proof. Then we state the Grönwall's inequality used in our proof. Finally, we state and prove Theorem 5 which includes Theorem 2 and sharper bounds when (49) holds.

**Lemma 1** (divergence theorem). *Let $\mathbf{F}(\cdot) : \Omega \to \mathbb{R}^d$, then $\int_\Omega \nabla \cdot \mathbf{F}(\boldsymbol{x}) \, \mathrm{d}\boldsymbol{x} = \int_{\partial\Omega} \mathbf{F} \cdot \boldsymbol{n} \, \mathrm{d}\boldsymbol{S}$.*

**Lemma 2** (Green's first identity). *Let $v(\cdot), u(\cdot) : \Omega \to \mathbb{R}$, then it holds that*

$$
\int_\Omega \nabla_{\boldsymbol{x}} v \cdot \nabla_{\boldsymbol{x}} u \, \mathrm{d}\boldsymbol{x} + \int_\Omega v \Delta u \, \mathrm{d}\boldsymbol{x} = \int_{\partial\Omega} v \frac{\partial u}{\partial \boldsymbol{n}} \, \mathrm{d}\boldsymbol{S}.
$$

**Lemma 3** (Grönwall's inequality). *Let $f(\cdot), \alpha(\cdot), \beta(\cdot) : [0, T] \to \mathbb{R}$, and suppose that $\forall \, 0 \leqslant t \leqslant T$,*

$$
f'(t) \leqslant \alpha(t) + \beta(t) f(t).
$$

*Then we have $\forall \, 0 \leqslant t \leqslant T$,*

$$
f(t) \leqslant e^{\int_0^t \beta(s) \, \mathrm{d}s} f(0) + \int_0^t e^{\int_s^t \beta(r) \, \mathrm{d}r} \alpha(s) \, \mathrm{d}s.
$$

*Proof of Lemma 3.* Consider $g(t) = e^{-\int_0^t \beta(s) \, \mathrm{d}s} f(t), \forall \, 0 \leqslant t \leqslant T$. Then we have $\forall \, 0 \leqslant t \leqslant T$,

$$
\begin{aligned}
g'(t) &= e^{-\int_0^t \beta(s) \, \mathrm{d}s} f'(t) - \beta(t) e^{-\int_0^t \beta(s) \, \mathrm{d}s} f(t) \\
&= e^{-\int_0^t \beta(s) \, \mathrm{d}s} \left( f'(t) - \beta(t) f(t) \right) \\
&\leqslant e^{-\int_0^t \beta(s) \, \mathrm{d}s} \alpha(t).
\end{aligned}
\tag{45}
$$

Integrating (45), we obtain

$$e^{-\int_0^t \beta(s)\,\mathrm{d}s} f(t) \leqslant f(0) + \int_0^t e^{-\int_0^s \beta(r)\,\mathrm{d}r} \alpha(s)\,\mathrm{d}s. \tag{46}$$

Hence, we complete our proof. □

**Theorem 5.** *Suppose that Assumption 1, 2, and 3 hold. We further assume that $u_\theta(\boldsymbol{x}, 0) = u_0^*(\boldsymbol{x})$ for any $\boldsymbol{x} \in \Omega$. Then for any positive constant $\varepsilon > 0$, the following holds for any $0 \leqslant t \leqslant T$,*

$$\|e_t(\cdot)\|_{L^2(\Omega;\nu_t)}^2 \leqslant \varepsilon L_{\mathrm{PINN}}(t; C_1(\varepsilon)). \tag{47}$$

*Moreover, for any $0 \leqslant t \leqslant T$,*

$$m_1 \|\nabla_{\boldsymbol{x}} e_t(\cdot)\|_{L^2(\Omega;\nu_t)}^2 \leqslant \varepsilon \|r_t(\cdot)\|_{L^2(\Omega;\nu_t)}^2 + C_3(\varepsilon) L_{\mathrm{PINN}}(t; C_1(\varepsilon)) + C_2 \sqrt{\varepsilon L_{\mathrm{PINN}}(t; C_1(\varepsilon))}. \tag{48}$$

*In addition, if there exists constant $\mathcal{C}_\nu(\Omega) > 0$ such that the following holds for any $0 \leqslant t \leqslant T$,*

$$\|\nabla_{\boldsymbol{x}} e_t(\cdot)\|_{L^2(\Omega;\nu_t)}^2 \geqslant \mathcal{C}_\nu^2(\Omega) \|e_t(\cdot)\|_{L^2(\Omega;\nu_t)}^2. \tag{49}$$

*Then for any positive constant $\varepsilon > 0$, the following holds for any $0 \leqslant t \leqslant T$,*

$$\|e_t(\cdot)\|_{L^2(\Omega;\nu_t)}^2 \leqslant \varepsilon L_{\mathrm{PINN}}(t; C_4(\varepsilon)). \tag{50}$$

*Moreover, for any $0 \leqslant t \leqslant T$,*

$$m_1 \|\nabla_{\boldsymbol{x}} e_t(\cdot)\|_{L^2(\Omega;\nu_t)}^2 \leqslant \varepsilon \|r_t(\cdot)\|_{L^2(\Omega;\nu_t)}^2 + C_3(\varepsilon) L_{\mathrm{PINN}}(t; C_4(\varepsilon)) + C_2 \sqrt{\varepsilon L_{\mathrm{PINN}}(t; C_4(\varepsilon))}, \tag{51}$$

*where $C_2 := 2\sqrt{2}(\widehat{B}_0^2 + B_0^{*2})^{1/2}$, $C_3(\varepsilon) := \varepsilon(C_1(\varepsilon) + B_0^\nu)$, $C_4(\varepsilon) := C_1(\varepsilon) - m_1\mathcal{C}_\nu^2(\Omega)$ and*

$$C_1(\varepsilon) := \frac{1}{\varepsilon} + \frac{M_1}{4}(B_1^\nu + 2B_1^* + 2\widehat{B}_1) + c_1(B_1^\nu + B_2^\nu) - \frac{c_2}{2}(B_2^* + \widehat{B}_2),$$

*where*

$$c_1 := \begin{cases} M_1, & \text{if } B_1^\nu + B_2^\nu \geqslant 0 \\ m_1, & \text{if } B_1^\nu + B_2^\nu < 0 \end{cases} \quad , \quad c_2 := \begin{cases} m_1, & \text{if } B_2^* + \widehat{B}_2 \geqslant 0 \\ M_1, & \text{if } B_2^* + \widehat{B}_2 < 0 \end{cases} .$$

*Proof of Theorem 5.* We first prove (47) and (50). Note that $u_t^*(\boldsymbol{x})$ satisfies

$$\partial_t u_t^*(\boldsymbol{x}) + \nabla_{\boldsymbol{x}} u_t^*(\boldsymbol{x}) \cdot \boldsymbol{f}(\boldsymbol{x}, t) + \nabla \cdot \boldsymbol{f}(\boldsymbol{x}, t) - \frac{1}{2}g^2(t)\Delta u_t^*(\boldsymbol{x}) - \frac{1}{2}g^2(t)\|\nabla_{\boldsymbol{x}} u_t^*(\boldsymbol{x})\|^2 = 0, \tag{52}$$

and $u_\theta(\boldsymbol{x}, t)$ satisfies

$$\partial_t u_\theta(\boldsymbol{x}, t) + \nabla_{\boldsymbol{x}} u_\theta(\boldsymbol{x}, t) \cdot \boldsymbol{f}(\boldsymbol{x}, t) + \nabla \cdot \boldsymbol{f}(\boldsymbol{x}, t) - \frac{1}{2}g^2(t)\Delta u_\theta(\boldsymbol{x}, t) - \frac{1}{2}g^2(t)\|\nabla_{\boldsymbol{x}} u_\theta(\boldsymbol{x}, t)\|^2 = r_t(\boldsymbol{x}). \tag{53}$$

Subtracting (52) for $u^*$ from (53) for $u_\theta$, we have

$$\partial_t e_t(\boldsymbol{x}) + \nabla_{\boldsymbol{x}} e_t(\boldsymbol{x}) \cdot \boldsymbol{f}(\boldsymbol{x}, t) - \frac{1}{2}g^2(t)\left(\|\nabla_{\boldsymbol{x}} u_\theta(\boldsymbol{x}, t)\|^2 - \|\nabla_{\boldsymbol{x}} u_t^*(\boldsymbol{x})\|^2\right) - \frac{1}{2}g^2(t)\Delta e_t(\boldsymbol{x}) = r_t(\boldsymbol{x}). \tag{54}$$

Note that $\frac{1}{2}\partial_t e_t^2(\boldsymbol{x}) = e_t(\boldsymbol{x})\partial_t e_t(\boldsymbol{x})$ and $\frac{1}{2}\nabla_{\boldsymbol{x}} e_t^2(\boldsymbol{x}) = e_t(\boldsymbol{x})\nabla_{\boldsymbol{x}} e_t(\boldsymbol{x})$, then we obtain

$$\begin{aligned}
\frac{1}{2}\partial_t e_t^2(\boldsymbol{x}) &= \frac{1}{2}g^2(t)e_t(\boldsymbol{x})\left(\|\nabla_{\boldsymbol{x}} u_\theta(\boldsymbol{x}, t)\|^2 - \|\nabla_{\boldsymbol{x}} u_t^*(\boldsymbol{x})\|^2\right) + \frac{1}{2}g^2(t)e_t(\boldsymbol{x})\Delta e_t(\boldsymbol{x}) \\
&\quad + e_t(\boldsymbol{x})r_t(\boldsymbol{x}) - e_t(\boldsymbol{x})\nabla_{\boldsymbol{x}} e_t(\boldsymbol{x}) \cdot \boldsymbol{f}(\boldsymbol{x}, t) \\
&= \frac{1}{2}g^2(t)e_t(\boldsymbol{x})\nabla_{\boldsymbol{x}} e_t(\boldsymbol{x}) \cdot (\nabla_{\boldsymbol{x}} u_\theta(\boldsymbol{x}, t) + \nabla_{\boldsymbol{x}} u_t^*(\boldsymbol{x})) + \frac{1}{2}g^2(t)e_t(\boldsymbol{x})\Delta e_t(\boldsymbol{x}) \\
&\quad + e_t(\boldsymbol{x})r_t(\boldsymbol{x}) - e_t(\boldsymbol{x})\nabla_{\boldsymbol{x}} e_t(\boldsymbol{x}) \cdot \boldsymbol{f}(\boldsymbol{x}, t) \\
&= \frac{1}{4}g^2(t)\nabla_{\boldsymbol{x}} e_t^2(\boldsymbol{x}) \cdot (\nabla_{\boldsymbol{x}} u_\theta(\boldsymbol{x}, t) + \nabla_{\boldsymbol{x}} u_t^*(\boldsymbol{x})) + \frac{1}{2}g^2(t)e_t(\boldsymbol{x})\Delta e_t(\boldsymbol{x}) \\
&\quad + e_t(\boldsymbol{x})r_t(\boldsymbol{x}) - \frac{1}{2}\nabla_{\boldsymbol{x}} e_t^2(\boldsymbol{x}) \cdot f(\boldsymbol{x}, t).
\end{aligned} \tag{55}$$

Note that $\partial_t(\nu_t(\boldsymbol{x})e_t^2(\boldsymbol{x})) = e_t^2(\boldsymbol{x})\partial_t\nu_t(\boldsymbol{x}) + \nu_t(\boldsymbol{x})\partial_t e_t^2(\boldsymbol{x})$, then we have

$$
\begin{aligned}
\partial_t(\nu_t(\boldsymbol{x})e_t^2(\boldsymbol{x})) = {} & \frac{1}{2}g^2(t)\nu_t(\boldsymbol{x})\nabla_{\boldsymbol{x}}e_t^2(\boldsymbol{x}) \cdot (\nabla_{\boldsymbol{x}}u_\theta(\boldsymbol{x},t) + \nabla_{\boldsymbol{x}}u_t^*(\boldsymbol{x})) \\
& + g^2(t)\nu_t(\boldsymbol{x})e_t(\boldsymbol{x})\Delta e_t(\boldsymbol{x}) \\
& + 2\nu_t(\boldsymbol{x})e_t(\boldsymbol{x})r_t(\boldsymbol{x}) - \nu_t(\boldsymbol{x})\nabla_{\boldsymbol{x}}e_t^2(\boldsymbol{x}) \cdot \boldsymbol{f}(\boldsymbol{x},t) \\
& + \frac{1}{2}g^2(t)e_t^2(\boldsymbol{x})\Delta\nu_t(\boldsymbol{x}) - e_t^2(\boldsymbol{x})\nabla \cdot [\boldsymbol{f}(\boldsymbol{x},t)\nu_t(\boldsymbol{x})] .
\end{aligned}
\tag{56}
$$

We integrate (56) to get an equation for $\|e_t(\cdot)\|_{L^2(\Omega;\nu_t)}^2$ given by

$$
\begin{aligned}
\partial_t \|e_t(\cdot)\|_{L^2(\Omega;\nu_t)}^2 = {} & \frac{1}{2}g^2(t)\int_\Omega \nu_t(\boldsymbol{x})\nabla_{\boldsymbol{x}}e_t^2(\boldsymbol{x}) \cdot (\nabla_{\boldsymbol{x}}u_\theta(\boldsymbol{x},t) + \nabla_{\boldsymbol{x}}u_t^*(\boldsymbol{x})) \ \mathrm{d}\boldsymbol{x} \\
& + g^2(t)\int_\Omega \nu_t(\boldsymbol{x})e_t(\boldsymbol{x})\Delta e_t(\boldsymbol{x}) \ \mathrm{d}\boldsymbol{x} \\
& + 2\int_\Omega \nu_t(\boldsymbol{x})e_t(\boldsymbol{x})r_t(\boldsymbol{x}) \ \mathrm{d}\boldsymbol{x} - \int_\Omega \nu_t(\boldsymbol{x})\nabla_{\boldsymbol{x}}e_t^2(\boldsymbol{x}) \cdot \boldsymbol{f}(\boldsymbol{x},t) \ \mathrm{d}\boldsymbol{x} \\
& + \frac{1}{2}g^2(t)\int_\Omega e_t^2(\boldsymbol{x})\Delta\nu_t(\boldsymbol{x}) \ \mathrm{d}\boldsymbol{x} - \int_\Omega e_t^2(\boldsymbol{x})\nabla \cdot [\boldsymbol{f}(\boldsymbol{x},t)\nu_t(\boldsymbol{x})] \ \mathrm{d}\boldsymbol{x}.
\end{aligned}
\tag{57}
$$

Note that

$$
\nabla \cdot \left[\nu_t(\boldsymbol{x})e_t^2(\boldsymbol{x})\boldsymbol{f}(\boldsymbol{x},t)\right] = \nu_t(\boldsymbol{x})\nabla_{\boldsymbol{x}}e_t^2(\boldsymbol{x}) \cdot \boldsymbol{f}(\boldsymbol{x},t) + e_t^2(\boldsymbol{x})\nabla \cdot [\nu_t(\boldsymbol{x})\boldsymbol{f}(\boldsymbol{x},t)] .
$$

Then using Lemma 1 and $e_t(\boldsymbol{x}) = 0$ for any $(\boldsymbol{x},t) \in \partial\Omega \times [0,T]$, we have

$$
\int_\Omega \nu_t(\boldsymbol{x})\nabla_{\boldsymbol{x}}e_t^2(\boldsymbol{x}) \cdot \boldsymbol{f}(\boldsymbol{x},t) \ \mathrm{d}\boldsymbol{x} + \int_\Omega e_t^2(\boldsymbol{x})\nabla \cdot [\nu_t(\boldsymbol{x})\boldsymbol{f}(\boldsymbol{x},t)] \ \mathrm{d}\boldsymbol{x} = 0.
\tag{58}
$$

Similarly, we have

$$
\begin{aligned}
& \int_\Omega \nu_t(\boldsymbol{x})\nabla_{\boldsymbol{x}}e_t^2(\boldsymbol{x}) \cdot (\nabla_{\boldsymbol{x}}u_\theta(\boldsymbol{x},t) + \nabla_{\boldsymbol{x}}u_t^*(\boldsymbol{x})) \ \mathrm{d}\boldsymbol{x} \\
= {} & -\int_\Omega \nu_t(\boldsymbol{x})e_t^2(\boldsymbol{x}) \left(\Delta u_\theta(\boldsymbol{x},t) + \Delta u_t^*(\boldsymbol{x})\right) \ \mathrm{d}\boldsymbol{x} \\
& -\int_\Omega e_t^2(\boldsymbol{x})\nabla_{\boldsymbol{x}}\nu_t(\boldsymbol{x}) \cdot (\nabla_{\boldsymbol{x}}u_\theta(\boldsymbol{x},t) + \nabla_{\boldsymbol{x}}u_t^*(\boldsymbol{x})) \ \mathrm{d}\boldsymbol{x},
\end{aligned}
\tag{59}
$$

and

$$
\int_\Omega \nu_t(\boldsymbol{x})e_t(\boldsymbol{x})\Delta e_t(\boldsymbol{x}) \ \mathrm{d}\boldsymbol{x} = -\frac{1}{2}\int_\Omega \nabla_{\boldsymbol{x}}\nu_t(\boldsymbol{x}) \cdot \nabla_{\boldsymbol{x}}e_t^2(\boldsymbol{x}) \ \mathrm{d}\boldsymbol{x} - \int_\Omega \nu_t(\boldsymbol{x})\|\nabla_{\boldsymbol{x}}e_t(\boldsymbol{x})\|^2 \ \mathrm{d}\boldsymbol{x}.
\tag{60}
$$

Plugging (58), (59), and (60) into (57), and using Lemma 2, we have

$$
\begin{aligned}
\partial_t\|e_t(\cdot)\|_{L^2(\Omega;\nu_t)}^2 = {} & -\frac{1}{2}g^2(t)\int_\Omega (\Delta u_\theta(\boldsymbol{x},t) + \Delta u_t^*(\boldsymbol{x}))e_t^2(\boldsymbol{x})\nu_t(\boldsymbol{x}) \ \mathrm{d}\boldsymbol{x} \\
& -\frac{1}{2}g^2(t)\int_\Omega \nabla_{\boldsymbol{x}}\nu_t(\boldsymbol{x}) \cdot (\nabla_{\boldsymbol{x}}u_\theta(\boldsymbol{x},t) + \nabla_{\boldsymbol{x}}u_t^*(\boldsymbol{x})) \, e_t^2(\boldsymbol{x}) \ \mathrm{d}\boldsymbol{x} \\
& -g^2(t)\int_\Omega \nu_t(\boldsymbol{x})\|\nabla_{\boldsymbol{x}}e_t(\boldsymbol{x})\|^2 \ \mathrm{d}\boldsymbol{x} + 2\int_\Omega \nu_t(\boldsymbol{x})e_t(\boldsymbol{x})r_t(\boldsymbol{x}) \ \mathrm{d}\boldsymbol{x} \\
& + g^2(t)\int_\Omega e_t^2(\boldsymbol{x})\Delta\nu_t(\boldsymbol{x}) \ \mathrm{d}\boldsymbol{x}.
\end{aligned}
\tag{61}
$$

Using $\nabla_{\boldsymbol{x}}\nu_t(\boldsymbol{x}) = \nu_t(\boldsymbol{x})\nabla_{\boldsymbol{x}}\log\nu_t(\boldsymbol{x})$ and $\Delta\nu_t(\boldsymbol{x}) = (\Delta\log\nu_t(\boldsymbol{x}) + \|\nabla_{\boldsymbol{x}}\log\nu_t(\boldsymbol{x})\|^2)\nu_t(\boldsymbol{x})$,

$$
\begin{aligned}
\partial_t\|e_t(\cdot)\|_{L^2(\Omega;\nu_t)}^2 = {} & -\frac{1}{2}g^2(t)\int_\Omega (\Delta u_\theta(\boldsymbol{x},t) + \Delta u_t^*(\boldsymbol{x}))e_t^2(\boldsymbol{x})\nu_t(\boldsymbol{x}) \ \mathrm{d}\boldsymbol{x} \\
& -\frac{1}{2}g^2(t)\int_\Omega \nabla_{\boldsymbol{x}}\log\nu_t(\boldsymbol{x}) \cdot (\nabla_{\boldsymbol{x}}u_\theta(\boldsymbol{x},t) + \nabla_{\boldsymbol{x}}u_t^*(\boldsymbol{x})) \, e_t^2(\boldsymbol{x})\nu_t(\boldsymbol{x}) \ \mathrm{d}\boldsymbol{x} \\
& -g^2(t)\int_\Omega \nu_t(\boldsymbol{x})\|\nabla_{\boldsymbol{x}}e_t(\boldsymbol{x})\|^2 \ \mathrm{d}\boldsymbol{x} + 2\int_\Omega \nu_t(\boldsymbol{x})e_t(\boldsymbol{x})r_t(\boldsymbol{x}) \ \mathrm{d}\boldsymbol{x} \\
& + g^2(t)\int_\Omega e_t^2(\boldsymbol{x})\nu_t(\boldsymbol{x})(\Delta\log\nu_t(\boldsymbol{x}) + \|\nabla_{\boldsymbol{x}}\log\nu_t(\boldsymbol{x})\|^2) \ \mathrm{d}\boldsymbol{x}.
\end{aligned}
\tag{62}
$$

By Assumption 2 and 3, then we have

$$
\begin{aligned}
\partial_t \|e_t(\cdot)\|^2_{L^2(\Omega;\nu_t)} &\leqslant \varepsilon \|r_t(\cdot)\|^2_{L^2(\Omega;\nu_t)} + C_1(\varepsilon)\|e_t(\cdot)\|^2_{L^2(\Omega;\nu_t)} - m_1\|\nabla_{\boldsymbol{x}} e_t(\cdot)\|^2_{L^2(\Omega;\nu_t)} \\
&\leqslant \varepsilon \|r_t(\cdot)\|^2_{L^2(\Omega;\nu_t)} + C_1(\varepsilon)\|e_t(\cdot)\|^2_{L^2(\Omega;\nu_t)},
\end{aligned}
\tag{63}
$$

which follows from applying Young's inequality and holds for any $\varepsilon > 0$. Note that $e_0(\boldsymbol{x}) = 0$ for any $\boldsymbol{x} \in \Omega$, then using Lemma 3, we have $\forall\, 0 \leqslant t \leqslant T$,

$$
\|e_t(\cdot)\|^2_{L^2(\Omega;\nu_t)} \leqslant \varepsilon \int_0^t e^{C_1(\varepsilon)(t-s)} \|r_s(\cdot)\|^2_{L^2(\Omega;\nu_s)} \,\mathrm{d}s := \varepsilon L_{\mathrm{PINN}}(t; C_1(\varepsilon)).
\tag{64}
$$

Hence, we have proved (47). In addition, if (49) holds, plugging (49) into (63), we have

$$
\partial_t \|e_t(\cdot)\|^2_{L^2(\Omega;\nu_t)} \leqslant \varepsilon \|r_t(\cdot)\|^2_{L^2(\Omega;\nu_t)} + C_4(\varepsilon)\|e_t(\cdot)\|^2_{L^2(\Omega;\nu_t)}.
\tag{65}
$$

Similarly, using Lemma 3, we obtain (50). Then we prove (48) and (51). From (63), we have

$$
m_1\|\nabla_{\boldsymbol{x}} e_t(\cdot)\|^2_{L^2(\Omega;\nu_t)} \leqslant \varepsilon \|r_t(\cdot)\|^2_{L^2(\Omega;\nu_t)} + C_1(\varepsilon)\|e_t(\cdot)\|^2_{L^2(\Omega;\nu_t)} - \partial_t \|e_t(\cdot)\|^2_{L^2(\Omega;\nu_t)}.
\tag{66}
$$

By Assumption 3, we bound $\partial_t \|e_t(\cdot)\|^2_{L^2(\Omega;\nu_t)}$ as follows,

$$
\begin{aligned}
\left| \partial_t \|e_t(\cdot)\|^2_{L^2(\Omega;\nu_t)} \right| &= \left| \partial_t \left( \int_\Omega e_t^2(\boldsymbol{x}) \nu_t(\boldsymbol{x}) \,\mathrm{d}\boldsymbol{x} \right) \right| \\
&= \left| \int_\Omega e_t^2(\boldsymbol{x}) \partial_t \nu_t(\boldsymbol{x}) + 2\nu_t(\boldsymbol{x}) e_t(\boldsymbol{x}) \partial_t e_t(\boldsymbol{x}) \,\mathrm{d}\boldsymbol{x} \right| \\
&\leqslant \int_\Omega e_t^2(\boldsymbol{x}) \nu_t(\boldsymbol{x}) |\partial_t \log \nu_t(\boldsymbol{x})| \,\mathrm{d}\boldsymbol{x} + 2\left| \int_\Omega \nu_t(\boldsymbol{x}) e_t(\boldsymbol{x}) \partial_t e_t(\boldsymbol{x}) \,\mathrm{d}\boldsymbol{x} \right| \\
&\leqslant B_0^\nu \|e_t(\cdot)\|^2_{L^2(\Omega;\nu_t)} + 2\left( \int_\Omega \nu_t(\boldsymbol{x}) e_t^2(\boldsymbol{x}) \,\mathrm{d}\boldsymbol{x} \right)^{1/2} \left( \int_\Omega \nu_t(\boldsymbol{x}) |\partial_t e_t(\boldsymbol{x})|^2 \,\mathrm{d}\boldsymbol{x} \right)^{1/2} \\
&\leqslant B_0^\nu \|e_t(\cdot)\|^2_{L^2(\Omega;\nu_t)} + 2\sqrt{2}\left( \widehat{B}_0^2 + B_0^{*2} \right)^{1/2} \|e_t(\cdot)\|_{L^2(\Omega;\nu_t)},
\end{aligned}
\tag{67}
$$

which follows from applying $|\partial_t e_t(\boldsymbol{x})|^2 = |\partial_t u_\theta(\boldsymbol{x}, t) - \partial_t u_t^*(\boldsymbol{x})|^2 \leqslant 2\widehat{B}_0^2 + 2B_0^{*2}$. Then plugging (67) into (66), we have

$$
m_1\|\nabla_{\boldsymbol{x}} e_t(\cdot)\|^2_{L^2(\Omega;\nu_t)} \leqslant \varepsilon \|r_t(\cdot)\|^2_{L^2(\Omega;\nu_t)} + (C_1(\varepsilon) + B_0^\nu)\|e_t(\cdot)\|^2_{L^2(\Omega;\nu_t)} + C_2\|e_t(\cdot)\|_{L^2(\Omega;\nu_t)}.
\tag{68}
$$

Plugging (47) and (50) into (68) gives (48) and (51) respectively. $\qquad\square$

## A.4 Proof of Theorem 3

Given the $L^2$ error of the score approximation, Chen et al. (2023a) provides an upper bound of KL divergence between the data distribution $\pi$ and the distribution of approximated samples $\hat{\pi}_T$ drawn from the sampling dynamics (19). We first summarize the results from Chen et al. (2023a) in Proposition 1. Then we prove Theorem 3 based on Proposition 1.

**Proposition 1** (Theorem 2.5 in Chen et al. (2023a)). *Suppose that $T \geqslant 1$, $K \geqslant 2$, and the $L^2$ error of the score approximation is bounded by*

$$
\sum_{k=1}^N h_k \mathbb{E}_{\boldsymbol{x}_{t_k} \sim \pi_{t_k}} \left\| \nabla_{\boldsymbol{x}} \log \pi_{t_k}(\boldsymbol{x}_{t_k}) - \boldsymbol{s}_{t_k}(\boldsymbol{x}_{t_k}) \right\|^2 \leqslant T\varepsilon_0^2.
\tag{69}
$$

*Then there is a universal constant $\alpha \geqslant 2$ such that the following holds. Under Assumption 4, by using the exponentially decreasing (then constant) step size $h_k = h \min\{\max\{t_k, 1/(4K)\}, 1\}$, $0 < h \leqslant 1/(\alpha d)$, the sampling dynamic (19) results in a distribution $\hat{\pi}_T$ such that*

$$
\mathrm{KL}(\pi \| \hat{\pi}_T) \lesssim (d + M_2) \cdot e^{-T} + T\varepsilon_0^2 + d^2 h(\log K + T),
\tag{70}
$$

*where the number of sampling steps satisfies that $N \lesssim \frac{1}{h}(\log K + T)$. Choosing $T = \log\left(\frac{M_2 + d}{\varepsilon_0^2}\right)$ and $h = \Theta\left(\frac{\varepsilon_0^2}{d^2(\log K + T)}\right)$, we have $N = \mathcal{O}\left(\frac{d^2(\log K + T)^2}{\varepsilon_0^2}\right)$ and make the KL divergence $\widetilde{\mathcal{O}}(\varepsilon_0^2)$.*

*Proof of Theorem 3.* As $\boldsymbol{s}_t(\boldsymbol{x}) = \nabla_{\boldsymbol{x}} u_\theta(\boldsymbol{x}, t) \cdot \mathbb{1}\{\boldsymbol{x} \in \Omega\}$, we have

$$
\sum_{k=1}^N h_k \mathbb{E}_{\boldsymbol{x}_{t_k} \sim \pi_{t_k}} \left\| \nabla_{\boldsymbol{x}} \log \pi_{t_k}(\boldsymbol{x}_{t_k}) - \boldsymbol{s}_{t_k}(\boldsymbol{x}_{t_k}) \right\|^2
$$

$$
= \sum_{k=1}^N h_k \int_{\Omega^c} \pi_{t_k}(\boldsymbol{x}) \| \nabla_{\boldsymbol{x}} \log \pi_{t_k}(\boldsymbol{x}) \|^2 \, \mathrm{d}\boldsymbol{x} + \sum_{k=1}^N h_k \int_\Omega \pi_{t_k}(\boldsymbol{x}) \| \nabla_{\boldsymbol{x}} e_{t_k}(\boldsymbol{x}) \|^2 \, \mathrm{d}\boldsymbol{x}
$$

$$
\leqslant \sum_{k=1}^N h_k \delta + \sum_{k=1}^N h_k R_{t_k} \| \nabla_{\boldsymbol{x}} e_{t_k}(\cdot) \|_{L^2(\Omega; \nu_{t_k})}^2 \quad \text{(using Assumption 5 and 6)} \tag{71}
$$

$$
\leqslant T\delta + \delta_1 + C_5(\varepsilon)\delta_2 + C_2 \sqrt{\sum_{k=1}^N h_k R_{t_k} \delta_2},
$$

where the last inequality follows from Theorem 2 ($m_1 = M_1 = 1$) and

$$
\sum_{k=1}^N h_k R_{t_k} \sqrt{\varepsilon L_{\text{PINN}}(t_k; C_1(\varepsilon))} \leqslant \left( \varepsilon \sum_{k=1}^N h_k R_{t_k} L_{\text{PINN}}(t_k; C_1(\varepsilon)) \right)^{1/2} \left( \sum_{k=1}^N h_k R_{t_k} \right)^{1/2}
$$

$$
\leqslant \sqrt{\sum_{k=1}^N h_k R_{t_k} \delta_2}. \tag{72}
$$

Then combining (71) and Proposition 1 together gives the results in Theorem 3. □

# B THEORETICAL COMPARISON BETWEEN DIFFERENT SAMPLING METHODS FOR COLLOCATION POINTS

## B.1 CONVERGENCE GUARANTEE OF PINN FOR SOLVING LOG-DENSITY FPE

In this section, we present a convergence guarantee of PINN for solving the log-density FPE on a constrained domain $\Omega$ and the convergence analysis of DPS when the collocation points are sampled from $\nu_t \sim \text{Unif}(\Omega)$. We make the following assumptions.

**Assumption 7.** *For any $t \in [0, T]$, $g^2(t)$ is lower-bounded: $g^2(t) \geqslant m_1$ for some $m_1 > 0$.*

**Assumption 8.** $u_t^*(\boldsymbol{x}), u_\theta(\boldsymbol{x}, t) \in \mathcal{C}^2(\Omega \times [0, T])$.

**Assumption 9.** *For any $(\boldsymbol{x}, t) \in \Omega \times [0, T]$, $\nabla \cdot \boldsymbol{f}(\boldsymbol{x}, t) \leqslant \mathrm{m}_2$ for some $\mathrm{m}_2 \in \mathbb{R}$.*

Based on Assumption 8, there exists $B_0^*, \widehat{B}_0, B_1^*, \widehat{B}_1 \in \mathbb{R}_+$ and $B_2^*, \widehat{B}_2 \in \mathbb{R}$ depended on $\Omega$ such that for any $(\boldsymbol{x}, t) \in \Omega \times [0, T]$, we have

$$
|\partial_t u_t^*(\boldsymbol{x})| \leqslant B_0^*, \quad \|\nabla_{\boldsymbol{x}} u_t^*(\boldsymbol{x})\|^2 \leqslant B_1^*, \quad \Delta u_t^*(\boldsymbol{x}) \geqslant B_2^*,
$$

$$
|\partial_t u_\theta(\boldsymbol{x}, t)| \leqslant \widehat{B}_0, \quad \|\nabla_{\boldsymbol{x}} u_\theta(\boldsymbol{x}, t)\|^2 \leqslant \widehat{B}_1, \quad \Delta u_\theta(\boldsymbol{x}, t) \geqslant \widehat{B}_2.
$$

**Theorem 6.** *Suppose that Assumption 1, 7, 8and 9 hold. And we define the PINN objective on $\Omega$ as*

$$
L_{\text{PINN}}^{\text{Unif}}(t; C) := \int_0^t e^{C(t-s)} \|r_s(\cdot)\|_{L^2(\Omega)}^2 \, \mathrm{d}s.
$$

*We further assume that $u_\theta(\boldsymbol{x}, 0) = u_0^*(\boldsymbol{x})$ for any $\boldsymbol{x} \in \Omega$. Then for any positive constant $\varepsilon > 0$, the following holds for any $t \in [0, T]$,*

$$
\|e_t(\cdot)\|_{L^2(\Omega)}^2 \leqslant \varepsilon L_{\text{PINN}}^{\text{Unif}}(t; C_1^{\text{U}}(\varepsilon)). \tag{73}
$$

*Moreover, for any $t \in [0, T]$,*

$$
m_1 \|\nabla_{\boldsymbol{x}} e_t(\cdot)\|_{L^2(\Omega)}^2 \leqslant \varepsilon \|r_t(\cdot)\|_{L^2(\Omega)}^2 + \varepsilon \cdot C_1^{\text{U}}(\varepsilon) L_{\text{PINN}}^{\text{Unif}}(t; C_1^{\text{U}}(\varepsilon)) + C_2^{\text{U}} \sqrt{\varepsilon L_{\text{PINN}}^{\text{Unif}}(t; C_1^{\text{U}}(\varepsilon))}, \tag{74}
$$

*where $C_2^{\text{U}} := 2\sqrt{2} \left( \widehat{B}_0^2 + B_0^{*2} \right)^{1/2}$ and $C_1^{\text{U}}(\varepsilon) := \frac{1}{\varepsilon} + \mathrm{m}_2 - \frac{m_1}{2} \left( B_2^* + \widehat{B}_2 \right)$.*

*Proof of Theorem 6.* Note that $u_t^*(\boldsymbol{x})$ satisfies

$$\partial_t u_t^*(\boldsymbol{x}) + \nabla_{\boldsymbol{x}} u_t^*(\boldsymbol{x}) \cdot \boldsymbol{f}(\boldsymbol{x}, t) + \nabla \cdot \boldsymbol{f}(\boldsymbol{x}, t) - \frac{1}{2} g^2(t) \Delta u_t^*(\boldsymbol{x}) - \frac{1}{2} g^2(t) \left\| \nabla_{\boldsymbol{x}} u_t^*(\boldsymbol{x}) \right\|^2 = 0, \quad (75)$$

and $u_\theta(\boldsymbol{x}, t)$ satisfies

$$\partial_t u_\theta(\boldsymbol{x}, t) + \nabla_{\boldsymbol{x}} u_\theta(\boldsymbol{x}, t) \cdot \boldsymbol{f}(\boldsymbol{x}, t) + \nabla \cdot \boldsymbol{f}(\boldsymbol{x}, t) - \frac{1}{2} g^2(t) \Delta u_\theta(\boldsymbol{x}, t) - \frac{1}{2} g^2(t) \left\| \nabla_{\boldsymbol{x}} u_\theta(\boldsymbol{x}, t) \right\|^2 = r_t(\boldsymbol{x}). \tag{76}$$

Subtracting (75) for $u^*$ from (76) for $u_\theta$, we have

$$\partial_t e_t(\boldsymbol{x}) + \nabla_{\boldsymbol{x}} e_t(\boldsymbol{x}) \cdot \boldsymbol{f}(\boldsymbol{x}, t) - \frac{1}{2} g^2(t) \left( \left\| \nabla_{\boldsymbol{x}} u_\theta(\boldsymbol{x}, t) \right\|^2 - \left\| \nabla_{\boldsymbol{x}} u_t^*(\boldsymbol{x}) \right\|^2 \right) - \frac{1}{2} g^2(t) \Delta e_t(\boldsymbol{x}) = r_t(\boldsymbol{x}). \tag{77}$$

Note that $\frac{1}{2} \partial_t e_t^2(\boldsymbol{x}) = e_t(\boldsymbol{x}) \partial_t e_t(\boldsymbol{x})$ and $\frac{1}{2} \nabla_{\boldsymbol{x}} e_t^2(\boldsymbol{x}) = e_t(\boldsymbol{x}) \nabla_{\boldsymbol{x}} e_t(\boldsymbol{x})$, then we obtain

$$\begin{aligned}
\frac{1}{2} \partial_t e_t^2(\boldsymbol{x}) &= \frac{1}{2} g^2(t) e_t(\boldsymbol{x}) \left( \left\| \nabla_{\boldsymbol{x}} u_\theta(\boldsymbol{x}, t) \right\|^2 - \left\| \nabla_{\boldsymbol{x}} u_t^*(\boldsymbol{x}) \right\|^2 \right) + \frac{1}{2} g^2(t) e_t(\boldsymbol{x}) \Delta e_t(\boldsymbol{x}) \\
&\qquad + e_t(\boldsymbol{x}) r_t(\boldsymbol{x}) - e_t(\boldsymbol{x}) \nabla_{\boldsymbol{x}} e_t(\boldsymbol{x}) \cdot \boldsymbol{f}(\boldsymbol{x}, t) \\
&= \frac{1}{2} g^2(t) e_t(\boldsymbol{x}) \nabla_{\boldsymbol{x}} e_t(\boldsymbol{x}) \cdot \left( \nabla_{\boldsymbol{x}} u_\theta(\boldsymbol{x}, t) + \nabla_{\boldsymbol{x}} u_t^*(\boldsymbol{x}) \right) + \frac{1}{2} g^2(t) e_t(\boldsymbol{x}) \Delta e_t(\boldsymbol{x}) \\
&\qquad + e_t(\boldsymbol{x}) r_t(\boldsymbol{x}) - e_t(\boldsymbol{x}) \nabla_{\boldsymbol{x}} e_t(\boldsymbol{x}) \cdot \boldsymbol{f}(\boldsymbol{x}, t) \\
&= \frac{1}{4} g^2(t) \nabla_{\boldsymbol{x}} e_t^2(\boldsymbol{x}) \cdot \left( \nabla_{\boldsymbol{x}} u_\theta(\boldsymbol{x}, t) + \nabla_{\boldsymbol{x}} u_t^*(\boldsymbol{x}) \right) + \frac{1}{2} g^2(t) e_t(\boldsymbol{x}) \Delta e_t(\boldsymbol{x}) \\
&\qquad + e_t(\boldsymbol{x}) r_t(\boldsymbol{x}) - \frac{1}{2} \nabla_{\boldsymbol{x}} e_t^2(\boldsymbol{x}) \cdot f(\boldsymbol{x}, t).
\end{aligned} \tag{78}$$

We integrate (78) to get an equation for $\| e_t(\cdot) \|_{L^2(\Omega)}^2$ given by

$$\begin{aligned}
\partial_t \| e_t(\cdot) \|_{L^2(\Omega)}^2 &= \frac{1}{2} g^2(t) \int_\Omega \nabla_{\boldsymbol{x}} e_t^2(\boldsymbol{x}) \cdot \left( \nabla_{\boldsymbol{x}} u_\theta(\boldsymbol{x}, t) + \nabla_{\boldsymbol{x}} u_t^*(\boldsymbol{x}) \right) \mathrm{d}\boldsymbol{x} + g^2(t) \int_\Omega e_t(\boldsymbol{x}) \Delta e_t(\boldsymbol{x}) \mathrm{d}\boldsymbol{x} \\
&\qquad + 2 \int_\Omega e_t(\boldsymbol{x}) r_t(\boldsymbol{x}) \mathrm{d}\boldsymbol{x} - \int_\Omega \nabla_{\boldsymbol{x}} e_t^2(\boldsymbol{x}) \cdot \boldsymbol{f}(\boldsymbol{x}, t) \mathrm{d}\boldsymbol{x} \\
&= -\frac{1}{2} g^2(t) \int_\Omega e_t^2(\boldsymbol{x}) \cdot \left( \Delta u_\theta(\boldsymbol{x}, t) + \Delta u_t^*(\boldsymbol{x}) \right) \mathrm{d}\boldsymbol{x} - g^2(t) \int_\Omega \left\| \nabla_{\boldsymbol{x}} e_t(\boldsymbol{x}) \right\|^2 \mathrm{d}\boldsymbol{x} \\
&\qquad + 2 \int_\Omega e_t(\boldsymbol{x}) r_t(\boldsymbol{x}) \mathrm{d}\boldsymbol{x} + \int_\Omega e_t^2(\boldsymbol{x}) \cdot \left[ \nabla \cdot \boldsymbol{f}(\boldsymbol{x}, t) \right] \mathrm{d}\boldsymbol{x} \\
&\leqslant -\frac{m_1}{2} \left( B_2^* + \widehat{B}_2 \right) \| e_t(\cdot) \|_{L^2(\Omega)}^2 - m_1 \| \nabla_{\boldsymbol{x}} e_t(\cdot) \|_{L^2(\Omega)}^2 + \varepsilon \| r_t(\cdot) \|_{L^2(\Omega)}^2 \\
&\qquad + \frac{1}{\varepsilon} \| e_t(\cdot) \|_{L^2(\Omega)}^2 + \mathrm{m}_2 \| e_t(\cdot) \|_{L^2(\Omega)}^2 \\
&= C_1^{\mathrm{U}}(\varepsilon) \| e_t(\cdot) \|_{L^2(\Omega)}^2 + \varepsilon \| r_t(\cdot) \|_{L^2(\Omega)}^2 - m_1 \| \nabla_{\boldsymbol{x}} e_t(\cdot) \|_{L^2(\Omega)}^2 \\
&\leqslant C_1^{\mathrm{U}}(\varepsilon) \| e_t(\cdot) \|_{L^2(\Omega)}^2 + \varepsilon \| r_t(\cdot) \|_{L^2(\Omega)}^2.
\end{aligned} \tag{79}$$

Note that $e_0(\boldsymbol{x}) = 0$ for any $\boldsymbol{x} \in \Omega$, then using the Grönwall inequality, we have for any $t \in [0, T]$,

$$\| e_t(\cdot) \|_{L^2(\Omega)}^2 \leqslant \varepsilon \int_0^t e^{C_1^{\mathrm{U}}(\varepsilon)(t-s)} \| r_s(\cdot) \|_{L^2(\Omega)}^2 \mathrm{d}s := \varepsilon L_{\mathrm{PINN}}^{\mathrm{Unif}}(t; C_1^{\mathrm{U}}(\varepsilon)). \tag{80}$$

Note that from (79),

$$m_1 \| \nabla_{\boldsymbol{x}} e_t(\cdot) \|_{L^2(\Omega)}^2 \leqslant \varepsilon \| r_t(\cdot) \|_{L^2(\Omega)}^2 + C_1^{\mathrm{U}}(\varepsilon) \| e_t(\cdot) \|_{L^2(\Omega)}^2 - \partial_t \| e_t(\cdot) \|_{L^2(\Omega)}^2. \tag{81}$$

We can bound $\partial_t \| e_t(\cdot) \|_{L^2(\Omega)}^2$ as follows

$$\begin{aligned}
\left| \partial_t \| e_t(\cdot) \|_{L^2(\Omega)}^2 \right| &= \left| \partial_t \left( \int_\Omega e_t^2(\boldsymbol{x}) \mathrm{d}\boldsymbol{x} \right) \right| = 2 \left| \int_\Omega e_t(\boldsymbol{x}) \partial_t e_t(\boldsymbol{x}) \mathrm{d}\boldsymbol{x} \right| \\
&\leqslant 2 \left( \int_\Omega e_t^2(\boldsymbol{x}) \mathrm{d}\boldsymbol{x} \right)^{1/2} \left( \int_\Omega |\partial_t e_t(\boldsymbol{x})|^2 \mathrm{d}\boldsymbol{x} \right)^{1/2} \leqslant 2\sqrt{2} \left( \widehat{B}_0^2 + B_0^{*2} \right)^{1/2} \| e_t(\cdot) \|_{L^2(\Omega)},
\end{aligned} \tag{82}$$

which follows from applying $|\partial_t e_t(\boldsymbol{x})|^2 = |\partial_t u_\theta(\boldsymbol{x}, t) - \partial_t u_t^*(\boldsymbol{x})|^2 \leqslant 2\widehat{B}_0^2 + 2B_0^{*2}$. Then, plugging (82) into (81), we have

$$m_1 \|\nabla_{\boldsymbol{x}} e_t(\cdot)\|_{L^2(\Omega)}^2 \leqslant \varepsilon \|r_t(\cdot)\|_{L^2(\Omega)}^2 + C_1^{\mathrm{U}}(\varepsilon) \|e_t(\cdot)\|_{L^2(\Omega)}^2 + C_2^{\mathrm{U}} \|e_t(\cdot)\|_{L^2(\Omega)}. \tag{83}$$

Plugging (81) into (83), we complete the proof. $\qquad\square$

## B.2 CONVERGENCE ANALYSIS OF DIFFUSION-PINN SAMPLER

In this section, we present our convergence analysis of DPS based on Theorem 6 and the analysis of score-based generative models in Chen et al. (2023a) when the collocation points are sampled from uniform distribution within the similar setting in section 5.2.

**Theorem 7.** *Suppose that $T \geqslant 1$, $K \geqslant 2$, and Assumption 1, 4, 5, 7, 8and 9 hold. For any $\delta > 0$, let $\Omega$ be chosen as in Assumption 5. For any positive constant $\varepsilon > 0$, we further assume that $u_\theta(\boldsymbol{x}, t)$ satisfies the following*[3],

$$\varepsilon \sum_{k=1}^{N} h_k \max_{\boldsymbol{x} \in \Omega} \{\pi_{t_k}(\boldsymbol{x})\} \cdot \|r_{t_k}(\cdot)\|_{L^2(\Omega)}^2 \leqslant \delta_1 \cdot \mathrm{Vol}(\Omega),$$

$$\varepsilon \sum_{k=1}^{N} h_k \max_{\boldsymbol{x} \in \Omega} \{\pi_{t_k}(\boldsymbol{x})\} \cdot L_{\mathrm{PINN}}^{\mathrm{Unif}}(t_k; C_1^{\mathrm{U}}(\varepsilon)) \leqslant \delta_2 \cdot \mathrm{Vol}(\Omega). \tag{84}$$

*Then there is a universal constant $\alpha \geqslant 2$ such that the following holds. Using step size $h_k := h \min\{\max\{t_k, \frac{1}{4K}\}\}$ for $0 < h \leqslant \frac{1}{\alpha d}$, and $\boldsymbol{s}_t(\boldsymbol{x}) = \nabla_{\boldsymbol{x}} u_\theta(\boldsymbol{x}, t) \cdot \mathbf{1}\{\boldsymbol{x} \in \Omega\}$, we have the following upper bound on the KL divergence between the target and the approximate distribution,*

$$\mathrm{KL}\left(\pi \| \widehat{\pi}_T\right) \lesssim (d + M_2) \cdot e^{-T} + d^2 h \left(\log K + T\right) + T\delta + \left(\delta_1 + C_1^{\mathrm{U}}(\varepsilon)\delta_2\right) \cdot \mathrm{Vol}(\Omega)$$

$$+ C_2^{\mathrm{U}} \sqrt{\sum_{k=1}^{N} h_k \max_{\boldsymbol{x} \in \Omega} \{\pi_{t_k}(\boldsymbol{x})\} \cdot \delta_2 \cdot \mathrm{Vol}(\Omega)},$$

*where $C_1^{\mathrm{U}}(\varepsilon)$ and $C_2^{\mathrm{U}}$ are defined in Theorem 6.*

*Proof of Theorem 7.* As $\boldsymbol{s}_t(\boldsymbol{x}) = \nabla_{\boldsymbol{x}} u_\theta(\boldsymbol{x}, t) \cdot \mathbf{1}\{\boldsymbol{x} \in \Omega\}$, we have

$$\sum_{k=1}^{N} h_k \mathbb{E}_{\boldsymbol{x}_{t_k} \sim \pi_{t_k}} \|\nabla_{\boldsymbol{x}} \log \pi_{t_k}(\boldsymbol{x}_{t_k}) - \boldsymbol{s}_{t_k}(\boldsymbol{x}_{t_k})\|^2$$

$$= \sum_{k=1}^{N} h_k \int_{\Omega^c} \pi_{t_k}(\boldsymbol{x}) \|\nabla_{\boldsymbol{x}} \log \pi_{t_k}(\boldsymbol{x})\|^2 \mathrm{d}\boldsymbol{x} + \sum_{k=1}^{N} h_k \int_{\Omega} \pi_{t_k}(\boldsymbol{x}) \|\nabla_{\boldsymbol{x}} e_{t_k}(\boldsymbol{x})\|^2 \mathrm{d}\boldsymbol{x} \tag{85}$$

$$\leqslant \sum_{k=1}^{N} h_k \delta + \sum_{k=1}^{N} h_k \max_{\boldsymbol{x} \in \Omega} \{\pi_{t_k}(\boldsymbol{x})\} \cdot \|\nabla_{\boldsymbol{x}} e_{t_k}(\cdot)\|_{L^2(\Omega)}^2$$

$$\leqslant T\delta + \delta_1 \cdot \mathrm{Vol}(\Omega) + C_1^{\mathrm{U}}(\varepsilon)\delta_2 \cdot \mathrm{Vol}(\Omega) + C_2^{\mathrm{U}} \sqrt{\sum_{k=1}^{N} h_k \max_{\boldsymbol{x} \in \Omega} \{\pi_{t_k}(\boldsymbol{x})\} \cdot \delta_2 \cdot \mathrm{Vol}(\Omega)},$$

where the last inequality follows from the result in Theorem 6 and

$$\sum_{k=1}^{N} h_k \max_{\boldsymbol{x} \in \Omega} \{\pi_{t_k}(\boldsymbol{x})\} \cdot \sqrt{\varepsilon L_{\mathrm{PINN}}^{\mathrm{Unif}}(t_k; C_1^{\mathrm{U}}(\varepsilon))}$$

$$\leqslant \left(\varepsilon \sum_{k=1}^{N} h_k \max_{\boldsymbol{x} \in \Omega} \{\pi_{t_k}(\boldsymbol{x})\} \cdot L_{\mathrm{PINN}}^{\mathrm{Unif}}(t_k; C_1^{\mathrm{U}}(\varepsilon))\right)^{1/2} \left(\sum_{k=1}^{N} h_k \max_{\boldsymbol{x} \in \Omega} \{\pi_{t_k}(\boldsymbol{x})\}\right)^{1/2} \tag{86}$$

$$\leqslant \sqrt{\sum_{k=1}^{N} h_k \max_{\boldsymbol{x} \in \Omega} \{\pi_{t_k}(\boldsymbol{x})\} \cdot \delta_2 \cdot \mathrm{Vol}(\Omega)}.$$

---

[3]Here, we contain the term $\mathrm{Vol}(\Omega)$ since the PINN residual objective used for uniform collocation points is given by $\|r_t(\cdot)\|_{L^2(\Omega)}^2 / \mathrm{Vol}(\Omega)$.

Table 3: Mixing proportions between 9 modes in 9-Gaussians.

| Modes | $(-5,-5)'$ | $(-5,0)'$ | $(-5,5)'$ | $(0,-5)'$ | $(0,0)'$ | $(0,5)'$ | $(5,-5)'$ | $(5,0)'$ | $(5,5)'$ |
|---|---|---|---|---|---|---|---|---|---|
| Weight | 0.2 | 0.04 | 0.2 | 0.04 | 0.04 | 0.04 | 0.2 | 0.04 | 0.2 |

Combining (85) and Proposition 1, we complete our proof. $\qquad\square$

## C  LIMITATIONS

As we use LMC for collocation generation in DPS, there is a risk of missing modes if short LMC runs do not adequately cover the high-density domain. In such cases, running LMC for an annealed path of target distributions or adopting the adversarial training method in Wang et al. (2022) for collocation points maybe helpful. Also, solving high dimensional PDEs via PINN can be challenging, and we may use techniques such as stochastic dimension gradient descent or the Hutchinson trick to scale DPS to high dimensional problems (Hu et al., 2024b;a).

## D  MORE ON RELATED WORKS

To sample from an unnormalized target distributions, vanilla methods based on ergodic sampling using Markov chain Monte Carlo (MCMC) (Kass et al., 1998; Neal, 2012) or stochastic differential equations (SDE) such as the Langevin dynamics (Roberts & Tweedie, 1996) typically have very slow convergence rates, making them inefficient in practice. In addition to those simulation-based VI approaches within the stochastic optimal control framework, Akhound-Sadegh et al. (2024) avoids the need to back-propagate through an SDE, at the price of introducing a bias into their objective function. Off-policy training has also been enabled for diffusion-based samplers where a log-variance objective function is employed instead of the KL divergence (Richter & Berner, 2024).

## E  ADDITIONAL EXPERIMENTAL DETAILS AND RESULTS

### E.1  BASELINES

We benchmark DPS performance against a wide range of strong baseline methods. For MCMC methods, we consider the Langevin Monte Carlo (LMC). For LMC, we run 100,000 iterations with step sizes 0.02, 0.002, 0.0002. Then we choose the samples with the best performance. As for sampling methods using reverse diffusion, we include RDMC (Huang et al., 2023), and SLIPS (Grenioux et al., 2024). We use the implementation of SLIPS and RDMC from Grenioux et al. (2024) and choose Geom(1, 1) as the SL scheme for SLIPS. For each algorithm, we search its hyper-parameters within a predetermined grid, similar to Grenioux et al. (2024). We also compare with VI-based PIS (Zhang & Chen, 2021) and DIS (Berner et al., 2022). We use the implementation of PIS and DIS from Berner et al. (2022). For particle-based VI method, SVGD, we use 1,000 particles in our experiments.

### E.2  TARGETS

**9-Gaussians** is a 2-dimensional Mixture of Gaussians where there are 9 modes designed to be well-separated from each other. The modes share the same variance of 0.3 and the means are located in the grid of $\{-5, 0, 5\} \times \{-5, 0, 5\}$. We set challenging mixing proportions between different modes as shown in Table 3.

**Rings** is the inverse polar reparameterization of a 2-dimensional distribution $p_z$ which has itself a decomposition into two univariate marginals $p_r$ and $p_\theta$: $p_r$ is a mixture of 4 Gaussian distributions $\mathcal{N}(i, 0.2^2)$ with $i = 2, 4, 6, 8$ describing the radial position and $p_\theta$ is a uniform distribution over $[0, 2\pi)$, which describes the angular position of the samples. We also set challenging mixing proportions between different modes of $p_r$ as shown in Table 4.

Table 4: Mixing proportions between 4 modes in rings.

| Modes | $r = 2$ | $r = 4$ | $r = 6$ | $r = 8$ |
|---|---|---|---|---|
| Weight | 0.05 | 0.45 | 0.05 | 0.45 |

**Funnel** is a classical sampling benchmark problem from Neal (2003); Hoffman et al. (2014). This 10-dimensional density is defined by

$$\mu(\boldsymbol{x}) := \mathcal{N}(x_0; 0, 9)\mathcal{N}(\boldsymbol{x}_{1:9}; \boldsymbol{0}, \exp(x_0)\boldsymbol{I}_9).$$

**Double-well** is a high-dimensional distribution which share the unnormalized density:

$$\mu(\boldsymbol{x}) := \exp\left(\sum_{i=0}^{w-1} -x_i^4 + 6x_i^2 + 0.5x_i - \sum_{i=w}^{d-1} 0.5x_i^2\right).$$

We choose $w = 3$ and $d = 30$ leading to a 30-dimensional distribution contained 8 modes with challenging mixing proportions between different modes.

### E.3 DIFFUSION-PINN SAMPLER

**Model.** The model architecture of $\text{NN}_\theta(\boldsymbol{x}, t) : \mathbb{R}^d \times [0, T] \to \mathbb{R}$ in $u_\theta(\boldsymbol{x}, t)$ is

$$\text{NN}_\theta(\boldsymbol{x}, t) = \text{MLP}^{\text{dec}}\left(\text{MLP}^{\text{embx}}(\boldsymbol{x}) + \text{MLP}^{\text{embt}}(\text{emb}(t))\right),$$

where $\text{MLP}^{\text{dec}}$ represents a decoder implemented as MLPs with layer widths $[128, 128, 128, 1]$. The component $\text{MLP}^{\text{embx}}$ serves as a data embedding block and is implemented as MLPs with layer widths $[2, 128]$. $\text{MLP}^{\text{embt}}$ functions as a time embedding block, implemented as MLPs with layer widths $[256, 128, 128]$. The input to $\text{MLP}^{\text{embt}}$ is derived from the sinusoidal positional embedding (Vaswani et al., 2017) of $t$. All these three MLPs utilize the GELU activation function.

**Training.** In our implementation, we choose $\boldsymbol{f}(\boldsymbol{x}, t) = -\frac{\boldsymbol{x}}{2(1-t)}$ and $g(t) = \sqrt{\frac{1}{1-t}}$ which lead to the following forward process

$$\mathrm{d}\boldsymbol{x}_t = -\frac{\boldsymbol{x}_t}{2(1-t)}\,\mathrm{d}t + \sqrt{\frac{1}{1-t}}\,\mathrm{d}\boldsymbol{B}_t, \quad \boldsymbol{x}_0 \sim \pi, \quad T_{\min} \leqslant t \leqslant T_{\max}. \tag{87}$$

This admits the explicit conditional distribution $\pi_{t|0}(\boldsymbol{x}_t|\boldsymbol{x}_0) = \mathcal{N}(\boldsymbol{x}_t; \sqrt{1-t} \cdot \boldsymbol{x}_0, t\boldsymbol{I}_d)$. We choose $T_{\min} = 0.001$ and $T_{\max} = 0.999$ in practice. The corresponding log-density FPE becomes

$$\partial_t u_t(\boldsymbol{x}) = \frac{1}{2(1-t)}\left[\Delta u_t(\boldsymbol{x}) + \|\nabla_{\boldsymbol{x}} u_t(\boldsymbol{x})\|^2 + \boldsymbol{x} \cdot \nabla_{\boldsymbol{x}} u_t(\boldsymbol{x}) + d\right] := \frac{1}{2(1-t)}\mathcal{L}_{\text{L-FPE}}^{\text{prac}} u_t(\boldsymbol{x}). \tag{88}$$

We choose $\beta(t) = 2(1-t)$ to make training more stable, leading the following training objective

$$
\begin{aligned}
L_{\text{train}}^{\text{prac}}(u_\theta) :=& \mathbb{E}_{t \sim \mathcal{U}[0,T]}\mathbb{E}_{\boldsymbol{x}_t \sim \nu_t}\left[\left\|2(1-t) \cdot \partial_t u_\theta(\boldsymbol{x}_t, t) - \mathcal{L}_{\text{L-FPE}}^{\text{prac}} u_\theta(\boldsymbol{x}_t, t)\right\|^2\right] \\
&+ \lambda \cdot \mathbb{E}_{\boldsymbol{z} \sim \mathcal{N}(\boldsymbol{0}, \boldsymbol{I}_d)}\left[\left\|\nabla_{\boldsymbol{x}} u_\theta(\boldsymbol{x}, t) + \boldsymbol{z}\right\|^2\right],
\end{aligned} \tag{89}
$$

where $\lambda$ is the regularization coefficient. It is enough for us to use PINN residual loss without regularization except for Funnel where the regularization is quite useful and we use $\lambda = 1$. To generate collocation points for PINN, we run a short chain of LMC with a large step size. The hyper-parameters used in LMC for different targets are reported in Table 5. We generate fresh collocation points per iteration except for Funnel where we resample new collocation points per $10,000$ iterations.

We train all models with Adam optimizer (Kingma & Ba, 2014). The hyper-parameters used in training are summarized in Table 6. We use a linear decay schedule for the learning rate in all experiments.

Table 5: Hyper-parameters used in LMC for generating collocation points.

|  | 9-Gaussians | Rings | Funnel | Double-well |
|---|---|---|---|---|
| step size | 1.0 | 0.15 | 0.02 | 0.02 |
| iterations | 60 | 100 | 10,000 | 100 |
| batch size | 128 | 200 | 200 | 700 |
| refresh samples per iteration | ✓ | ✓ | ✗ | ✓ |

Table 6: Hyper-parameters for training PINN.

|  | 9-Gaussians | Rings | Funnel | Double-well |
|---|---|---|---|---|
| learning rate | 0.0005 | 0.0005 | 0.0001 | 0.0005 |
| max norm of gradient clipping | 1.0 | 1.0 | 1000.0 | 1.0 |
| regularization coefficient $\lambda$ | 0 | 0 | 1 | 0 |
| total training iterations | 400k | 1,000k | 800k | 1,500k |

---

**Algorithm 2** : Sampling from reverse process

---

**Require:** Starting time $T_{\min}$, Terminal time $T_{\max}$, Sample size $M$, Discretization steps $N$, Bounded domain $\Omega$, Approximated log-density $u_\theta(\boldsymbol{x}, t)$ provided by PINN.

1: Compute the step size $h := (T_{\max} - T_{\min})/N$.
2: Obtain the approximated score function $\boldsymbol{s}_t(\boldsymbol{x}) := \nabla_{\boldsymbol{x}} u_\theta(\boldsymbol{x}, t) \cdot \mathbb{1}\{\boldsymbol{x} \in \Omega\}$.
3: Sample i.i.d. $\boldsymbol{x}_i^0 \sim \mathcal{N}(\boldsymbol{0}, \boldsymbol{I}_d)$, $\forall 1 \leqslant i \leqslant M$
4: **for** $n = 1, \cdots, N$ **do**
5:    Sample i.i.d. $\boldsymbol{z}_i \sim \mathcal{N}(\boldsymbol{0}, \boldsymbol{I}_d)$, $\forall 1 \leqslant i \leqslant M$.
6:    Compute $t_{n-1} := T_{\min} + (n-1)h$.
7:    Update by simulating the reverse process: $\forall 1 \leqslant i \leqslant M$

$$\boldsymbol{x}_i^n \leftarrow \sqrt{1 + \frac{h}{t_{n-1}}} \boldsymbol{x}_i^{n-1} + 2 \left( \sqrt{1 + \frac{h}{t_{n-1}}} - 1 \right) \boldsymbol{s}_{1-t_{n-1}}(\boldsymbol{x}_i^{n-1}) + \sqrt{\frac{h}{t_{n-1}}} \boldsymbol{z}_i,$$

8: **end for**
9: **return** Approximated samples $\boldsymbol{x}_1^N, \cdots, \boldsymbol{x}_M^N$.

---

**Sampling.** The corresponding reverse process is given by

$$\mathrm{d}\boldsymbol{x}_t = \left( \frac{\boldsymbol{x}_t}{2t} + \frac{\nabla_{\boldsymbol{x}} \log \pi_{1-t}(\boldsymbol{x}_t)}{t} \right) \mathrm{d}t + \sqrt{\frac{1}{t}} \mathrm{d}\boldsymbol{B}_t, \quad \boldsymbol{x}_0 \sim \pi_{T_{\max}}, \quad T_{\min} \leqslant t \leqslant T_{\max}. \quad (90)$$

To simulate (90), we approximate $\pi_{T_{\max}} \approx \mathcal{N}(\boldsymbol{0}, \boldsymbol{I}_d)$ and use the exponential integrator scheme with the score approximation $\boldsymbol{s}_t(\boldsymbol{x}) \approx \nabla_{\boldsymbol{x}} \log \pi_t(\boldsymbol{x})$. In practice, we use $\boldsymbol{s}_t(\boldsymbol{x}) := \nabla_{\boldsymbol{x}} u_\theta(\boldsymbol{x}, t) \cdot \mathbb{1}\{\boldsymbol{x} \in \Omega\}$ where $u_\theta(\boldsymbol{x}, t)$ is the approximated log-density provided by PINN which is trained by Algorithm 1 and $\Omega$ is a chosen bounded region that covers the high density domain of $\pi_t$ for any $t \in [T_{\min}, T_{\max}]$. We use $\Omega := \{\boldsymbol{x} \in \mathbb{R}^d : \|\boldsymbol{x}\| \leqslant R\}$ in all experiments, the choice of $R$ is reported in Table 7. Our sampling process is summarized in Algorithm 2. We provide more sampling performances of different methods for different targets in Figure 6 and sample trajectories from DPS in Figure 7.

Table 7: The diameter of the truncated region for different targets.

|  | 9-Gaussians | Rings | Funnel | Double-well |
|---|---|---|---|---|
| $R$ | 20 | 20 | 2000 | 30 |

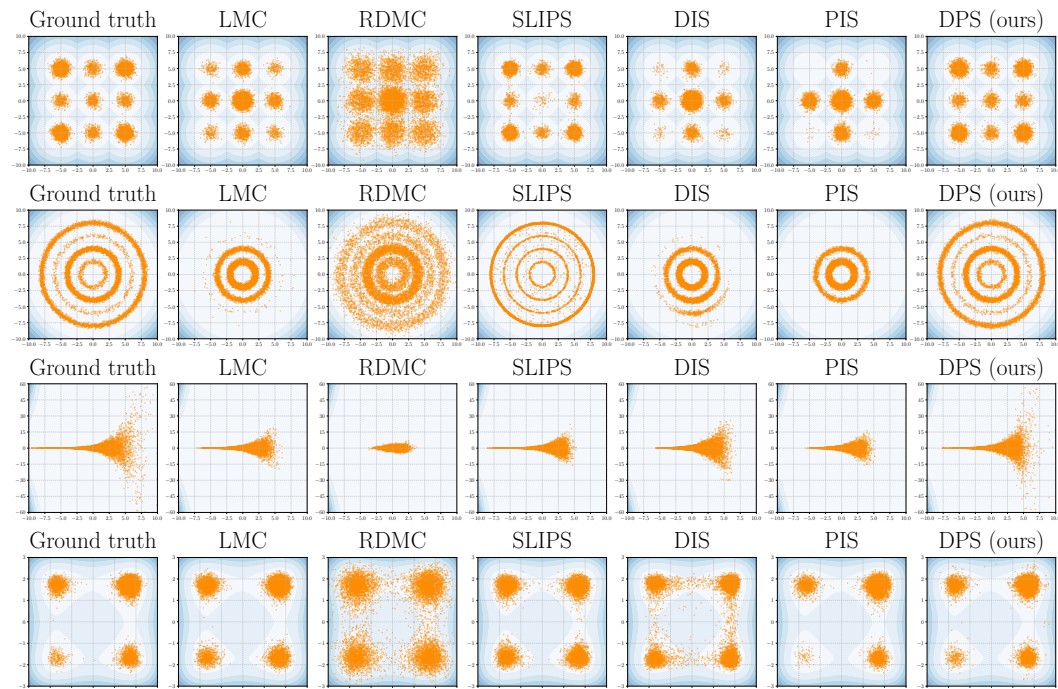

Figure 6: Sampling performance of different methods for 9-Gaussians ($d = 2$), Rings ($d = 2$), Funnel ($d = 10$) and Double-well ($d = 30$).

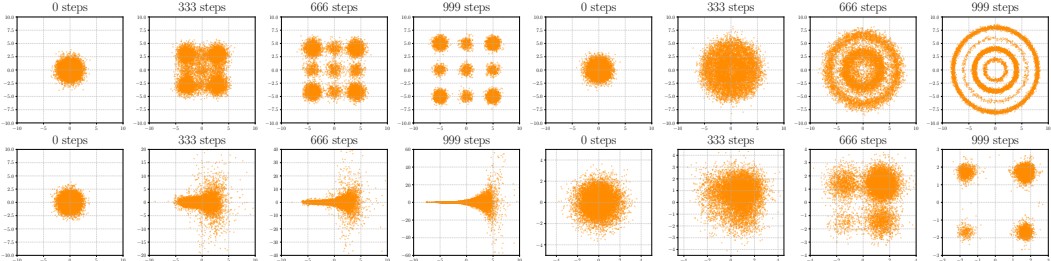

Figure 7: Sample trajectories from DPS for 9-Gaussians ($d = 2$), Rings ($d = 2$), Funnel ($d = 10$) and Double-well ($d = 30$).

### E.4 ADDITIONAL EXPERIMENTAL RESULTS

**Additional higher-dimensional experiments.** We provide a higher-dimensional experiments on 50-dimensional Double-well with 32 separated modes and challenging mixing proportions. Our results are show in Table 8.

Table 8: (Sliced) KL divergence to the ground truth and mixing proportions estimation error obtained by different methods on 50-dim Double-well.

| Metric | LMC | SLIPS | RDMC | PIS | DIS | SVGD | HMC | DPS (ours) |
|---|---|---|---|---|---|---|---|---|
| Sliced KL divergence ($\downarrow$) | $0.321_{\pm 0.009}$ | $0.745_{\pm 0.018}$ | $2.769_{\pm 0.022}$ | $0.362_{\pm 0.012}$ | $0.973_{\pm 0.017}$ | $24.349_{\pm 0.067}$ | $5.212_{\pm 0.084}$ | $\mathbf{0.101_{\pm 0.011}}$ |
| Mixing proportions estimation error ($\downarrow$) | $0.0681_{\pm 0.0053}$ | $0.1323_{\pm 0.0062}$ | $0.1641_{\pm 0.0048}$ | $0.0203_{\pm 0.0021}$ | $0.0808_{\pm 0.0051}$ | $0.1710_{\pm 0.0046}$ | $1.0503_{\pm 0.2497}$ | $\mathbf{0.0008_{\pm 0.0003}}$ |

**Sample quality measured by more evaluation metrics.** Additionally, we provide results using additional evaluation metrics in Table 9 and 10, including the Wasserstein-distance and EMC value

proposed in Blessing et al. (2024), for specific sampling tasks. These supplementary evaluations further highlight the superior sampling performance of our method, while PIS-LV and DIS-LV exhibit comparable performance, consistent with our main evaluation results using KL divergence and mixing proportions estimation error.

Table 9: $W_2$-distance ($\downarrow$) comparison for different methods on the 9-Gaussians and Rings targets.

| Target | LMC | RDMC | SLIPS | DIS | PIS | SVGD | HMC | PIS-LV | DIS-LV | DPS (ours) |
|---|---|---|---|---|---|---|---|---|---|---|
| 9-Gaussians | $5.0064_{\pm 0.076}$ | $3.1329_{\pm 0.084}$ | $0.9884_{\pm 0.143}$ | $5.4442_{\pm 0.056}$ | $5.0341_{\pm 0.052}$ | $3.9018_{\pm 0.054}$ | $7.8570_{\pm 1.664}$ | $4.5839_{\pm 0.047}$ | $1.0534_{\pm 0.214}$ | $\mathbf{0.7794_{\pm 0.127}}$ |
| Rings | $3.4096_{\pm 0.075}$ | $1.2195_{\pm 0.102}$ | $2.2280_{\pm 0.064}$ | $3.2716_{\pm 0.078}$ | $3.7429_{\pm 0.070}$ | $2.0395_{\pm 0.127}$ | $3.3051_{\pm 0.696}$ | $\mathbf{0.7146_{\pm 0.076}}$ | $0.7078_{\pm 0.118}$ | $0.7726_{\pm 0.099}$ |

Table 10: EMC value ($\uparrow$) comparison for different methods on 9-Gaussians and Rings targets.

| Method | LMC | SLIPS | RDMC | PIS | DIS | SVGD | HMC | PIS-LV | DIS-LV | DPS (ours) |
|---|---|---|---|---|---|---|---|---|---|---|
| 9-Gaussians | $0.3562_{\pm 0.005}$ | $0.9822_{\pm 0.005}$ | $0.6195_{\pm 0.014}$ | $0.3341_{\pm 0.003}$ | $0.2988_{\pm 0.003}$ | $0.5095_{\pm 0.008}$ | $0.3302_{\pm 0.064}$ | $0.3208_{\pm 0.001}$ | $\mathbf{0.9942_{\pm 0.002}}$ | $\mathbf{0.9965_{\pm 0.002}}$ |
| Rings | $0.3844_{\pm 0.007}$ | $0.7434_{\pm 0.008}$ | $0.8754_{\pm 0.009}$ | $0.3328_{\pm 0.004}$ | $0.4081_{\pm 0.007}$ | $0.7093_{\pm 0.067}$ | $0.4411_{\pm 0.142}$ | $\mathbf{0.9976_{\pm 0.002}}$ | $\mathbf{0.9983_{\pm 0.001}}$ | $0.9928_{\pm 0.002}$ |

**Complexity analysis.** We examine the impact of our proposed unbiased Hutchinson gradient estimator on training time. Our results of training time are shown in Table 11. Notably, without this estimator, training time increases significantly as the dimensionality grows. In contrast, using the proposed unbiased estimator effectively mitigates this issue.

Furthermore, We examine the impact of score computation at every time step for sampling. We present our sampling time in Table 12. Once the log-density approximation is obtained, sampling can be performed in a remarkably short time. Moreover, we compare the sampling time using direct score estimation versus taking the gradient of the approximated log-density in Table 12. Our results show that sampling time is halved when score estimation is directly employed. Nonetheless, the sampling time of our method is already highly efficient. For instance, sampling 10,000 points over 1,000 time steps for 100-dim tasks takes less than 2 seconds.

Table 11: Per iteration training time (in seconds).

| Time (s) | 10d | 20d | 30d | 40d | 50d | 60d | 70d | 80d | 90d | 100d |
|---|---|---|---|---|---|---|---|---|---|---|
| Without unbiased estimator | 0.037 | 0.062 | 0.086 | 0.111 | 0.135 | 0.160 | 0.184 | 0.210 | 0.235 | 0.260 |
| With unbiased estimator | **0.018** | **0.018** | **0.018** | **0.018** | **0.018** | **0.018** | **0.018** | **0.018** | **0.018** | **0.018** |

Table 12: Sampling time (in seconds).

| Time (s) | 10d | 20d | 30d | 40d | 50d | 60d | 70d | 80d | 90d | 100d |
|---|---|---|---|---|---|---|---|---|---|---|
| Taking gradient | 1.537 | 1.534 | 1.538 | 1.551 | 1.558 | 1.567 | 1.570 | 1.581 | 1.587 | 1.596 |
| Using score directly | 0.683 | 0.685 | 0.701 | 0.731 | 0.736 | 0.742 | 0.759 | 0.775 | 0.794 | 0.812 |

**Ablation studies on unbiased Hutchinson gradient estimator.** We compute Laplacian in the PINN loss directly in our experiments for better results. In addition, we conduct ablation experiments using unbiased Hutchinson gradient estimator in high-dimension case (30-dimensional Double-well) to demonstrate the validity of the proposed estimator. The results are shown in Table 13.

**Ablation studies on incur errors from log-density approximation.** After obtaining an accurate log-density approximation via PINN, we obtain the score approximation by taking gradient of the log-density approximation, and plug the obtained score approximation into the reverse process of diffusion models for sampling. Our theoretical results in Theorem 2 show that the approximation error of both log-density and score function can be controlled by the PINN residual loss. Numerically, for 9-Gaussian targets, our results shown in the left figure of Figure 8 support that a good score approximation can be obtained as long as we have an accurate log-density approximation, i.e., the incur errors are negligible.

**Ablation studies on target-informed parameterization.** When querying the log-density of the target is expensive, we could use a simple neural networks for parameterization and utilize the

Table 13: (Sliced) KL divergence to the ground truth and mixing proportions estimation error obtained by DPS with/without Hutchinson unbiased gradient estimator on 30-dim Double-well.

| Metric | Without unbiased estimator | With unbiased estimator |
|---|---|---|
| Sliced KL divergence ($\downarrow$) | $\mathbf{0.0273}_{\pm \mathbf{0.0113}}$ | $0.043_{\pm 0.008}$ |
| Mixing proportions estimation error ($\downarrow$) | $\mathbf{0.0004}_{\pm \mathbf{0.0002}}$ | $0.0008_{\pm 0.0005}$ |

following training objective in Algorithm 1, instead of (13),

$$L_{\text{MCMC}}^{\text{simple}}(u_\theta) := \frac{1}{M} \sum_{i=1}^{M} \beta^2(t_i) \cdot \left\| \partial_t u_\theta(\boldsymbol{x}_i^{t_i}, t_i) - \mathcal{L}_{\text{L-FPE}} u_\theta(\boldsymbol{x}_i^{t_i}, t_i) \right\|^2 + \frac{\lambda}{M} \sum_{i=1}^{M} \ell_{\text{reg}}(u_\theta; T, \boldsymbol{z}_i)$$

$$+ \frac{1}{M} \sum_{i=1}^{M} \left\| u_\theta(\boldsymbol{x}_i^0, 0) - \log \mu(\boldsymbol{x}_i^0) \right\|^2 .$$

Notably, the last term can be estimated via stochastic estimation in practice. Numerically, we conduct an ablation study on comparison between the two methods (using target-informed parameterization versus using simple parameterization with the above modified objective) for 9-Gaussians task. Our results are shown in the right figure of Figure 8. We can easily find that both methods are valid to obtain an accurate score approximation (thus perfect sampling).

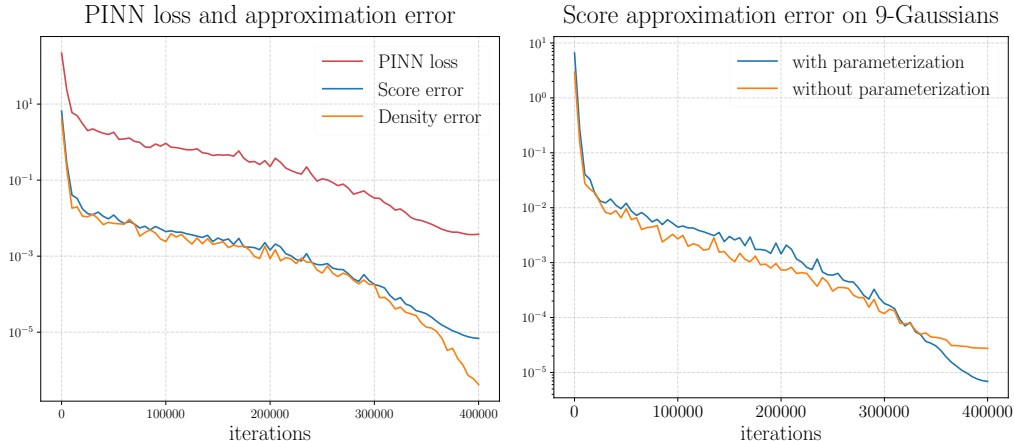

Figure 8: **Left**: PINN loss and (log-density and score) approximation error on 9-Gaussians for our method. **Right**: Comparison of score approximation error with and without parameterization based on the initial log density.

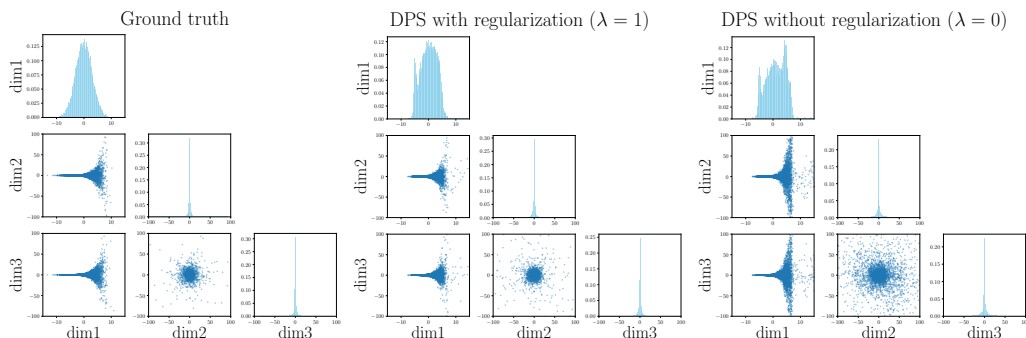

Figure 9: Sampling performance of DPS with/without regularization for Funnel.

