# OpenReview forum: "Diffusion-PINN Sampler"
_ICLR.cc/2025/Conference — Submitted to ICLR 2025_

### Official Review · Reviewer_NDLS · 2024-10-22

**Soundness:** 3
**Presentation:** 3
**Contribution:** 2
**Rating:** 6
**Confidence:** 4

**Summary:**

This paper introduces a novel diffusion-based sampling algorithm called the Diffusion-PINN Sampler (DPS). The DPS estimates the drift term in the reverse diffusion process by solving the governing partial differential equation of the log-density of the underlying stochastic differential equation (SDE) marginals using physics-informed neural networks (PINN). The authors prove that the error of log-density approximation can be controlled by the PINN residual loss, which allows them to establish convergence guarantees for the DPS. Experiments on sampling tasks demonstrate the effectiveness of this method.

**Strengths:**

1. The paper is well-composed and the concepts are clearly articulated, making it easy to comprehend.
2. The innovative approach of learning a diffused log-density via the log-density-PF equation is novel.
3. The math derivation is rigorous.

**Weaknesses:**

1. When the score of the target distribution is accessible, we can perform sampling using MCMC[1] and ParVI[2][3][4]. Renowned algorithms such as HMC and SVGD should be considered for comparison. It would be interesting to see if PINN-Sampling can surpass these methods.

2. The training objective in equation 14 requires gradients of the neural network. This could potentially be time-intensive as the dimension increases. I am skeptical about the scalability of this method. Could you provide a complexity analysis of the training process in relation to dimensionality?

3. Assumption 1 is not plausible, as the suggested scenario is unlikely to occur. Consequently, it's not feasible to bound the error within a constrained domain. It's probable that the subsequent theorems will also hold when $\Omega = \mathbb{R}^d$ and $\nu_t$ are light-tailed distributions.

4. In Example 1, as $\tau$ approaches zero, the variance of $\pi^M$ and $\hat{\pi}^M$ becomes unbounded. However, in real-world applications, the variance of our data distribution is always bounded. Could you give an example where the data distribution has a bounded variance but still has an arbitrarily small $\tau$?

5. The scope of the experiments appears to be rather limited, being mostly synthetic and low-dimensional. It would be beneficial to see applications to higher-dimensional cases, Gaussian processes, Bayesian Neural Networks, and real-world data experiments.

References:

[1] Neal R M. MCMC using Hamiltonian dynamics[J]. arXiv preprint arXiv:1206.1901, 2012.

[2] Liu Q, Wang D. Stein variational gradient descent: A general-purpose Bayesian inference algorithm[J]. Advances in neural information processing systems, 2016, 29.

[3] Liu C, Zhuo J, Cheng P, et al. Understanding and accelerating particle-based variational inference[C]//International Conference on Machine Learning. PMLR, 2019: 4082-4092.

[4] Wang F, Zhu H, Zhang C, et al. GAD-PVI: A General Accelerated Dynamic-Weight Particle-Based Variational Inference Framework[C]//Proceedings of the AAAI Conference on Artificial Intelligence. 2024, 38(14): 15466-15473.

**Questions:**

see Weaknesses.

---

> ### Author Response · Authors · 2024-11-24
> **Rebuttal**
>
> Thank you for your careful review and helpful feedback! We address your concerns as follows.
>
> ---
>
> 1. When the score of the target distribution is accessible, we can perform sampling using MCMC[1] and ParVI.  ...
>
>
> Additional experimental results for more baselines, including HMC and SVGD, are provided in the global response.
>
> ---
>
> 2. The training objective in equation 14 requires gradients of the neural network. This could potentially be time-intensive as the dimension increases. ...
>
>
> We propose an unbiased Hutchinson gradient estimator to reduce the computational cost of evaluating the Laplacian in the training objective. This approach effectively addresses the issue of time-intensive computations that arise as the dimensionality increases, which would otherwise hinder the scalability of our method, as demonstrated in the numerical complexity analysis presented in the global response.
>
> ---
>
> 3. Assumption 1 is not plausible, as the suggested scenario is unlikely to occur. ...
>
> First, making or using this assumption is common in the analysis of PINN, as seen in [1] and [2].  Second, this assumption is essential for two reasons: (1) it ensures the uniqueness of the solution within the bounded region, and (2) it guarantees the existence of the continuous constants required for our analysis. Moreover, it is still worthwhile to explore the possibility of extending our analysis to the domain $\Omega = \mathbb{R}^d$ and certain light-tailed distributions $\nu_t$. Specifically, we could construct a continuous modifier function $\delta_\theta^*(\boldsymbol{x}, t)$ such that $\delta_\theta^*(\boldsymbol{x}, t) = u_t^*(\boldsymbol{x}) - u_\theta(\boldsymbol{x}, t)$ for any $(\boldsymbol{x}, t) \in \partial \Omega \times [0, T]$. This would allow us to extend our analysis by leveraging the light-tailed property of $\nu_t$ and the construction of $\delta_\theta^*(\boldsymbol{x}, t)$.
>
> ---
>
> 4. In Example 1, as $\tau$ approaches zero, the variance of $\pi^M$ and $\hat{\pi}^M$ becomes unbounded. However, in real-world applications, the variance of our data distribution is always bounded. ...
>
>
> We provide an example that exhibits the same properties as in Example 1, with bounded variance. Specifically, we consider the mixture distributions:
> $$
> \pi^M = w_1 \mathcal{N}(\boldsymbol{a}_1, \sigma^2 \boldsymbol{I}_d) + w_2 \mathcal{N}(\boldsymbol{a}_2, \sigma^2 \boldsymbol{I}_d)\quad {\rm and}\quad \hat{\pi}^M = \hat{w}_1 \mathcal{N}(\boldsymbol{a}_1, \sigma^2 \boldsymbol{I}_d) + \hat{w}_2 \mathcal{N}(\boldsymbol{a}_2, \sigma^2 \boldsymbol{I}_d),
> $$
> then we obtain the following lower bound for the KL divergence:
> $$
> \operatorname{KL}\left(\pi^M | \hat{\pi}^M\right) \geqslant w_1 \left(\log w_1 - \log\left(\hat{w}_1 + \exp\left(-\frac{\\|\boldsymbol{a}_1 - \boldsymbol{a}_2\\|^2}{4\sigma^2}\right)\right)\right) + w_2 \left(\log w_2 - \log\left(\hat{w}_2 + \exp\left(-\frac{\\|\boldsymbol{a}_1 - \boldsymbol{a}_2\\|^2}{4\sigma^2}\right)\right)\right) - \left(\log 4 + d \right)\exp\left(\frac{d}{2} \log 2 - \frac{\\|\boldsymbol{a}_1 - \boldsymbol{a}_2\\|^2}{64\sigma^2}\right).
> $$
>
> Additionally, we provide an upper bound for the Fisher divergence:
> $$
> F(\pi^M, \hat{\pi}^M) \leqslant 2 \exp\left(-\frac{\\|\boldsymbol{a}_1 - \boldsymbol{a}_2\\|^2}{2\sigma^2}\right) \left[\frac{w_2^2}{w_1^2} + \frac{\hat{w}_2^2}{\hat{w}_1^2} + \frac{w_1^2}{w_2^2}+ \frac{\hat{w}_1^2}{\hat{w}_2^2}\right] \frac{\\|\boldsymbol{a}_1 - \boldsymbol{a}_2\\|^2}{\sigma^4} + \frac{8 \left[\\|\boldsymbol{a}_1\\|^2 + \\|\boldsymbol{a}_2\\|^2\right]}{\sigma^4} \exp\left(\frac{d}{2} \log 2 - \frac{\\|\boldsymbol{a}_1 - \boldsymbol{a}_2\\|^2}{64\sigma^2}\right).
> $$
>
> Finally, for given $\boldsymbol{a}_1, \boldsymbol{a}_2 \in \mathbb{R}^d$ and as $\sigma^2 \to 0$, we observe that $\tau$ approaches zero in Example 1, while the variance of both $\pi^M$ and $\hat{\pi}^M$ remains bounded.
>
> ---
>
> 5. The scope of the experiments appears to be rather limited, being mostly synthetic and low-dimensional. ...
>
> We present additional high-dimensional experiments on a 50-dimensional Double-Well with 32 separated modes, incorporating challenging weights in the global response.
>
> ---
>
> References
>
> [1] Deveney T, Stanczuk J, Kreusser L M, et al. Closing the ODE-SDE gap in score-based diffusion models through the Fokker-Planck equation[J]. arXiv preprint arXiv:2311.15996, 2023.
>
> [2] Wang C, Li S, He D, et al. Is $ L^ 2$ Physics Informed Loss Always Suitable for Training Physics Informed Neural Network?[J]. Advances in Neural Information Processing Systems, 2022, 35: 8278-8290.

---

> ### Comment · Reviewer_NDLS · 2024-11-25
>
> Thanks for your clarification!
>
> Do you plan to give a revised manuscript?

---

> > ### Author Response · Authors · 2024-11-25
> > **Thanks for your response**
> >
> > We are revising our paper and plan to provide an updated version as soon as possible.

---

> ### Comment · Reviewer_NDLS · 2024-11-27
>
> Thanks for your reply.
> I also have a further concern about the newly provided experiments.
>
> 1. Your algorithm can achieve a 90\% distance reduction on your provided experiments compared to HMC or SVGD. How many particles you use for SVGD, this is a key hyperparameter.
>
> 2. Why does HMC perform worse than LMC? From my knowledge, HMC is a momentum-augmented accelerated version of LMC; their convergence behavior after enough iterations should be similar.
>
> 3. These MCMC or ParVI algorithms need mix time, I want to see the convergence curve and detailed hyperparameters setting of these baselines in the revised paper, or I cannot be convinced. Or can you provide your code with scripts?
>
> 4. I also want to know the visual time-costs comparison, how much time do you need to train on the 9-gaussian level task?

---

> ### Author Response · Authors · 2024-12-02
>
> Thanks for your response! We have addressed your concerns as follows.
>
> ---
>
> 1. How many particles used for SVGD ...
>
> The 9-Gaussians, Rings, and Double-Well targets in our experiments are all multi-modal with challenging (unequal) mixing proportions, making them particularly difficult for traditional particle-based variational inference methods like SVGD to sample from. Additionally, the time and memory costs for SVGD increase significantly with the number of particles. Therefore, we chose to use 1,000 particles for SVGD in our experiments, a standard and practical setting for this method. This hyper-parameter choice has been explicitly included in our revised paper.
>
> ---
>
> 2. HMC ...
>
> First, we want to clarify that we utilize a parallel version of LMC in our paper. Specifically, we run multiple independent LMC chains, one for each initial sample. For example, to generate 10,000 samples from the target distribution, we begin with 10,000 initial points (e.g., sampled from a standard Gaussian) and run Langevin dynamics independently for these 10,000 points over several iterations (e.g., 100,000). The final iteration outputs of the LMC chains are then collected as samples from the target distribution. This parallel approach leverages the diversity of initializations and independent chain updates to produce significantly higher-quality samples than standard LMC.
>
> Previously, we used a non-parallel (standard) version of HMC as an additional baseline, which resulted in poorer performance compared to parallel LMC. We have updated the HMC implementation to a parallel version and fine-tuned the hyper-parameters (step size = 0.1, number of Leapfrog iterations = 10, total iterations = 10,000), achieving results that are comparable to or better than those of LMC. Additionally, we have clarified our use of the parallel version of LMC in the revision to avoid any misunderstanding.
>
> ---
>
> 3. convergence curve ...
>
> The convergence curves for well-trained DPS, SVGD, and HMC for 9-Gaussians are presented in the table below. Notably, leveraging the reverse process of diffusion models, our sampling algorithm achieves significantly faster convergence compared to MCMC and ParVI methods. In contrast, the MCMC method (HMC) converges to the target distribution at an extremely slow rate.
>
> Table: Convergence curves of well-trained DPS, SVGD, and HMC for 9-Gaussians (KL divergence vs. number of steps).
> | Steps                | 111                 | 222                 | 333                 | 444                 | 555                 | 666                 | 777                 | 888                 | 999                 |
> | -------------------- | ------------------- | ------------------- | ------------------- | ------------------- | ------------------- | ------------------- | ------------------- | ------------------- | ------------------- |
> | **SVGD**             | $1.166_{\pm 0.041}$ | $1.056_{\pm 0.034}$ | $1.044_{\pm 0.049}$ | $1.011_{\pm 0.03}$  | $1.004_{\pm 0.038}$ | $0.986_{\pm 0.038}$ | $0.972_{\pm 0.031}$ | $0.973_{\pm 0.03}$  | $0.978_{\pm 0.041}$ |
> | **HMC**              | $5.805_{\pm 0.096}$ | $5.74_{\pm 0.095}$  | $5.708_{\pm 0.1}$   | $5.676_{\pm 0.068}$ | $5.577_{\pm 0.106}$ | $5.539_{\pm 0.109}$ | $5.542_{\pm 0.09}$  | $5.499_{\pm 0.098}$ | $5.419_{\pm 0.097}$ |
> | **Well-trained DPS (ours)** | $5.289_{\pm 0.064}$ | $4.668_{\pm 0.06}$  | $3.896_{\pm 0.044}$ | $3.083_{\pm 0.043}$ | $2.255_{\pm 0.041}$ | $1.442_{\pm 0.046}$ | $0.725_{\pm 0.034}$ | $0.245_{\pm 0.022}$ | $0.027_{\pm 0.012}$ |

---

> ### Author Response · Authors · 2024-12-02
>
> ---
>
> 4. time-costs ...
>
> The following table compares the training time (in iterations) and sampling accuracy (measured by KL divergence) achieved by our DPS. On the 9-Gaussians experiment, DPS requires only 20,000 training iterations—approximately 12 minutes—to achieve nearly perfect sampling, demonstrating its high efficiency.
>
> Table: Training time (iterations) vs. sampling accuracy (KL divergence) for DPS on 9-Gaussians.
> | Iteration     | 0                   | 1000                | 2000                | 3000                | 4000                | 5000                | 6000                | 7000                | 8000                | 9000                | 10000              |
> | ------------- | ------------------- | ------------------- | ------------------- | ------------------- | ------------------- | ------------------- | ------------------- | ------------------- | ------------------- | ------------------- | ------------------ |
> | **Time**      | 0s                  | 36s                 | 72s                 | 108s                | 144s                | 180s                | 216s                | 252s                | 288s                | 324s                | 360s               |
> | **KL**        | $5.323_{\pm 0.019}$ | $0.485_{\pm 0.011}$ | $0.112_{\pm 0.009}$ | $0.055_{\pm 0.011}$ | $0.018_{\pm 0.007}$ | $0.017_{\pm 0.013}$ | $0.011_{\pm 0.011}$ | $0.009_{\pm 0.01}$  | $0.006_{\pm 0.011}$ | $0.007_{\pm 0.01}$  | $0.006_{\pm 0.01}$ |
> | **Iteration** | **11000**           | **12000**           | **13000**           | **14000**           | **15000**           | **16000**           | **17000**           | **18000**           | **19000**           | **20000**           |                    |
> | **Time**      | 396s                | 432s                | 468s                | 504s                | 540s                | 576s                | 612s                | 648s                | 684s                | 720s                |                    |
> | **KL**        | $0.01_{\pm 0.009}$  | $0.009_{\pm 0.011}$ | $0.009_{\pm 0.009}$ | $0.011_{\pm 0.01}$  | $0.005_{\pm 0.007}$ | $0.011_{\pm 0.01}$  | $0.008_{\pm 0.008}$ | $0.019_{\pm 0.012}$ | $0.012_{\pm 0.011}$ | $0.012_{\pm 0.011}$ |                    |

---

### Official Review · Reviewer_X3WA · 2024-11-02

**Soundness:** 3
**Presentation:** 2
**Contribution:** 2
**Rating:** 3
**Confidence:** 4

**Summary:**

The paper studies sampling via time-reversing a diffusion process that starts in the target distribution, i.e. it considers diffusion-based generative modeling, where, however, no data samples from the target distribution are available, but the (unnormalized) target density can be evaluated. The authors suggest to learn the score function that is necessary to sample accurately by approximating the underlying Fokker-Planck PDE (in log-space) with Physics-informed neural networks (PINNs). In particular, it is argued that solving the Fokker-Planck PDE and taking derivatives afterwards is advantageous over solving a corresponding PDE for the score directly. The paper presents theoretical analyses that bound the error of the approximations of the log-density and the score, respectively, by the (weighted) PINN objective. Leveraging previous work, this in consequence allows to bound the sampling error by the PINN objective. Some numerical evaluation on synthetic sampling problems in moderate dimensions is provided, demonstrating a proof of concept of the method.

**Strengths:**

Sampling via learned diffusion processes is an interesting and active field of research, which is arguably more challenging than the typical generative modeling task since no samples from the target are available. The connection to the underlying PDEs is interesting, however, not novel (see weaknesses). Still, the presentation is sound and offers a detailed and interesting theoretical analysis.

The paper is mathematically well written, however, some (practical) implications and motivations could be made clearer for the convenience of the reader.

**Weaknesses:**

In my opinion, the paper has the following main weaknesses.

1. **Novelty.** The idea of employing PINNs for diffusion-based sampling has already been suggested in [1], [2], [3] and [4]. While [1] and [2] have been cited, only a preliminary version of [3] seems to be mentioned. In fact, in [3] (almost) the same algorithm has been presented (without relying on sampling from an MCMC algorithm) and multiple numerical experiments (including also variants of the suggested method, such as e.g. deterministic evolutions, general evolutions, annealed/prescribed evolutions) have been conducted.

2. **Numerical evaluation.** While the numerical evaluation in the paper showcases good performance of the suggested algorithm, the considered examples are rather synthetic and of moderate dimension. The question of how the algorithm scales to real-world problems and higher-dimensional examples (say, $d > 100$, which might be challenging for PINNs due to computation of higher-order derivatives) remains open.

3. **Practical relevance of theoretical results.** While the theoretical analyses look sound and are nice from a mathematical perspective, I am not certain about their practical relevance. It is clear that in principle a zero PINN loss implies zero score error and (up to discretization and up to the prior error) perfect sampling. At the same time, the provided bound seems to not allow for quantitative statements since constants cannot be computed and it is (due to the Grönwall inequality) probably far from being tight. Furthermore, it seems that the theoretical results heavily rely on earlier works by [5] and [6].

Some additional comments are the following:

4. It is unclear to me how generating collocation points with MCMC-like algorithms (e.g. LMC) is sufficient. It is known that those algorithms converge exponentially slowly for multimodal targets where the modes are sufficiently separated. Therefore, some exploration strategy seems to be necessary in order to get collocation points that cover regions containing distant modes.
5. You write that trajectory-based alternatives lead to "computational complexity associated with differentiating through SDE solvers". However, this is not necessarily true since off-policy training can be chosen also for trajectory-based methods, e.g. by using divergences other than the KL-divergence, see, e.g., [7], [8].
6. The related work section barely lists any of the many alternative methods for diffusion-based sampling that have been suggested in the last 1-2 years.
7. Concurrently to the work of [9], [1] has also derived the PDE for the log-density (and in fact, also approached it via PINNs, see above).
8. You write about "a linear interpolation (i.e., annealing) path between the target distribution and a simple prior", however it’s linear only in log-space.
9. It would be helpful to define the Fisher divergence in the main text (and not only in Appendix A.2).
10. Sampling quality can usually not be measured well by only one metric and therefore it would be even better to have more metrics (e.g. Wasserstein, ELBO, ESS, comparison to expectation reference values etc.).

I apologize in case I misunderstood certain aspects of your paper and am looking forward to potential corrections.

[1] Julius Berner, Lorenz Richter, and Karen Ullrich. An optimal control perspective on diffusion-based generative modeling. Transactions on Machine Learning Research, 2024.

[2] Bálint Máté and François Fleuret. Learning interpolations between Boltzmann densities. Transactions on Machine Learning Research, 2023.

[3] Sun, Jingtong, et al. "Dynamical measure transport and neural PDE solvers for sampling." arXiv preprint arXiv:2407.07873 (2024).

[4] Albergo, Michael S., and Eric Vanden-Eijnden. "NETS: A Non-Equilibrium Transport Sampler." arXiv preprint arXiv:2410.02711 (2024).

[5] Sitan Chen, Sinho Chewi, Jerry Li, Yuanzhi Li, Adil Salim, and Anru Zhang. Sampling is as easy as learning the score: theory for diffusion models with minimal data assumptions. In The Eleventh International Conference on Learning Representations, 2023b.

[6] Teo Deveney, Jan Stanczuk, Lisa Maria Kreusser, Chris Budd, and Carola-Bibiane Schönlieb. Closing the ode-sde gap in score-based diffusion models through the fokker-planck equation. arXiv preprint arXiv:2311.15996, 2023.

[7] Lorenz Richter and Julius Berner. Improved sampling via learned diffusions. In International Conference on Learning Representations, 2024.

[8] Sendera, Marcin, et al. "On diffusion models for amortized inference: Benchmarking and improving stochastic control and sampling." arXiv preprint arXiv:2402.05098 (2024).

[9] Chieh-Hsin Lai, Yuhta Takida, Naoki Murata, Toshimitsu Uesaka, Yuki Mitsufuji, and Stefano Ermon. Fp-diffusion: Improving score-based diffusion models by enforcing the underlying score fokker-planck equation. In International Conference on Machine Learning, pp. 18365–18398. PMLR, 2023.

**Questions:**

1. If I understand it correctly, you claim that Example 1 shows that the Fisher divergence can be arbitrarily small, while the KL divergence stays large at the same time. However, clearly, if the Fisher divergence goes to zero, so does the KL divergence (by the definition of a divergence). Therefore, your statement seems to mean that "the KL divergence can be (much) larger than the Fisher divergence", is this correct? This implies the following questions:
    - a) Since the "scale" of any divergence is rather arbitrary, why should the scale of the KL divergence be better suited than the scale of the Fisher divergence? (That said, I think it could be helpful to consider log-scales in Figure 1.)
    - b) Does your observation generalize to distributions that are not Gaussian?
    - c) Are you aware of any bounds that directly relate the KL divergence to the Fisher divergence? (I’m thinking of something similar to the Poincare inequality.)
2. I find the numerical experiments in Figure 4 more enlightening than the statement in Example 1. Do those observations carry over to other (more complicated, high-dimensional) examples? It seems to me that Lai et al. (2023) have found that using the score FPE (as a regularizer) seems to work well. Can you comment on this finding, given your argument against the score FPE? Also, what happens if you use more gradient steps in the example from Figure 4 (both losses seem to not have converged yet)? Does the score FPE eventually converge sufficiently? (Actually, how do you define "score error"? Is it some kind of $L^2$ error?)
3. If one approximates the log-density as you do, one later must take gradients to get a score approximation, which might incur errors. Can you comment on potential numerical implications?
4. Did you use Hutchinson’s gradient estimator that you introduce in lines 262 and following? While it removes a derivative, it is known to increase the variance of gradient estimators.
5. In (20) one considers $L_\mathrm{PINN}(t_k; C_1(\varepsilon))$ for multiple $t_k$, however, the integral in $L_\mathrm{PINN}$ always starts at $t=0$. Does this mean that the larger $N$ the larger is the sum? (I would not have expected the term to increase with finer time discretization.)

Some comments on notation and typos:
1. The notation seems to be not consistent all the time, e.g. regarding the time index: $g(t), f(x,t)$, but $p_t(x)$ and not $p(x, t)$; also: $u_t(x)$ vs. $u_\theta(x, t)$.
2. Why do you write $\nabla_x$, but not $\Delta_x$ or $\nabla_x \cdot f$? Also, you write $\nabla_x u_\theta(z, T)$ (e.g. in l. 254). Shouldn’t it be $\nabla_z u_\theta(z, T)$? (Alternatively, you should explain your notation. Suggestion: Writing $\nabla$ instead of $\nabla_x$ solves the issue.)
3. Some citations are not correctly formatted (e.g. "monte carlo" instead of "Monte Carlo"), some versions are old.
4. Inconsistent writing: Fokker-Planck vs. Fokker Planck.
5. 83: Definition of norm: "+" is missing.
6. 83: "Let ν denotes".
7. 199: "until the every end".
8. 320: Assumption -> Assumptions (this typo appear multiple times throughout the paper).
9. 326: After (17) replace "." with ",".
10. 333: (Arjovsky et al. (2017)).
11. 355: "Let $\hat{\pi}_T$ denotes".
12. In the appendix, there are often commas or periods missing after equations, see, e.g. (22), (23) etc.
13. 750: "Let $p(x)$ denotes".

---

> ### Author Response · Authors · 2024-11-24
> **Rebuttal**
>
> Thank you for your thoughtful review and helpful feedback! We address your concerns as follows.
>
> ---
>
> 1. Novelty ...
>
> The application of PINN to diffusion-based sampling has gained significant attention in recent research. However, it is important to highlight the unique contributions of our work compared to related studies. First, we utilize LMC for generating collocation points, which proves to be more efficient, particularly in high-dimensional settings, compared to uniform sampling. This claim is supported by our theoretical results in Section 5.3. Additionally, our work identifies the limitations of using PINN to solve the score Fokker–Planck equation (FPE) for mixing proportion identification in sampling tasks. Furthermore, we provide a comprehensive convergence analysis for our proposed algorithm. These points collectively demonstrate the novelty of our approach. A more detailed discussion of related work will be included in our revision.
>
> ---
>
> 2. Numerical Evaluation ...
>
> First, we extend our experiments on the Double-Well target to 50 dimensions, consistent with the setup in [3]. The results are shown in the global response. Notably, our target distribution is characterized by challenging mixing proportions, making it particularly difficult to sample from. While we recognize that solving higher-dimensional PDEs using PINN can be challenging, recent works have introduced techniques to scale PINN to higher-dimensional scenarios. Incorporating some of these methods could help address this limitation in our approach.
>
> ---
>
> 3. Practical relevance of theoretical results ...
>
> It is intuitively clear that achieving zero PINN loss would lead to zero score error and perfect sampling via diffusion models. However, in practice, attaining zero PINN loss is impossible. Our theoretical results aim to provide a quantitative error analysis for PINN and a convergence guarantee for our sampling algorithm. Specifically, we bound the score error in terms of the PINN residual loss and upper-bound the sampling error in KL divergence by the PINN loss.
>
> While the constants in our analysis cannot be explicitly computed, the convergence results remain valid. Moreover, some constants can be controlled through techniques such as weight clipping.
>
> We agree that the analysis presented in this work may not be very tight. Improving the analysis and tightening the bounds is an avenue for future exploration.
>
> Additionally, while we derive our results using techniques from [6] and theoretical results on diffusion models, it is important to highlight that we extend the analysis in [6]. Specifically, we consider the distribution of collocation points $\nu_t$, whereas [6] only addresses uniform sampling which would become inefficient in higher dimensions.
>
> ---
>
> 4. It is unclear to me how generating collocation points with MCMC-like algorithms (e.g. LMC) is sufficient. ...
>
> In general, obtaining perfect samples using MCMC methods is challenging. However, for our purposes, we only require samples that adequately cover the high-density domain, which is more feasible. In this context, MCMC methods are sufficient. Specifically, in practice, a short LMC chain with a large learning rate can generate samples that effectively cover the high-density domain, even though it may struggle to accurately capture the correct mixing proportions for multimodal distributions. Nonetheless, these `rough' samples are sufficient for training within the PINN framework for accurate log-potential estimation, as demonstrated by our theoretical analysis and numerical experiments (our method can capture the correct mixing proportions).
>
> ---
>
> 5. You write that trajectory-based alternatives lead to "computational complexity associated with differentiating through SDE solvers" ...
>
> We acknowledge that certain modifications can address this drawback. However, it is worth noting that these techniques have only emerged recently. We will revise our discussion accordingly in the next version.
>
> ---
>
> 6. The related work section barely lists any of the many alternative methods for diffusion-based sampling that have been suggested in the last 1-2 years.
>
> The related works section primarily summarizes and compares recent diffusion-based sampling techniques via solving PDEs. Other diffusion-based methods, such as RDMC, PIS, DIS, and DDS, are discussed in the introduction. We will revise our discussion of related works on diffusion-based sampling in the next version and sincerely appreciate the valuable comments and suggestions regarding additional references.

---

> > ### Author Response · Authors · 2024-11-24
> > **Rebuttal**
> >
> > ---
> >
> > 7. Concurrently to the work of [9], [1] has also derived the PDE for the log-density ...
> >
> > We do not consider the PDE for the log-density to be a novel contribution of this work. Notably, [9] employs this PDE and the corresponding score FPE to regularize diffusion models in the context of generative modeling, whereas we utilize it specifically for posterior sampling problem. The differences between our approach and [1] have already been discussed in the related works section. Specially, it is worth emphasizing that similar method in [1] (in the appendix of their paper) lacks a comprehensive and detailed investigation, both theoretically and numerically.
> >
> > ---
> >
> > 8. You write about "a linear interpolation (i.e., annealing) path between the target distribution and a simple prior", however it’s linear only in log-space.
> >
> > Thank you for your clarification—it is linear only in the log-space.
> >
> > ---
> >
> > 9. It would be helpful to define the Fisher divergence in the main text (and not only in Appendix A.2).
> >
> > Thank you for your suggestion! We will incorporate it into our revision.
> >
> > ---
> >
> > 10. Sampling quality can usually not be measured well by only one metric and therefore it would be even better to have more metrics
> >
> > Thank you for your suggestions! In our experiments, we focus on target distributions with challenging mixing proportions (Mixtures of Gaussians, Rings, and Double-Well) to highlight our method’s ability to correctly identify these proportions, in comparison to other baseline methods. From this perspective, we also use the $L^2$ error in mixing proportion identification, in addition to KL divergence, to evaluate the samples. Furthermore, we include additional evaluation results using Wasserstein distance and EMC value in the global rebuttal. Results using other evaluation metric also demonstrate the advantages of our method over baseline approaches.

---

> ### Author Response · Authors · 2024-11-24
> **Rebuttal**
>
> ---
>
> 11. If I understand it correctly, you claim that Example 1 shows that the Fisher divergence can be arbitrarily small, while the KL divergence stays large at the same time. ...
>
> Our statement demonstrates that the KL divergence can be (much) larger than the Fisher divergence. To build intuition, consider the mixture model
> $$
> \pi(x) = w_1 \pi_1(x) + w_2 \pi_2(x),
> $$
>
> where $\pi_1(x)$ and $\pi_2(x)$ are two distributions with well-separated high-density regions, and $w_1, w_2 > 0$ such that $w_1 + w_2 = 1$. The score function of this mixture model is given by
> $$
> \nabla_x \log \pi(x) = \frac{w_1 \nabla_x \pi_1(x) + w_2 \nabla_x \pi_2(x)}{w_1 \pi_1(x) + w_2 \pi_2(x)}.
> $$
>
> Let $\Omega_1$ and $\Omega_2$ denote the high-density regions of $\pi_1(x)$ and $\pi_2(x)$, respectively, which are assumed to be well-separated. Suppose $\pi_1(x)$ and $\pi_2(x)$ are light-tailed distributions (e.g., Gaussian) and satisfy
> $$
> \pi _1(x) \approx 0, \quad \nabla _x \pi _1(x) \approx 0 \quad \text{for } x \in \Omega_2, \quad \text{and} \quad
> \pi _2(x) \approx 0, \quad \nabla _x \pi _2(x) \approx 0 \quad \text{for } x \in \Omega_1.
> $$
>
> The high-density region of the mixture model $\pi(x)$ is thus $\Omega_1 \cup \Omega_2$. Within this domain, for $x \in \Omega_1$, we approximately have
> $$
> \nabla_x \log \pi(x) \approx \nabla_x \log \pi_1(x),
> $$
>
> and for $x \in \Omega_2$,
> $$
> \nabla_x \log \pi(x) \approx \nabla_x \log \pi_2(x).
> $$
>
> Consequently, the score function $\nabla_x \log \pi(x)$ provides little information about the weights $w_1$ and $w_2$ on the high-density regions of $\pi(x)$, leading to a nearly zero Fisher divergence. In contrast, the KL divergence between two mixture models with different weights $w_1$ and $w_2$ remains a positive constant. This observation highlights a fundamental difference between the two divergence measures and suggests that it is general for mixture models with light-tailed distributions, including Gaussians.
> We want to emphasize here that it is not the scale of different divergence that matters here. It is all about the power of distinguishing the weights of isolated components for different divergence, which in turn affect the magnitude of the corresponding signals when used for optimization. This is known as blindness of score matching in the literature ([i], [ii]).
>
> In Figure 1, the Fisher divergence is observed to be very small, approaching zero in our numerical implementation. Hence, changing the scale should not affect this result.
>
> Additionally, recall that if a distribution $\mu$ satisfies the logarithmic Sobolev inequality (LSI) with constant $C_{\rm LSI}$, we have
> $$
> {\rm KL}(\nu \| \mu) \leqslant C_{\rm LSI} \cdot {\rm FI}(\nu \| \mu),
> $$
>
> for any distribution $\nu$, where the Fisher divergence is defined as
> $$
> {\rm FI}(\nu \| \mu)=\mathbb{E} _{x \sim \nu} \left[\\|\nabla _x \log \nu(x) - \nabla _x \log \mu(x)\\| _2^2\right].
> $$
>
> Therefore, our observation aligns with the fact that the LSI constant $C_{\rm LSI}$ for the mixture model can be large (even unbounded) when the high-density regions $\Omega_1$ and $\Omega_2$ are well-separated.
>
> [i] L. K. Wenliang. Blindness of score-based methods to isolated components and mixing proportions.
> arXiv preprint arXiv:2008.10087, 2020.
>
> [ii] Mingtian Zhang, Oscar Key, Peter Hayes, David Barber, Brooks Paige, and Francois-Xavier Briol.
> Towards healing the blindness of score matching. In NeurIPS 2022 Workshop on Score-Based
> Methods, 2022.

---

> > ### Author Response · Authors · 2024-11-24
> > **Rebuttal**
> >
> > ---
> >
> > 12. I find the numerical experiments in Figure 4 more enlightening than the statement in Example 1. ...
> >
> > First, the “score error” mentioned in Figure 4 is a type of $L^2$ error, defined as:
> >
> > $$
> > \text{score error} := \mathbb{E} _{t \sim \text{Unif}[0,T]} \mathbb{E} _{\boldsymbol{x} _t \sim \pi _t} \left[\|\boldsymbol{s} _\theta(\boldsymbol{x} _t,t) - \nabla _{\boldsymbol{x} _t}\log \pi _t(\boldsymbol{x} _t)\| _2^2 \right].
> > $$
> >
> > We want to emphasize that this phenomenon has been widely observed in a series of studies where they call it the blindness of score matching. Unless the target distribution is regular enough (e.g., satisfying the log sobolev inequality), we believe it would carry over to other examples as well.
> > In Figure 4, training for 400k iterations is sufficient for 2D experiments. We observe that while the PINN residual loss for the score FPE decreases with additional training iterations, the score error remains nearly unchanged after a few iterations. In contrast, when solving the log-density FPE, the score error consistently decreases as the PINN residual loss decreases with more training iterations. These observations highlight the limitations of solving the score FPE and the advantages of solving the log-density FPE instead, especially for challenging target distributions with isolated modes.
> >
> > It is also important to note that the score FPE is sufficient for regularizing diffusion models. While we show that solving the score FPE has limitations in identifying mixing proportions for multimodal sampling, denoising score matching on training data can help mitigate this issue. However, in our sampling setting, we lack access to samples from the target distribution (so we cannot use denoising score matching directly) and must rely more heavily on the partial differential equations. In this scenario, the log-density FPE is shown to be more effective in capturing the correct mixing proportions.
> >
> > ---
> >
> > 13. If one approximates the log-density as you do, one later must take gradients to get a score approximation, which might incur errors. ...
> >
> > Our theoretical results demonstrate that the score error can be effectively controlled as long as the log-density is well-approximated. Our numerical results also indicate that these approximations are good enough to provide accurate samples from the target distributions. An accurate score approximation can be achieved when the log-density approximation is sufficiently accurate. For the experimental results, please refer to the ablation study section in the global response.
> >
> > ---
> >
> > 14. Did you use Hutchinson’s gradient estimator that you introduce in lines 262 and following? ...
> >
> > First, the numerical experiments in our work are relatively low-dimensional, where directly computing the Laplacian is fast enough (especially those with $d=2$) and yields better results. However, we also applied the proposed Hutchinson’s gradient estimator in our high-dimensional experiments ($30$-dimensional Double-Well). While it is true that using Hutchinson’s estimator increases variance during training, making it less stable, it is important to note that the approach remains valid and one can also use a smaller learning rate to make it more stable.
> >
> > Table 1: KL divergence to the ground truth and mixing proportions estimation error obtained by different methods.
> >
> > | Methods                           | KL divergence ($\downarrow$)   | mixing proportions estimation error ($\downarrow$) |
> > | --------------------------------- | ------------------------------ | -------------------------------------------------- |
> > | **LMC**                           | $0.1915_{\pm 0.0122}$          | $0.0673_{\pm 0.0082}$                              |
> > | **RDMC**                          | $1.5735_{\pm 0.0162}$          | $0.2154_{\pm 0.0075}$                              |
> > | **SLIPS**                         | $0.4840_{\pm 0.0145}$          | $0.1645_{\pm 0.0113}$                              |
> > | **PIS**                           | $0.0969_{\pm 0.0114}$          | $0.0044_{\pm 0.0011}$                              |
> > | **DIS**                           | $0.6796_{\pm 0.0139}$          | $0.0684_{\pm 0.0035}$                              |
> > | **SVGD**                          | $1.3768_{\pm 0.0683}$          | $0.2400_{\pm 0.0174}$                              |
> > | **HMC**                           | $1.6729_{\pm 0.0303}$          | $0.9773_{\pm 0.4020}$                              |
> > | **PIS-LV**                        | $0.0478_{\pm 0.0280}$          | $\mathbf{0.0005_{\pm 0.0004}}$                     |
> > | **DIS-LV**                        | $0.0358_{\pm 0.0256}$          | $0.0012_{\pm 0.0009}$                              |
> > | **DPS (ours) without Hutchinson** | $\mathbf{0.0273_{\pm 0.0113}}$ | $\mathbf{0.0004_{\pm 0.0002}}$                     |
> > | **DPS (ours) with Hutchinson**    | $0.0435_{\pm 0.008}$            | $\mathbf{0.0008_{\pm 0.0005}}$                     |

---

> > > ### Author Response · Authors · 2024-11-24
> > > **Rebuttal**
> > >
> > > ---
> > >
> > > 15. In (20) one considers $L_{\rm PINN}(t_k,C_1(\varepsilon))$ for multiple $t_k$, however, the integral in $L_{\rm PINN}$ always starts at t=0. ...
> > >
> > > The term will not increase with finer time discretization. Specifically, we have:
> > > $$\\begin{aligned}
> > > \varepsilon\sum _{k=1}^N h _k R _{t _k}L _{\rm PINN}(t _k, C _1(\varepsilon)) \leqslant &\ \varepsilon\max _{t\in [0,T]}\\{R _t\\}\cdot \sum _{k=1}^N h _k\int _0^{t _k}e^{C _1(\varepsilon)(t _k-s)}\\|r _s(\cdot)\\| _{L^2(\Omega;\nu_s)}^2\mathrm{d}s \\\\ \leqslant &\ \varepsilon\max _{t\in [0,T]}\\{R_t\\}\cdot\sum _{k=1}^N h _k\int _0^{t _k} e ^{C _1(\varepsilon)(T-s)} \\|r _s(\cdot)\\| _{L ^2(\Omega;\nu _s)}^2 \mathrm{d}s \\\\
> > > =&\ \varepsilon \max _{t\in [0,T]}\\{R _t\\}\cdot \sum _{k=1}^N \int _{t _{k-1}}^{t _k} \left(\sum _{l=k}^N h _l\right) e ^{C _1(\varepsilon)(T-s)} \\|r _s(\cdot)\\| _{L ^2(\Omega;\nu_s)}^2 \mathrm{d}s \\\\
> > > =&\ \varepsilon\max _{t\in [0,T]}\\{R _t\\}\cdot \sum _{k=1}^N \int _{t _{k-1}}^{t _k} \left(T-t _{k-1}\right) e ^{C_1(\varepsilon)(T-s)}\\|r _s(\cdot)\\| _{L ^2(\Omega;\nu _s)}^2 \mathrm{d}s \\\\
> > > \leqslant &\ \varepsilon\max _{t\in [0,T]}\\{R _t\\}\sum _{k=1}^N \int _{t _{k-1}}^{t _k} T e ^{C _1(\varepsilon)(T-s)} \\|r _s(\cdot)\\| _{L
> > >  ^2(\Omega;\nu _s)}^2\mathrm{d}s \\\\
> > > =&\ T\varepsilon\max _{t\in [0,T]}\\{R_t\\}\cdot \int _{0}^{T}e^{C _1(\varepsilon)(T-s)}\\|r _s(\cdot)\\| _{L ^2(\Omega;\nu _s)}^2 \mathrm{d}s.
> > > \\end{aligned}
> > > $$
> > >
> > > Thus, this term can be upper-bounded by the universal objective
> > > $$
> > > T\varepsilon\max_{t\in [0,T]}\\{R_t\\}\cdot \int_{0}^{T}e^{C_1(\varepsilon)(T-s)}\\|r_s(\cdot)\\|^2_{L^2(\Omega;\nu_s)}\mathrm{~d}s,
> > > $$
> > > which is independent of the discretization schedule. As $N \to \infty$ , this term does not diverge.
> > >
> > > ---
> > >
> > > Thanks for the careful reading! We will modify the notations and correct the typos in our revision.

---

> > > > ### Comment · Reviewer_X3WA · 2024-11-27
> > > >
> > > > Dear authors, thank you very much for providing such a detailed response - I really appreciate your effort. Some of my points have been clarified, however, some remain open or at least did not lead to an agreement yet. Let me comment on those ones.
> > > >
> > > > 1. **Novelty.** This is still my biggest concern. The fact that the works [1, 2, 3] have been out before your work certainly lowers your contribution, since, in fact, the algorithms are almost identical (with the difference of using LMC, which is not a big change in algorithmic design, essentially also relying on already existing methods; also, see point 4).
> > > > 2. **Numerical evaluation.** Thank you for adding one more experiment - which is, however, very similar to the already considered double well example in $d=30$ with $8$ modes. Also, note that compared to other diffusion-based sampling papers, e.g. [10, 11], your experiments are rather toyish and it is not clear how your method will cope with even more complicated problems (in even higher dimensions, say, $d > 1000$). Also, is there a reason why you didn't compare against the log-variance versions of PIS and DIS for this example, as you did for some other examples in your general response?
> > > > 3. **Practical relevance.** Thank you for further explaining this aspect, however, I must admit that I am not fully convinced yet, especially since it is clear that a PINN loss converging to zero implies that also the sampling error converges to zero (neglecting time discretization aspects and a potential misfit of the prior density). Regarding this aspect, please also note point 13.
> > > >
> > > > 4. **MCMC algorithms.** You say that "MCMC methods are sufficient ... [if one requires]  samples that adequately cover the high-density domain". I would politely disagree since this very much depends on the energy surface. For instance, one can imagine high dimensional, very multimodal densities where (a variant of) an MCMC algorithm would not find certain modes at all - unless it runs for a very (!) long time. In fact, it is known that crossing certain energy barriers with, e.g., Langevin diffusion, scales exponentially in the height of the barriers, see, e.g., [12].
> > > >
> > > >
> > > > 10) **Comparable results with other papers.** Thanks for adding the Wasserstein metric, this helps to compare with other papers. When assessing your numbers attained for the "9-Gaussians" example, it seems that they are worse than the ones reported in, e.g., [3] and [7]. Do you agree?
> > > >
> > > > 11. **Fisher vs. KL divergence.** Thank you for your answer, which helps me to further understand your result. Do I understand it correctly that your observation cannot be carried over to non-mixture models though?
> > > >
> > > > 13) **Relation of log-density and score errors.** Can you maybe further elaborate on this aspect? Let me give you the following example. Assume the log-density  $V(x,t) := \log p(x, t) = -\frac{1}{2} x^2$ and its perturbed version $V_p(x,t) := V(x, t) + 0.1 \sin(100x)$. Since the perturbation is rather small, $V(x,t)$ and $V_p(x,t)$ are rather similar (e.g. in $L^2$ norm). The scores can be readily computed as $\nabla V(x,t) = -x$ and $\nabla V_p(x, t) = -x + 10 \cos(100x)$. Notably, $\nabla V(x,t)$ and $\nabla V_p(x, t)$ are rather different, for instance the sign of $\nabla V_p(x, t)$ is changing frequently. This gives me the intuition that approximating the log-density well does not necessarily imply that the score is approximated well too. Even more so, the error in the score approximation can be orders of magnitudes larger than the log-density approximation. Do you agree? If so, how is this reflected in your theoretical analyses and results? Does this not say that, due to the fact that you can't really say much about the constants, e.g., in your Theorem 2, the results are not necessarily very relevant in practice? You write that your "theoretical results demonstrate that the score error can be effectively controlled as long as the log-density is well-approximated". I think the above example shows that this is not necessarily correct, at least not in what is relevant in practice.
> > > >
> > > >
> > > > In view of the above (and realizing that the next evaluation level would already be two points more), I would like to keep my previous score. Please let me know in case I misunderstood something.
> > > >
> > > >
> > > >
> > > >
> > > >
> > > >
> > > > [10] Blessing D, Jia X, Esslinger J, et al. Beyond ELBOs: A Large-Scale Evaluation of Variational Methods for Sampling.
> > > >
> > > > [11] Tara Akhound-Sadegh, Jarrid Rector-Brooks, Avishek Joey Bose, Sarthak Mittal, Pablo Lemos, Cheng-Hao Liu, Marcin Sendera, Siamak Ravanbakhsh, Gauthier Gidel, Yoshua Bengio, et al. Iterated denoising energy matching for sampling from boltzmann densities. arXiv preprint
> > > > arXiv:2402.06121, 2024
> > > >
> > > > [12] N. Berglund. Kramers’ law: validity, derivations and generalisations. Markov Processes and Related fields, 19(3):459–490, 2013.

---

> ### Author Response · Authors · 2024-12-02
>
> Thanks for your response! We have addressed your concerns as follows.
>
> ---
>
> 1. Novelty ...
>
> Compared to [1], where the combination of PINN and sampling is only briefly mentioned **in the appendix**, our work provides a comprehensive investigation of diffusion-based sampling using PINN. This includes investigating limitations highlighting the drawbacks of using the score Fokker-Planck equation (FPE) in diffusion-based sampling to **identify mixing proportions**, which is known as blindness of score matching, as well as offering a complete theoretical analysis of the sampling algorithm combined with the PINN approach.
>
> In contrast to [2], it is important to note that [2] considers a linear interpolation path (in log space) for sampling, which differs from the reverse process of diffusion models employed in our work.
>
> Finally, in comparison to [3], which is a concurrent work to ours (**notably, our work was submitted to NeurIPS 2024 and was very close to an acceptance with an average score of $6$!**), our study includes additional theoretical analysis and focuses on more challenging situations on **identifying mixing proportions** for target distributions with isolated modes, setting it apart in terms of depth and contribution.
>
> ------
>
> 2. Numerical evaluation ...
>
> The scalability of our approach primarily depends on the effectiveness of PINN in high-dimensional settings. Fortunately, recent advancements ([1,2]) have provided several methods that successfully improve the ability of PINN to solve relatively high-dimensional PDEs. These scaling techniques for PINN can be directly incorporated into our sampling framework, enabling our sampling algorithm to scale effectively as well.
>
> We did not include the results of the log-variance versions of PIS and DIS because their execution time, using the officially released implementation, exceeds two days. Nonetheless, we agree that PIS-LV and DIS-LV would perform comparably to our method.
>
> ---
>
> 3. Practical relevance ...
>
> We want to emphasize that our theoretical results establish a connection between the approximation error of both the log-density and the score in terms of the PINN residual loss. Specifically, if the PINN residual loss is sufficiently small, then the approximation errors for both the log-density and the score will also be small.
>
> Intuitively, the PINN residual loss provides a higher-order constraint on the log-density approximation, which is different from a simple $L_2$ loss. **In order to get a small PINN residual loss, not only the log-density approximation needs to be accurate, but also its higher-order gradients (including the score) that are involved in the PINN residual evaluation need to be accurate as well.**
> To illustrate this, we provide the following example. In our implementation, the PINN residual is defined as:
> $$
> r_t(\boldsymbol{x}) = \partial_t u_t(\boldsymbol{x}) - \frac{1}{2(1-t)}\left[\Delta u_t(\boldsymbol{x}) + \\|\nabla u_t(\boldsymbol{x})\\|^2 + \boldsymbol{x} \cdot \nabla u_t(\boldsymbol{x}) + d\right].
> $$
> For simplicity, let’s consider the one-dimensional case where $u_t(x) = -0.5x^2$ . In this case, the residual becomes:
> $$
> r_t(x) = \partial_t u_t(x) - \frac{1}{2(1-t)}\left[u^{\prime\prime}(x) + \left(u^\prime(x)\right)^2 + x u^\prime(x) + 1\right] = 0.
> $$
> However, if we introduce a perturbation $\tilde{u}_t(x) = -0.5x^2 + 0.1\sin(100x)$ , the PINN residual becomes:
> $$
> \tilde{r}_t(x) = -\frac{1}{2(1-t)}\left(-1 - 1000\sin(100x) + (-x + 10\cos(100x))^2 + x(-x + 10\cos(100x)) + 1\right) = \frac{1}{2(1-t)}\left(1000\sin(100x) - 100\cos^2(100x) + 10x\cos(100x)\right),
> $$
> which can be significantly large. This example is consistent with our theoretical findings, demonstrating that a small PINN residual loss (not the $L_2$ loss) is critical for ensuring accurate (score) approximations.
>
> Furthermore, we conducted numerical experiments to address your concern. Specifically, we computed the PINN residual loss for both the well-trained model and perturbed well-trained model on the 9-Gaussian target. The results are summarized in the following table:
>
> Table: Comparison of PINN residual between well-trained models with and without slight perturbations
> | | **PINN residual loss** |
> | ---------------------- | -------------- |
>  | **Well-trained Model** | $0.00172$ |
> | **Well-trained Model (Slight Perturbed)** | $2018105.634$|
>
> From the above results, it is evident that the perturbation causes a significantly large PINN residual loss, despite only a slight variation in the log-density. Due to the large PINN residual loss, achieving a small score approximation error is not feasible.

---

> ### Author Response · Authors · 2024-12-02
>
> 4. MCMC algorithms. ...
>
> Traditional MCMC algorithms typically require an extremely long time to generate perfect samples, especially when dealing with high-dimensional multi-modal target distributions. This is due to the inherent trade-off between exploration and accuracy, which leads to prolonged mixing times. However, in our work, we use MCMC methods (e.g., LMC) solely to generate collocation points, with the goal of sufficiently covering the entire high-density domain rather than obtaining perfect samples. This relaxed requirement allows us to adopt a larger step size for LMC, enabling sufficient exploration while eliminating the original trade-off. This approach highlights a key advantage of off-policy training in our framework.
>
> ---
>
> 5. Comparable results with other papers ...
>
> It is important to highlight that, in our experiments, the multi-modal target distributions are designed with challenging (unequal) mixing proportions, making them more difficult to sample from. This may explain why most baseline methods tend to perform worse.
>
> ---
>
> 6. Fisher v.s. KL ...
>
> Yes. Our results are mainly for mixture models. In this paper, our goal is to develop an efficient sampling algorithm for multi-modal distributions, which is challenging for traditional methods. For simpler distributions (i.e., single-mode distributions), traditional MCMC methods such as HMC would be enough to provide good samples.
>
> ---
>
> References:
>
> [1]. Zheyuan Hu, Zekun Shi, George Em Karniadakis, and Kenji Kawaguchi. Hutchinson trace estimation
> for high-dimensional and high-order physics-informed neural networks. Computer Methods in
> Applied Mechanics and Engineering, 424:116883, 2024a.
>
> [2]. Zheyuan Hu, Khemraj Shukla, George Em Karniadakis, and Kenji Kawaguchi. Tackling the curse of
> dimensionality with physics-informed neural networks. Neural Networks, pp. 106369, 2024b.

---

> > ### Comment · Reviewer_X3WA · 2024-12-02
> >
> > Dear authors, thank you very much for your additional reply. I would still prefer to keep my score - for the reasons I have mentioned in my previous response. Additionally, let me add the following:
> >
> > 1. **Novelty.** While it is true that [2] does not explicitly consider score-based generative modeling, it not only considers a "linear interpolation path (in log space) for sampling". In equation (20) in [2] one can see that the path is learned and only initialized with a linear interpolation in log-space. I am sorry to hear about your NeurIPS experience, but of course I cannot verify this.
> >
> > 3) **Practical relevance.** In your previous response you wrote that the "theoretical results demonstrate that the score error can be effectively controlled as long as the log-density is well-approximated". This is why I provided the example with the perturbation, that shows that the score error can be quite large even though the log-density is rather well approximated (in $L^2$). So do you agree that your statement might have been misleading? I agree with your example that the PINN loss might change drastically with small perturbations though.
> >
> > 4. **MCMC.** As said before, I want to note that also with MCMC-like methods high-probability regions often get missed.
> >
> >
> > In case there will be an internal discussion period, I will of course listen to the opinions of the other reviewers.

---

> ### Author Response · Authors · 2024-12-04
>
> Thanks for your response! We address your concerns as follows.
>
> ---
>
> 1. Novelty ...
>
> We agree that [2] also considered learnable paths other than linear interpolation in the log space. However, learning these paths requires extra computation and may incur instability during training.
>
> ---
>
> 3. Practical relevance ...
>
> Apologies for the misunderstanding. To clarify, our original goal is to show that the approximation error for the score function can be effectively controlled as long as the log-density is well-approximated by minimizing the PINN residual.
>
> ---
>
> 4. MCMC ...
>
> As mentioned before, in our case, we are not running traditional MCMC to get samples from the target distribution for the collocation points. Due to the off-policy nature of our method, we can run a much shorter and noisier (and parallel) MCMC (e.g., LMC) with large step sizes that are more likely to capture the high-probability regions. This is more practical than uniform sampling from a pre-selected high probability region (adopted in previous PINN-based sampling algorithms) that are often hard to obtain, especially in high dimensions.

---

### Official Review · Reviewer_RBAG · 2024-11-03

**Soundness:** 3
**Presentation:** 4
**Contribution:** 3
**Rating:** 8
**Confidence:** 4

**Summary:**

The paper presents a new approach for diffusion sampling. Inspired by existing methods that use physics-informed neural networks, they address the problem of solving the reverse diffusion process using the Fokker Plank equation which models the evolution of marginals $p_t(x)$ in the forward process. Specifically, they identify flaws of existing methods and address them. The first one is the failure of the score FPE and the second is the lack of theoretical analysis of proposed methods. They end up with a method called DPS (Diffusion-PINN sampler) that solves for log-density FPE. They show promising results and consistent improvement over other methods, providing, in addition, convergence analysis of the method.

**Strengths:**

The paper is very well written! It was pleasant to read and easy to follow.

- The authors provide observations of failure of score FPE and propose a motivated and illustrated solution : solving the log-density FPE and plugging it in the reverse process afterwards.

- The authors provide a good amount of experiments and parsimonious illustrations (just what is needed to illustrate their claims).

- The authors assess performances of their methods with good metrics. It could seem naive to say, but a lot of people miss the use of probabilistic measures for probabilistic problems. The choice of KL/Fisher divergence is welcomed along with the $L^2$ of the log-density.

- The use of PINN for such a problem is relevant. It bases the method on well-established (and timely) fields which is nice!

- The overall structure of the work is clean and each part is self-sufficient.

- Analysis, performed on toy problems at first, allows for good interpretation of the results.

- The proposed sampling methods show big improvement over other baselines for considered problems which further motivates it.

**Weaknesses:**

I'm really sorry, but I think you will have to change the name (at least the reduced name) of your method! DPS is already a well known and well established sampling method for posterior problems [1]. I think it would be beneficial to avoid being shadowed by existing method. Moreover, it's more than just a sampling method that you present. It's a conjugate training and sampling algorithm. Your main change is about what your network is trained for (the log density) and not about the FPE which is not new in itself (Although the training and the sampling make a whole).

Otherwise, I point below a few questions and small concerns.

**Small details**

- Line 60: *to ensure convergence guarantee*, it's nitpicking but isn't it a bit redundant?

**Performance concerns**

- I'm a little bit concerned about the performances of the method both in time and resources. It would be nice to, at least discuss them and at best provide some measures in Appendix for example. I detail below my concerns.

- You model the log-density, then you have to take its gradient in order to compute the score that you plug into the reverse process, right? What is the cost of this compared to direct score matching? Indeed, I presume that taking the grad at every step is not without cost.

- The PINN loss involves gradient and Laplacian operators. What is the cost of computing such a loss compared to simple denoising score matching (which, at the end, is not more complex than an mse)? Does it slow down training? Do you use optimized tricks like jvp/vjp when possible? Could you comment a bit on that please?

- In table 1, isn't it misleading to report KL only on first dimensions for Funnel and Double-well? Can't you report either on several sub-combinations of dimensions or use another measure? Is it due to the cost of the KL in higher dimension? What about more tractable divergence metrics such as the Sinkhorn divergence?

**Sampling method**

- From your text and as the first requirement in **Algorithm 1** you speak about access to the initial condition which derives from the unnormalized density. It is still not clear to me how you access to such density?

- You present a sampling method, and even more, for diffusion models. It would be nice to have a higher dimensional problem (I agree that it is not necessary to cover your claims here). This method should be studied in more realistic settings also (with images, high dimensional data, ...) and I'm worried about the cost of computing explicitly the score from the log-density or the cost of the loss in such settings. At least comment on that in the main paper.

- Echoing with the previous comment: What about conditional sampling? Posterior problems. Technically your method can extend easily. You compute the prior score as for now and plug the likelihood to guide the sampling. It would be nice (and I think easy) to show, in Appendix if you lack space, the conditional generation for one toy problem. For example, generating only on one mode in the 9-Gaussians or Rings problem. I'm curious to see the benefit of more accurate prior score to such problems. I'm also curious to see if you still perform much better than other considered methods. Indeed, the likelihood guiding score can sometimes help and simplify a bit the problem.

- Nitpick: In figure 3 you say "Sampling performance". Those figures do not show any performances. Maybe change it to "Samples from different methods ...".


**Additional suggestion**

Figure 5 (center and right) are not really revealing. I know that it's hard to display samples in high dim. Could you put corner plots instead? Even if it's on a subset of dimension of the problem. It would be easier to see the differences with and without regularization if you display marginals along this 2-dimensional joint.


My current score is 6. I would gladly increase my score once my concerns have been addressed and discussed.
Thanks for your good work!

**[1] Diffusion Posterior Sampling for General Noisy Inverse Problems - ICLR 2023 spotlight**

**Questions:**

See weaknesses.

---

> ### Author Response · Authors · 2024-11-24
> **Rebuttal**
>
> Thanks for your suggestion! We will modify the name accordingly in our revision.
>
> ---
>
> 1. I'm a little bit concerned about the performances of the method both in time and resources. ...
>
> We provide measurements of training time and sampling time across different dimensional cases in the complexity analysis section of the global response.
>
> ---
>
> 2. You model the log-density, then you have to take its gradient in order to compute the score ...
>
> In the complexity analysis section of the global response, we compare the sampling time using direct score estimation versus taking the gradient of the approximated log-density. Our results show that sampling time is halved when score estimation is directly employed. Nonetheless, the sampling time of our method is already highly efficient. For instance, sampling 10,000 points over 1,000 time steps takes less than 2 seconds.
>
> ---
>
> 3. The PINN loss involves gradient and Laplacian operators. ...
>
> Our training objective is inherently more complex than the simple denoising score matching objective, as it includes terms like Laplacian operators. Consequently, our training process is slower compared to simple denoising score matching. Notably, we have not yet incorporated optimized techniques such as jvp/vjp for implementation. Instead, we address the computational challenges by proposing an **unbiased Hutchinson gradient estimator** to avoid the costly evaluation of the Laplacian. This estimator significantly reduces computational costs and accelerates training, as detailed in the numerical complexity analysis section of the global response.
>
> Moreover, it is important to highlight a key distinction: simple denoising score matching relies on access to a large amount of training data sampled from the target distribution, which is available in the context of generative modeling. In contrast, our focus is on sampling from an unnormalized density, where such training data is unavailable. As a result, we have to utilize a more complex objective to accurately approximate the score function in this setting.
>
> ---
>
> 4. In table 1, isn't it misleading to report KL only on first dimensions for Funnel and Double-well? ...
>
> This evaluation standard leverages the properties of the target distributions (Funnel and Double-Well). In the Funnel case, given the first dimension, the remaining dimensions are independent and identically distributed. Similarly, in the Double-Well case, the distribution is dimension-independent, with all dimensions following a standard Gaussian distribution except for the first three (in the 30-dimensional case) or the first five (in the 50-dimensional case).
>
> As a result, in both cases, analyzing the KL divergence over the first five dimensions is sufficient to capture the differences between the target and approximate distributions. This approach effectively highlights the key discrepancies while avoiding unnecessary computation across redundant dimensions.
>
> ---
>
> 5. From your text and as the first requirement in **Algorithm 1** you speak about access to the initial condition which derives from the unnormalized density. ...
>
> In this work, we focus on sampling from an unnormalized density, such as posterior sampling in Bayesian inference. In this setting, we assume access to the exact formula of the unnormalized density, but we do not have access to samples from the target distribution. Therefore, our approach inherently requires access to the unnormalized density, unlike generative modeling, which typically relies on samples from the target distribution.
>
> ---
>
> 6. You present a sampling method, and even more, for diffusion models. It would be nice to have a higher dimensional problem (I agree that it is not necessary to cover your claims here). This method should be studied ...
>
> The main purpose of our paper is to leverage the reverse process of diffusion models to design efficient sampling algorithms, instead of generative models.
> Therefore, we do not expect our method to be studied in real high dimensional data such as images. In those cases, one could use diffusion models directly and with data samples, score approximation can be easily obtained with denoising score matching.
>
> Furthermore, leveraging PyTorch’s vectorized implementation ensures that the cost of score computation during sampling remains efficient. While it is slower than using direct score estimation, it is still highly performant—sampling 10,000 points over 1,000 time steps for 100-dimensional tasks takes less than 2 seconds. In addition, we introduce an unbiased Hutchinson gradient estimator to mitigate the high computational cost associated with evaluating the Laplacian term in the PINN loss. Our numerical results, presented in the complexity analysis section of the global response, substantiate these claims.
>
> For higher-dimensional problems, see the results in our global response.

---

> > ### Author Response · Authors · 2024-11-24
> > **Rebuttal**
> >
> > ---
> >
> > 7. Echoing with the previous comment: What about conditional sampling? Posterior problems. Technically your method can extend easily. You compute the prior score as for now and plug the likelihood to guide the sampling. It would be nice (and I think easy) to show, in Appendix if you lack space, the conditional generation for one toy problem. For example, generating only on one mode in the 9-Gaussians or Rings problem. I'm curious to see the benefit of more accurate prior score to such problems. I'm also curious to see if you still perform much better than other considered methods. Indeed, the likelihood guiding score can sometimes help and simplify a bit the problem.
> >
> > Thanks for raising up this interesting question! There is a major difference between conditional sampling and our goal, that is to sample from a target distribution with an unnormalized density (e.g., the posterior distribution). For conditional sampling, the prior is the clean data distribution whose density is unavailable and is often estimated via diffusion models. The forward likelihood then is learned with labeled noisy data and is used to guide the diffusion model for conditional generation. In our case, the prior is often specified instead of learned from the data, and the likelihood is given as the probabilistic model. Therefore, posterior sampling can be done directly using our method (this is exactly what we want to do with the proposed DPS algorithm!). We hope this resolves your concerns.
> >
> > ---
> >
> > 8. Figure 5 (center and right) are not really revealing. ...
> >
> > Thanks for your suggestions! We have already presented corner plots in the Appendix D.3.1 of our revision (Please see Figure 9).

---

> > > ### Comment · Reviewer_RBAG · 2024-11-24
> > > **Answer to the rebuttal**
> > >
> > > Thanks for your modifications and discussions. As said, I increase my score accordingly.

---

> > > > ### Author Response · Authors · 2024-11-25
> > > > **Thanks for raising the score!**
> > > >
> > > > Thanks for raising the score!

---

### Official Review · Reviewer_YacX · 2024-11-04

**Soundness:** 2
**Presentation:** 2
**Contribution:** 2
**Rating:** 5
**Confidence:** 5

**Summary:**

This paper explores diffusion-based models for unnormalized sampling and introduces the Diffusion-PINN Sampler (DPS). Using physics-informed neural networks (PINNs), DPS directly approximates the log-density of SDE marginals, enabling more precise modeling of complex distributions. The authors provide theoretical convergence analysis and validate the method on several synthetic datasets.

**Strengths:**

- The paper is generally easier to follow. Theoretical results in Sec 5 could be of interested for readers from other domains.

**Weaknesses:**

- The proposed method is computationally expansive compared to other diffusion-based models to train—due to the Laplacian in PINN loss—and to sample from—due to the evaluation of gradient at every time step.
- Insufficient comparison to modern baselines — in addition to  MC methods like HMC or SMC, there’re many other diffusion/SDE based methods, such as [1,2], just to name a few.
- Experiments were only conducted on rather simple, synthetic, target. I’ll be more convinced to see some higher-dim experiments and/or real-world dataset.
- Parametrize NN with log mu (12) seems like a strong inductive bias and can be infeasible for many practical applications (e.g., sample Boltzmann distribution for conformation generation) when querying energy functions are expansive. Can the authors provide ablation study without such parametrization?
- Related works Section is rather short and should be extended.

[1] Particle Denoising Diffusion Sampler (ICML 2024)
[2] Improved sampling via learned diffusions (ICLR 2024)

**Questions:**

- $\ell_reg$ seems ad-hoc. Why regress onto the boundary condition of the score instead of (11b), which seems more aligned with PINN loss?
- Given that x0 comes from LMC already, is the method computationally cheaper compared to standard MCMC methods?

---

> ### Author Response · Authors · 2024-11-24
> **Rebuttal**
>
> Thank you for your thoughtful review and valuable feedback. We address your specific questions and comments below.
>
> ---
>
> 1. The proposed method is computationally expansive compared to other diffusion-based models to train—due to the Laplacian in PINN loss—and to sample from ...
>
> First, we propose an **unbiased** Hutchinson gradient estimator to compute the PINN loss, significantly reducing the computational cost associated with evaluating the Laplacian in the loss function. Additionally, we compare the sampling time using direct score estimation versus taking the gradient of the approximated log-density. Our results show that sampling time is halved when score estimation is directly employed. Nonetheless, the sampling time of our method is already highly efficient. For instance, sampling 10,000 points over 1,000 time steps takes less than 2 seconds. For further discussion and additional experimental results, please refer to the complexity analysis section of the global response.
>
> ---
>
> 2. Insufficient comparison to modern baselines — in addition to MC methods like HMC or SMC, there’re many other diffusion/SDE based methods, such as [1,2], just to name a few.
>
> We have included additional experimental results for modern baselines, such as HMC and the recent diffusion-based methods introduced in [2], in the global response.
>
> ---
>
> 3. Experiments were only conducted on rather simple, synthetic, target. I’ll be more convinced to see some higher-dim experiments and/or real-world dataset.
>
>
> We present additional higher-dimensional experiments on $50$-dimensional Double-Well with $32$ separated modes, following the same setup as in [1] but with more challenging weights in the global response.
>
> ---
>
> 4. Parametrize NN with log mu (12) seems like a strong inductive bias ...
>
>
> First, our parameterization ensures that the parameterized solution satisfies the initial condition, eliminating the need to consider the initial loss in our training objective. However, we agree that this parameterization may incur higher computational costs, especially when evaluating the energy of the target distribution is expensive.
>
> In such cases, we can use a simple neural network and incorporate the initial loss term into the training objective, which can be computed via unbiased stochastic estimation to reduce the cost of evaluating the energy. Namely, we use the following training objective,
>
> $$
> L_{\textrm{MCMC}}(u_\theta):=\frac{1}{M}\sum _{i=1}^M \beta^2(t_i)\cdot  \\| \partial_t u _{\theta} (\boldsymbol{x}_i^{t_i},t_i) -
> \mathcal{L} _{\textrm{L-FPE}} u _{\theta}(\boldsymbol{x}_i^{t_i},t_i)\\|^2 + \frac{1}{M}\sum _{i=1}^M\\|u _\theta(\boldsymbol{x}_i^0,0)-\log\mu(\boldsymbol{x}_i^0)\\|^2 + \frac{\lambda}{M} \sum _{i=1}^M  \ell _{\textrm{reg}}(u _\theta; T, \boldsymbol{z}_i).
> $$
>
> Notably, in our experiments, the density function can be computed efficiently, so we chose this parameterization to avoid the additional burden of the initial loss term.
>
> Additionally, we conducted an ablation study on the 9-Gaussians task. The results demonstrate that including the initial loss in the training objective, rather than using this parameterization, still provides a good score approximation (and naturally achieves effective sampling). These experimental results are presented in the ablation study section of the global response.
>
> ---
>
> 5. Related works Section is rather short and should be extended.
>
>
> Thanks for the suggestion! We want to mention that the introduction section also discussed several related methods, including VI methods, MCMC methods, RDMC, PIS, DIS, and DDS. In the related works section, we further discuss other diffusion-based sampling techniques that involve solving ODEs/PDEs, to highlight the unique contribution of our approach. We will add more discussion to related works in our revision. Given that this is a rapidly growing research area, we admit that there may be some missing references and would greatly appreciate it if you could provide more related works as well.

---

> > ### Author Response · Authors · 2024-11-24
> > **Rebuttal**
> >
> > ---
> >
> > 6. $\ell_{\rm reg}$ seems ad-hoc. ...
> >
> >
> > It is important to note that the special parameterization used in this paper inherently satisfies the boundary condition (11b). As a result, there is no need to include a boundary loss during training. The term $\ell_{\rm reg}$ serves as a regularization term, leveraging the property of the forward process in the diffusion model, specifically that $\pi_T \approx \pi_{\rm prior}$ for large $T$, where $\pi_{\rm prior}$ is typically a standard Gaussian distribution. This prior knowledge motivates the design of the regularization term, making the training process more efficient. The positive impact of this regularization term has also been demonstrated in our ablation study of the main paper (for the Funnel case).
> >
> > ---
> >
> > 7. Given that x0 comes from LMC already, is the method computationally cheaper compared to standard MCMC methods?
> >
> >
> > First, during training, we only need to run a much shorter LMC chain with a large learning rate to generate collocation points. This is because the collocation points only need to cover the high-density region, rather than being perfect samples. Additionally, our method functions as a learning-based sampling technique. Once the approximate perturbed log-density is trained, sampling can be performed in a relatively small number of steps (e.g., 1,000), much fewer than in standard MCMC methods. This process, similar to the generation step in diffusion models, takes only a few seconds (for more details on the sampling time, please refer to the global response).
> >
> > Moreover, our sampling framework can leverage various acceleration techniques from diffusion models, such as deterministic sampling, to further speed up the sampling process. In contrast, standard MCMC methods like LMC and HMC typically require much longer iterations to converge to the target distribution. For challenging multi-modal distributions, standard MCMC methods often need $10,000$ or more iterations to achieve convergence or even fail to converge. As a result, our sampling process is significantly more efficient compared to traditional MCMC methods.
> >
> > ---
> >
> > References
> >
> > [1] Sun, Jingtong, et al. "Dynamical measure transport and neural PDE solvers for sampling." arXiv preprint arXiv:2407.07873 (2024).

---

> > > ### Comment · Reviewer_YacX · 2024-11-25
> > >
> > > I thank the authors for the detailed reply. I acknowledge the unbiased Hutchinson estimator and additional comparison. I understand the proposed method as an implicit matching of log-density (rather than conventional score), in which its training loss is evaluated on an approximate optimal distribution using LMC samples (whereas prior works use MCMC / importance sampling; all attempt to approximate the unknown optimal distribution). That being said, I still hold concerns on its scalability (as experiments remain synthetic), the validity of approximate distribution as the dimension grows, and novelty compared to prior works.
> > >
> > > Nevertheless, I adjust my score the reflect the current experiment results.

---

> ### Author Response · Authors · 2024-12-02
> **Thanks for raising the score!**
>
> Thanks for raising the score! For further discussions on the scalability and novelty of our work, please refer to the global response!

---

### Author Response · Authors · 2024-11-24
**Global response**

## Ablation study

We conduct several ablation studies in this section. First, we demonstrate that a good score approximation can be achieved as long as the log-density is well approximated, as supported by our theoretical results. The corresponding results are shown in the left panel of Figure 8 in the Appendix D.3.1 of our revision. Additionally, we show that both using the parameterization trick and including the initial loss in the objective lead to a good score approximation, as evidenced by the results in the right panel of Figure 8 in the Appendix D.3.1 of our revision.

## Complexity analysis

We propose an **unbiased** Hutchinson gradient estimator to reduce the computational cost of evaluating the Laplacian in the PINN loss. We examine its impact on training time and report the sampling time in the following table. Notably, without this estimator, training time increases significantly as the dimensionality grows. In contrast, our approach effectively mitigates this issue. Furthermore, once the log-density approximation is obtained, sampling can be performed in a remarkably short time, taking only $2$ seconds. Moreover, we compare the sampling time using direct score estimation versus taking the gradient of the approximated log-density. Our results show that sampling time is halved when score estimation is directly employed. Nonetheless, the sampling time of our method is already highly efficient. For instance, sampling 10,000 points over 1,000 time steps for 100-dim tasks takes less than 2 seconds.

Table 6: Complexity analysis for our method.

| **Time (s)**                              | 10d       | 20d       | 30d       | 40d       | 50d       | 60d       | 70d       | 80d       | 90d       | 100d      |
| -------------------------------------- | --------- | --------- | --------- | --------- | --------- | --------- | --------- | --------- | --------- | --------- |
| **Per iteration training (without trick)** | 0.037     | 0.062     | 0.086     | 0.111     | 0.135     | 0.160     | 0.184     | 0.210     | 0.235     | 0.260     |
| **Per iteration training (with trick)**   | **0.018** | **0.018** | **0.018** | **0.018** | **0.018** | **0.018** | **0.018** | **0.018** | **0.018** | **0.018** |
| **Sampling (taking gradient)**      | 1.537     | 1.534     | 1.538     | 1.551     | 1.558     | 1.567     | 1.570     | 1.581     | 1.587     | 1.596     |
| **Sampling (using score directly)**    | 0.683     | 0.685     | 0.701     | 0.731     | 0.736     | 0.742     | 0.759     | 0.775     | 0.794     | 0.812     |

---



References:

[1] Improved sampling via learned diffusions (ICLR 2024)

[2] Blessing D, Jia X, Esslinger J, et al. Beyond ELBOs: A Large-Scale Evaluation of Variational Methods for Sampling.

[3] Sun, Jingtong, et al. "Dynamical measure transport and neural PDE solvers for sampling." arXiv preprint arXiv:2407.07873 (2024).

---

### Author Response · Authors · 2024-11-24
**Global response**

## More evaluation metrics

Additionally, we provide results using additional evaluation metrics, including the $W_2$-distance and EMC value proposed in [2], for specific sampling tasks. These supplementary evaluations further highlight the superior sampling performance of our method, while PIS-LV and DIS-LV exhibit comparable performance, consistent with our main evaluation results.

Table 3: $W_2$-distance to the ground truth obtained by different methods

| **$W_2$-distance ($\downarrow$)** | **$9$-Gaussians**                 | **Rings**                         |
| --------------------------------- | --------------------------------- | --------------------------------- |
| **LMC**                           | $5.0064_{\pm 0.076}$              | $3.4096_{\pm 0.075}$              |
| **RDMC**                          | $3.1329_{\pm 0.084}$              | $1.2195_{\pm 0.102}$              |
| **SLIPS**                         | $0.9884_{\pm 0.143}$              | $2.2280_{\pm 0.064}$              |
| **DIS**                           | $5.4442_{\pm 0.056}$              | $3.2716_{\pm 0.078}$              |
| **PIS**                           | $5.0341_{\pm 0.052}$              | $3.7429_{\pm 0.070}$              |
| **SVGD**                          | $3.9018_{\pm 0.054}$              | $2.0395_{\pm 0.127}$              |
| **HMC**                           | $4.5945_{\pm 0.075}$              | $2.3255_{\pm 0.111}$              |
| **PIS-LV**                        | $4.5839_{\pm 0.047}$              | $\mathbf{0.7146_{\pm 0.076}}$     |
| **DIS-LV**                        | $1.0534_{\pm 0.214}$              | $\mathbf{0.7078_{\pm 0.118}}$     |
| **DPS (ours)**                    | $\boldsymbol{0.7794_{\pm 0.127}}$ | $\boldsymbol{0.7726_{\pm 0.099}}$ |

Table 4: EMC value obtained by different methods

| **EMC ($\uparrow$)** | **9-Gaussians**                   | Rings                         |
| :------------------- | --------------------------------- | ----------------------------- |
| **LMC**              | $0.3562_{\pm 0.005}$              | $0.3844_{\pm 0.007}$          |
| **SLIPS**            | $0.9822_{\pm 0.005}$              | $0.7434_{\pm 0.008}$          |
| **RDMC**             | $0.6195_{\pm 0.014}$              | $0.8754_{\pm 0.009}$          |
| **PIS**              | $0.3341_{\pm 0.003}$              | $0.3328_{\pm 0.004}$          |
| **DIS**              | $0.2988_{\pm 0.003}$              | $0.4081_{\pm 0.007}$          |
| **SVGD**             | $0.5095_{\pm 0.008}$              | $0.7093_{\pm 0.067}$          |
| **HMC**              | $0.4139_{\pm 0.075}$              | $0.4235_{\pm 0.008}$          |
| **PIS-LV**           | $0.3208_{\pm 0.001}$              | $\mathbf{0.9976_{\pm 0.002}}$ |
| **DIS-LV**           | $\mathbf{0.9942_{\pm 0.002}}$     | $\mathbf{0.9983_{\pm 0.001}}$ |
| **DPS (ours)**       | $\boldsymbol{0.9965_{\pm 0.002}}$ | $\mathbf{0.9928_{\pm 0.002}}$ |

## Higher-dimensional experiments

Furthermore, we extend our experiments on the Double-Well target to $50$ dimensions with $32$ separated modes for several baselines, following the setup in [3]. The results are presented in the table below. Notably, our target distribution features challenging mixing proportions, making it particularly difficult to sample effectively.

Table 5: Sliced KL divergence to the ground truth and mixing proportions estimation error obtained by different methods.

| **Method**     | **Sliced KL divergence  ($\downarrow$)** | **Mixing proportions estimation error ($\downarrow$)** |
| -------------- | ---------------------------------------- | ------------------------------------------------------ |
| **LMC**        | $0.321_{\pm 0.009}$                      | $0.0681_{\pm 0.0053}$                                  |
| **SLIPS**      | $0.745_{\pm 0.018}$                      | $0.1323_{\pm 0.0062}$                                  |
| **RDMC**       | $2.769_{\pm 0.022}$                      | $0.1641_{\pm 0.0048}$                                  |
| **PIS**        | $0.362_{\pm 0.012}$                      | $0.0203_{\pm 0.0021}$                                  |
| **DIS**        | $0.973_{\pm 0.017}$                      | $0.0808_{\pm 0.0051}$                                  |
| **SVGD**       | $24.349_{\pm 0.067}$                     | $0.1710_{\pm 0.0046}$                                  |
| **HMC**        | $0.245_{\pm 0.057}$                      | $0.0212_{\pm 0.0038}$                                  |
| **DPS (ours)** | $\boldsymbol{0.101_{\pm 0.011}}$         | $\boldsymbol{0.0008_{\pm 0.0003}}$                     |

---

### Author Response · Authors · 2024-11-24
**Global response**

# Global response

We sincerely thank all reviewers for their constructive feedback. In response, we present additional experimental results as follows.

## More baselines

We provide a comprehensive comparison with additional baselines, including MCMC methods (LMC, HMC), particle-based VI methods (SVGD), diffusion-based sampling approaches (RDMC, SLIPS, DIS, PIS), and recent variants of diffusion-based sampling methods (PIS-LV, DIS-LV) as outlined in [1]. The results demonstrate that our method consistently achieves superior sampling performance across various tasks. Notably, recent variants of diffusion-based sampling methods (PIS-LV, DIS-LV) exhibit comparable performance to ours on certain tasks but tend to be less stable, as evidenced by the issue such as missing several modes in the 9-Gaussians example for PIS-LV.

Table1: KL divergence to the ground truth obtained by different methods.

| KL divergence ($\downarrow$) | $9$-Gaussians                  | Rings                          | Funnel                         | Double-well                    |
| ---------------------------- | ------------------------------ | ------------------------------ | ------------------------------ | ------------------------------ |
| **LMC**                      | $1.6568_{\pm 0.0189}$          | $2.4754_{\pm 0.0302}$          | $0.1908_{\pm 0.0156}$          | $0.1915_{\pm 0.0122}$          |
| **RDMC**                     | $1.0844_{\pm 0.0132}$          | $0.7487_{\pm 0.0073}$          | $2.0250_{\pm 0.0364}$          | $1.5735_{\pm 0.0162}$          |
| **SLIPS**                    | $0.0901_{\pm 0.0071}$          | $0.4127_{\pm 0.0144}$          | $0.1971_{\pm 0.0133}$          | $0.4840_{\pm 0.0145}$          |
| **DIS**                      | $2.2758_{\pm 0.0240}$          | $2.3433_{\pm 0.0275}$          | $0.2383_{\pm 0.0169}$          | $0.6796_{\pm 0.0139}$          |
| **PIS**                      | $2.0042_{\pm 0.0203}$          | $2.6985_{\pm 0.0290}$          | $0.4377_{\pm 0.0199}$          | $0.0969_{\pm 0.0114}$          |
| **SVGD**                     | $0.9712_{\pm 0.0153}$          | $0.1608_{\pm 0.0119}$          | $0.1006_{\pm 0.0188}$          | $1.3768_{\pm 0.0683}$          |
| **HMC**                      | $1.3814_{\pm 0.0171}$          | $1.0258_{\pm 0.0297}$          | $\mathbf{0.0289_{\pm 0.0266}}$          | $0.1546_{\pm 0.0474}$          |
| **PIS-LV**                   | $2.1301_{\pm 0.0224}$          | $\mathbf{0.0124_{\pm 0.0204}}$ | $0.1521_{\pm 0.0230}$          | $0.0478_{\pm 0.0280}$          |
| **DIS-LV**                   | $0.0682_{\pm 0.0081}$          | $0.0369_{\pm 0.0178}$          | $0.0362_{\pm 0.0167}$ | $0.0358_{\pm 0.0256}$          |
| **DPS (ours)**               | $\mathbf{0.0131_{\pm 0.0093}}$ | $\mathbf{0.0176_{\pm 0.0059}}$ | $0.0846_{\pm 0.0122}$ | $\mathbf{0.0273_{\pm 0.0113}}$ |

Table 2: Mixing proportions estimation error obtained by different methods.

| **Mixing proportions estimation error ($\downarrow$)** | $9$-Gaussians                  | Rings                          | Double-well                    |
| ------------------------------------------------------ | ------------------------------ | ------------------------------ | ------------------------------ |
| **LMC**                                                | $0.5199_{\pm 0.0159}$          | $0.6005_{\pm 0.0251}$          | $0.0673_{\pm 0.0082}$          |
| **RDMC**                                               | $0.1313_{\pm 0.0099}$          | $0.0537_{\pm 0.0035}$          | $0.2154_{\pm 0.0075}$          |
| **SLIPS**                                              | $0.0018_{\pm 0.0005}$          | $0.2471_{\pm 0.0144}$          | $0.1645_{\pm 0.0113}$          |
| **DIS**                                                | $0.7268_{\pm 0.0146}$          | $0.5233_{\pm 0.0194}$          | $0.0684_{\pm 0.0035}$          |
| **PIS**                                                | $0.4893_{\pm 0.0110}$          | $0.8016_{\pm 0.0194}$          | $0.0044_{\pm 0.0011}$          |
| **SVGD**                                               | $0.2098_{\pm 0.0097}$          | $0.1767_{\pm 0.0355}$          | $0.2400_{\pm 0.0174}$          |
| **HMC**                                                | $0.3766_{\pm 0.0154}$          | $0.4994_{\pm 0.0200}$          | $0.0159_{\pm 0.0046}$          |
| **PIS-LV**                                             | $0.4217_{\pm 0.0009}$          | $0.0010_{\pm 0.0013}$          | $\mathbf{0.0005_{\pm 0.0004}}$ |
| **DIS-LV**                                             | $0.0013_{\pm 0.0004}$          | $\mathbf{0.0007_{\pm 0.0004}}$ | $0.0012_{\pm 0.0009}$          |
| **DPS (ours)**                                         | $\mathbf{0.0006_{\pm 0.0003}}$ | $\mathbf{0.0006_{\pm 0.0006}}$ | $\mathbf{0.0004_{\pm 0.0002}}$ |

---

### Author Response · Authors · 2024-12-04
**Global response**

# Global response

---

We sincerely thank the reviewers for their valuable and constructive feedback. In response, we emphasize the novelty of our work, address shared concerns, and provide further clarification of our theoretical contributions as outlined below.

---

**Novelty**: Recently, sampling algorithms using PINN have garnered significant attention in research. However, we would like to highlight the unique contributions and novelty of our work in comparison to existing studies. Our approach leverages the reverse process of diffusion models for sampling, distinguishing it from related works ([3,4,5]) that rely on alternative paths, such as annealing path. Additionally, we identify the limitations of using the score Fokker-Planck equation for sampling from multi-modal distributions, specifically in **identifying mixing proportions for multi-modal distributions**, which is known as the blindness of score matching ([6]). This limitation demonstrates the importance of selecting the appropriate equation (log-density or score Fokker-Planck equation) for diffusion-based sampling. Moreover, our work provides a **comprehensive theoretical analysis of the proposed method and considers more challenging multi-modal distributions with unequal mixing proportions**, highlighting the uniqueness of our contributions in comparison to related studies ([7,8]). Notably, our work is concurrent with [8] and was submitted to NeurIPS 2024, where it received strong feedback, achieving an average review score of 6 and narrowly missing acceptance.

---

**Scalability**: The scalability of our approach primarily hinges on the performance of PINN in high-dimensional settings. Encouragingly, recent advancements ([1,2]) have introduced various methods that enhance PINN’s capacity to solve high-dimensional partial differential equations effectively. These advancements can be seamlessly integrated into our sampling framework, ensuring that our algorithm remains scalable and efficient even in higher-dimensional scenarios.

---

**Practical relevance of our theoretical results**: We clarify that our theoretical results for PINN demonstrate that if the PINN residual loss is sufficiently small, the approximation errors for both the log-density and the score will also be small. This establishes a solid theoretical foundation for our method, showing that minimizing the PINN residual to approximate the log-density can also yield a reliable approximation of the score.

---

References:

[1]. Zheyuan Hu, Zekun Shi, George Em Karniadakis, and Kenji Kawaguchi. Hutchinson trace estimation for high-dimensional and high-order physics-informed neural networks. Computer Methods in Applied Mechanics and Engineering, 424:116883, 2024a.

[2]. Zheyuan Hu, Khemraj Shukla, George Em Karniadakis, and Kenji Kawaguchi. Tackling the curse of dimensionality with physics-informed neural networks. Neural Networks, pp. 106369, 2024b.

[3]. Máté B, Fleuret F. Learning interpolations between boltzmann densities[J]. arXiv preprint arXiv:2301.07388, 2023.

[4]. Fan M, Zhou R, Tian C, et al. Path-Guided Particle-based Sampling[C]//Forty-first International Conference on Machine Learning.

[5]. Tian Y, Panda N, Lin Y T. Liouville Flow Importance Sampler[J]. arXiv preprint arXiv:2405.06672, 2024.

[6]. L. K. Wenliang. Blindness of score-based methods to isolated components and mixing proportions.
arXiv preprint arXiv:2008.10087, 2020.

[7].  Julius Berner, Lorenz Richter, and Karen Ullrich. An optimal control perspective on diffusion-based generative modeling. Transactions on Machine Learning Research, 2024.

[8]. Sun, Jingtong, et al. "Dynamical measure transport and neural PDE solvers for sampling." arXiv preprint arXiv:2407.07873 (2024).

---

### Meta-Review · Area_Chair_bi17 · 2024-12-19

**Metareview:**

This paper uses physics-informed neural networks to establish a diffusion-based sampling algorithm. There are two salient concerns from the reviewers about this submission. One is that physics informed neural networks (PINN) have already been used in diffusion-based samplers. It is not yet clear that this paper's modification / improvement is significant enough (measured in terms of effectiveness against existing benchmarks) to grant it a publication at ICLR. Another concern is about the scalability of PINN. I would suggest that the authors take another pass to address the above mentioned problems.

**Additional Comments On Reviewer Discussion:**

The authors have interacted with the reviewers during the rebuttal period. However, the above mentioned concerns have not yet been addressed.

---

### Decision · Program_Chairs · 2025-01-22

Reject